# Impacts of post-depositional processing on nitrate isotopes in the snow and the overlying atmosphere at Summit, Greenland

Zhuang Jiang[1], Joel Savarino[2], Becky Alexander[3], Joseph Erbland[2], Jean-Luc Jaffrezo[2], Lei Geng[1,4,5,6*]

[1]Anhui Province Key Laboratory of Polar Environment and Global Change, School of Earth and Space Sciences, University of Science and Technology of China, Hefei, Anhui, China

[2]Univ. Grenoble Alpes, CNRS, IRD, G-INP, Institut des Géosciences de l'Environnement, Grenoble, France

[3]Department of Atmospheric Sciences, University of Washington, Seattle WA, USA

[4]Laboratory for Ocean Dynamics and Climate, Pilot Qingdao National Laboratory for Marine Science and Technology, Qingdao, Shandong, China

[5]CAS Center for Excellence in Comparative Planetology, University of Science and Technology of China, Hefei, Anhui, China

[6]Hefei National Laboratory for Physical Sciences at the Microscale, University of Science and Technology of China, Hefei, Anhui, China

*Correspondence to*: Lei Geng (genglei@ustc.edu.cn)

## Abstract

The effect of post-depositional processing on the preservation of snow nitrate isotopes at Summit, Greenland remains a subject of debate and is relevant to the quantitative interpretation of ice core nitrate (isotopic) records at high snow

accumulation sites. Here we present the first year-round observations of atmospheric nitrate and its isotopic compositions at Summit, and compare them with published surface snow and snowpack observations. The atmospheric $\delta^{15}N(NO_3^-)$ remained negative throughout the year, ranging from –3.1 ‰ to –47.9 ‰ with a mean of (–14.8

± 7.3) ‰ (n = 54), and displayed minima in spring which is distinct from the observed spring $\delta^{15}N(NO_3^-)$ maxima in snowpack. The spring average atmospheric $\delta^{15}N(NO_3^-)$ was (−17.9 ± 8.3) ‰ (n = 21), significantly depleted compared to the snowpack spring average of (4.6 ± 2.1) ‰, while the surface snow $\delta^{15}N(NO_3^-)$ of (−6.8 ± 0.5) ‰ was in between the atmosphere and the snowpack. The differences in atmospheric, surface snow and snowpack $\delta^{15}N(NO_3^-)$ are best explained by the photo-driven post-depositional processing of snow nitrate, with potential contributions from fractionation during nitrate deposition. In contrast to $\delta^{15}N(NO_3^-)$, the atmospheric $\Delta^{17}O(NO_3^-)$ was of similar seasonal pattern and magnitude of change to that in the snowpack, suggesting little to no changes in $\Delta^{17}O(NO_3^-)$ from photolysis, consistent with previous modeling results. The atmospheric $\delta^{18}O(NO_3^-)$ varied similarly as atmospheric $\Delta^{17}O(NO_3^-)$, with summer low and winter high values. However, the difference between atmospheric and snow $\delta^{18}O(NO_3^-)$ was larger than that of $\Delta^{17}O(NO_3^-)$. We found a strong correlation between atmospheric $\delta^{18}O(NO_3^-)$ and $\Delta^{17}O(NO_3^-)$ that is very similar to previous measurements for surface snow at Summit, suggesting that that atmospheric $\delta^{18}O/\Delta^{17}O(NO_3^-)$ relationships were conserved during deposition. However, we found the linear relationships between $\delta^{18}O/\Delta^{17}O(NO_3^-)$ were significantly different for snowpack compared to atmospheric samples. This likely suggests the oxygen isotopes are also affected before preservation in the snow at Summit, but the degree of change for $\delta^{18}O(NO_3^-)$ should be larger than that of $\Delta^{17}O(NO_3^-)$. This is because photolysis is a mass-dependent process that would directly affect $\delta^{18}O(NO_3^-)$ in snow but not $\Delta^{17}O(NO_3^-)$ as the latter is a mass-independent signal. Although there were uncertainties associated with the complied dataset, the results suggested that post-depositional processing at Summit can induce changes in nitrate isotopes especially $\delta^{15}N(NO_3^-)$, consistent with a previous modeling study. This reinforces the importance of understanding the effects of post-depositional processing before ice-core nitrate isotope interpretation, even for sites with relatively high snow accumulation rate.

## 1. Introduction

Ice-core nitrate and its isotopes are potential proxies to constrain atmospheric variability of $NO_x$ and oxidants concentrations in past atmospheres. However, this can be compromised by the impacts of post-depositional processing on nitrate concentrations and isotopes (i.e., $\delta^{15}N$, $\delta^{18}O$ and $\Delta^{17}O$, where $\Delta^{17}O = \delta^{17}O - 0.52 \times \delta^{18}O$) (Alexander et al., 2020; Erbland et al., 2013; Röthlisberger et al., 2002; Wolff et al., 2008). Nitrate is chemically reactive in snow upon exposure to sunlight and thus its deposition to snow is not irreversible (Grannas et al., 2007). Numerous studies across Greenland and Antarctica have observed decreases in snow nitrate concentrations with depth in the snowpack (Erbland et al., 2013; Frey et al., 2009; Mulvaney et al., 1998; Röthlisberger et al., 2000) and/or emissions of $NO_x$ and HONO from snowpack (Barbero et al., 2021; Dibb et al., 1998; Frey et al., 2015; Honrath et al., 2002; Jones et al., 2001). Follow-up studies further indicate changes in the isotopic compositions of snow nitrate in the snowpack, e.g., increases in $\delta^{15}N$ and decreases in $\delta^{18}O/\Delta^{17}O$ with depth or an increasing trend in $\delta^{15}N$ from costal to inland sites (Blunier et al., 2005; Curtis et al., 2018; Erbland et al., 2013; Frey et al., 2009; Shi et al., 2015). Processes leading to such changes were referred to as post-depositional processing, and $\delta^{15}N$ of the archived nitrate was used to reflect the degree of post-depositional processing due to its high sensitivity to these processes (Erbland et al., 2013; Frey et al., 2009; Geng et al., 2015; Jiang et al., 2021; Shi et al., 2015; Winton et al., 2020).

Post-depositional processing of snow nitrate is mainly initiated by photolysis (Berhanu et al., 2014; Erbland et al., 2013; Frey et al., 2009; Zatko et al., 2016). The evaporation of nitrate from snow grains may also contribute, but this process has been suggested to have a minimal effect under typical ranges of temperatures in polar regions (Shi et al., 2019). Observations and modelling of snowpack nitrate concentration and isotope profiles across many different sites (e.g., Summit in Greenland, Dronning Maud Land (DML) and Dome A/Dome C in Antarctica) generally agree that photolysis dominates post-depositional processing (Erbland et al., 2013; Frey et al., 2009; Jiang et al., 2021; Winton et al., 2020; Shi et al., 2015). Photolysis of snow nitrate would emit NOx to the overlying atmosphere, which would subsequently reform nitrate under local

oxidation conditions and redeposits. This recycling of snow nitrate not only changes the initially deposited nitrate (isotope) signal, but also leads to a redistribution of snowpack nitrate. Thus, the final archival snow nitrate, defined as nitrate buried below the photic zone, would be largely impacted by post-depositional processing, which needs to be fully understood to interpret ice-core nitrate records. The degree of the photo-driven post-depositional processing is influenced by three main factors including snow accumulation rate, surface actinic flux and light penetration depth in snow (i.e., the photic zone where actinic flux decreases exponentially) (Zatko et al., 2013). Snow and ice-core nitrate isotope records have shown variations in $\delta^{15}N(NO_3^-)$ in response to varying snow accumulation rate as well as light-absorbing impurities (e.g., BC, dust, etc.) that influences light penetration depth in snow. For example, Geng et al. (2014) found correlations between $\delta^{15}N(NO_3^-)$ and snow accumulation rate across the GISP2 ice core record except in periods with very low snow accumulation rate (<0.08 m ice a$^{-1}$) and high dust concentrations. In the latter situation, $\delta^{15}N(NO_3^-)$ became negatively correlated with dust concentration. These correlations reflect the effect of snow accumulation rate and snow light absorbing impurities on the degree of post-depositional processing, respectively. The higher dust concentration during glacial periods could also reduce the volatilization of snow nitrate (Röthlisberger et al., 2000). At the West Antarctica ice sheet divide, where snow accumulation rate is high (0.24 m ice a$^{-1}$) at present, a decreasing trend in snow accumulation rate since 2400 yr BP led to an increasing trend in the degree of post-depositional processing as indicated by the elevated $\delta^{15}N(NO_3^-)$ (Sofen et al., 2014).

Variations in surface actinic flux (especially the UVB radiation) would also induce changes in the degree of post-depositional processing and leave signals in the preserved nitrate in snow and ice cores. Previous studies (Erbland et al 2013; Frey et al., 2009; McCabe et al. 2007) proposed that $\delta^{15}N(NO_3^-)$ preserved in snow and ice cores may serve as a proxy of total column ozone (TCO) due to its influence on surface UVB radiation, while a recent study suggested the preserved $\delta^{15}N(NO_3^-)$ is more sensitive to snow accumulation rate and light penetration depth than to changes in TCO (Winton et al., 2020). Nevertheless, in periods with relatively constant snow accumulation rate but

distinct surface actinic flux, e.g., the switch of the polar night and polar day over a year, and the Antarctic ozone hole period, changes in the degree of post-depositional processing and the associated isotope effects should be expected. Using a snow column photochemical model (the TRANSITS model by Erbland et al. (2015)), Jiang et al. (2021) explicitly quantified the effects of post-depositional processing on snow nitrate and its isotopes on a seasonal scale at Summit, Greenland. Owing to the seasonal differences in surface actinic flux, the model predicted a seasonal variation in $\delta^{15}N(NO_3^-)$ snowpack similar to the observations. On an annual scale, the model predicted a $\approx 4$ % net nitrate mass loss, which is within the range estimated by previous studies (Burkhart et al., 2004; Dibb et al., 2007) but is subject to uncertainties in the fraction of the snow-sourced nitrate exported from the region. In contrast, the model predicted minimum changes in $\Delta^{17}O$ of snow nitrate on both seasonal and annual scales because the photo-driven post-depositional processing affects $\Delta^{17}O$ mainly from the cage effect (i.e., the intermediate photo-products ($NO_2^-$ and $NO_2$) exchange with water oxygen or react with radicals such as OH in snow grains to regenerate nitrate before being emitted to the atmosphere) (McCabe et al., 2005; Meusinger et al., 2014), and the cage effect is minimum at Summit given the high snow accumulation. The study by Jiang et al. (2021) further suggested that seasonal $\delta^{15}N(NO_3^-)$ variations in the snowpack at Summit, Greenland is caused by photo-driven post-depositional processing, an alternative to previous interpretations that attributed the seasonality to $NO_x$ source variability (Hastings et al. 2004). Jarvis et al. (2009) also found enrichment in snowpack $\delta^{15}N(NO_3^-)$ compared to the surface snow samples at Summit, Greenland, providing observational evidence of post-depositional processing altering snow $\delta^{15}N(NO_3^-)$ at this high snow accumulation rate site. These results are in conflict with the conclusion of Fibiger et al. (2013, 2016) who suggested that there is little to no isotope effect caused by post–depositonal processing relying on the oxygen isotopes of nitrate. However, as argued by Jiang et al. (2021), the nitrogen isotopes are more sensitive to post-depositional processing. In addition, Fibiger et al. (2013, 2016) collected atmospheric and surface snow samples in May and June. The process of photolysis of snow nitrate to $NO_x$, oxidation of snow-sourced $NO_x$ to nitrate, followed

by re-deposition of snow-sourced nitrate will render the isotopic composition of atmospheric and surface snow nitrate similar to each other. Nitrate at depth, isolated

from surface deposition but still in the photic zone, would continue to experience photolysis, making post-depositional loss more apparent in the isotope observations. Therefore, in order to reflect the full picture of post-depositional processing, snow samples covering the entire photic zone (~ 40 cm at Summit) must be considered (Jiang et al., 2021).

To thoughtfully evaluate the effects of post-depositional processing at Summit, Greenland, and to verify the modeling results by Jiang et al. (2021), nitrate isotopes in the atmosphere and in snow covering a full cycle of polar seasons with distinct actinic flux variations are necessary. Here, we present the first year-round observations of nitrate isotopes in the air at Summit, and compare them with similar observations in

surface snow and in snow at depth (i.e., snowpack) to conduct a comprehensive evaluation on the seasonality in nitrate isotopes in both air and snow, as has already been done in Antarctica (Erbland et al., 2013; Frey et al., 2009; Winton et al., 2020). These observations provide information regarding the evolution of nitrate isotopes from atmospheric nitrate to its final preservation in snowpack, which is critical for assessing

the post-depositional changes of nitrate isotopes.

## 2. Methods
### 2.1 Atmospheric nitrate sampling and measurements

From July 2001 to July 2002, atmospheric samples were collected at Summit,
Greenland using a high-volume air sampler (HVAS) with glass fiber filters ($20.3 \times 25.4$ cm). All glass fiber filters were pre-cleaned by an overnight soak and several rinses with ultra-pure water, then dried in a clean room and stored in clean plastic food storage bags until used. Glass fiber filters have been shown to be capable of collecting atmospheric nitrate with high efficiency even when gas phase nitrate dominates total

atmospheric nitrate (Erbland et al., 2013). This is likely due to the high NaCl blank in the glass fiber filter, which is known to promote collection efficiency of atmospheric nitrate (Morin et al., 2009; Erbland et al., 2013). The quantitative collection of

atmospheric nitrate is further supported by the similar concentration range of our measurements with previous Summit studies (SI). In this study we assumed that the collected filtered nitrate sample representing the total atmospheric nitrate in the passed air, i.e., the sum of aerosol nitrate and gas phase nitric acid. Each sample covering 3-4 days were routinely collected over the year, with a total of 97 samples. We have also collected 9 field blanks during the sampling period in different months, with the same sampling procedure but limited the sampling time to 1 minute. These samples were stored frozen until analysis.

Measurements of nitrate concentrations and isotopes were conducted in the laboratory at the Institute des Géosciences de l'Environnement, Grenoble, France in 2013. Nitrate collected on the glass fiber filters was first extracted by about 40 ml of 18 MΩ water via centrifugation using Millipore Centricon™ filter units. The samples were then measured for nitrate concentrations by colorimetry using the Saltzman method (Vicar et al., 2012). The average nitrate concentration in the filtrate for all atmospheric samples was $(1363 \pm 1603)$ ng g$^{-1}$, while that of the nine blank samples was $(183 \pm 44)$ ng g$^{-1}$. Among these samples, 54 out of 97 were determined to be valid by comparing the extracted nitrate concentration with the blank, i.e., only samples with concentrations exceed 3 times of the blank samples were judged as valid for further analyses. These samples were then individually concentrated on a 0.3 mL resin bed with anionic exchange resin (Bio-Rad™ AG1-X8, chloride form) and eluted with $5 \times 2$ mL of NaCl solution (1M). The isotopic compositions of each sample were determined by using the bacterial denitrifier method. Briefly, $NO_3^-$ in each sample was converted to $N_2O$ by denitrifying bacteria under anaerobic conditions. $N_2O$ was then thermally decomposed into $N_2$ and $O_2$ on a gold tube heated at 800 °C. The $N_2$ and $O_2$ were separated by a gas chromatography column and injected into an isotope ratio mass spectrometer (Thermo Finnigan™ MAT 253) for isotope analyses of $^{15}N/^{14}N$, $^{17}O/^{16}O$ and $^{18}O/^{16}O$. To correct for the potential isotope fractionation during laboratory isotope analysis, international reference materials (IAEA-NO$_3$, USGS-32, 34 and 35) were used for data calibration. We treated the reference materials the same as the filtrations from filter samples, e.g., making the reference material solution using

1M NaCl solution. The blank filter samples were processed following the same procedure as atmospheric samples and measured for their isotope ratios. The measured nitrate isotope ratio of each atmospheric sample was further corrected by deducting the contribution of the filter blanks. The measurement uncertainty was assessed based on the reduced standard deviations of the residuals from the linear regression between the measured reference materials and their expected values as detailed in Erbland et al. (2013). The overall measurement uncertainties were estimated to be 0.6 ‰ for $\delta^{18}O$, and 0.3 ‰ for both $\Delta^{17}O$ and $\delta^{15}N(NO_3^-)$.

**2.2 Atmosphere, surface snow and snowpack data compilation**

From the literature, we collected nitrate isotope data ($\delta^{15}N$, $\delta^{18}O$ and $\Delta^{17}O$) of atmospheric particulate or gas-phase nitrate, surface snow and snowpack nitrate available at Summit, Greenland (Fibiger et al., 2013; Fibiger et al., 2015; Geng et al., 2014; Hastings et al., 2004; Jarvis et al., 2009; Kunasek et al., 2008). Details about these data (e.g., sample type, depth, age, sampling technique) and the corresponding references are listed in Table 1. Note that in some early publications only the seasonal averages instead of the original data with finer resolution were available. These data were compiled to produce a dataset including all seasons for nitrate in the air, surface snow and snowpack by averaging samples covering multiple years and/or by different groups to reduce the spatial and temporal heterogeneities. For samples with resolution finer than monthly, we compiled them as their mass-weighted monthly averages (if the mass information for each sample is known), and for samples with coarser than monthly resolution, seasonal averages were used, and we here reported seasonal averages of multiple years if more than one year's data are available in the literature.

For atmospheric and surface snow samples, age information was indicated as the time of sampling. Snowpack samples require a conversion from depth to age. The snowpack samples from Hastings et al. (2004) and Kunasek et al. (2008) were dated by seasonal binning according to the measured accumulation rate and water isotopes, and their age information was used as is. For samples from Geng et al. (2014), we recalculated the dating by bamboo stake measured snow accumulation data (Burkhart

et al., 2004; Dibb et al., 2004; Kuhns et al, 1997) constrained by snow density and further justified by seasonal peaks of $Na^+$ and $Cl^-/Na^+$ ratio. This is similar to the dating method in Hastings et al. (2004) and the only difference is which proxy was used as the seasonal marker. Briefly, we used the bamboo stake measurements of weekly snow accumulation at Summit and the snowpack density profile to estimate the deposition timing of each samples in the 2.1 m snowpack that was collected in July of 2007. We first converted the thickness of each sample (referred to as $D_m$) to a fresh snow thickness (referred to as $D_f$) by the following equation:

$$D_f = D_m \times \frac{\rho_m}{\rho_f} \tag{1}$$

Where $\rho_m$ is the real snow density at each depth from field measurement (Geng et al., 2014), and $\rho_f$ is the fresh snow density (0.32 g cm$^{-3}$; Dibb et al., 2004). These fresh snow thicknesses were then stacked to construct an idealized snow depth profile without densification due to compaction and/or metamorphism. This idealized depth profile was then matched to the stacked depth by the observed average weekly snow accumulation rate to determine the exact age for each sample. A previous study showed that using the stack measured accumulation rate is capable of reconstructing the vertical profile of snowpack nitrate (Burkhart et al., 2004). This dating method has uncertainties, mostly owing to the large variability of measured accumulation rate among different stakes (Burkhart et al., 2004). To reduce the uncertainties in our dating results, we calculated their monthly average and compared with atmospheric and/or surface snow data with a similar or coarser time resolution. The compiled $\delta^{15}N$ and $\Delta^{17}O$ data in monthly resolution display seasonal patterns similar to their original seasonal variations observed in snowpack, and the $Cl^-/Na^+$ ratio of the compiled samples also displays summer high and winter low as has been previously observed snowpack or firn cores (Geng et al., 2014), corroborating the dating method in terms of capturing the seasonality (Figure 2e). The monthly $\Delta^{17}O$ values compiled from Geng et al. (2014) data were further averaged with the monthly $\Delta^{17}O$ values reported by Kunasek et al. (2008) to generate the final snowpack monthly $\Delta^{17}O$ data. In comparison to $\Delta^{17}O$, the $\delta^{18}O$ data from different groups indicated a much larger range of variability, or even

being inconsistent as the data from Jarvis et al (2009) indicated a winter peak of $\delta^{18}$O instead of summer which is different from other studies (e.g., Geng et al., 2014) and difficult to explain from the current understanding of nitrate chemistry. Therefore, we didn't average the $\delta^{18}$O data from different groups.

| | Isotopes | Period | Resolution | Depth (Method) | Reference |
|---|---|---|---|---|---|
| Atmosphere | $\delta^{15}$N/$\Delta^{17}$O/$\delta^{18}$O | July 2001 to July 2002 | 3-5 day | HVAS + GF | This study |
| | $\delta^{15}$N/$\Delta^{17}$O/$\delta^{18}$O | June to July 2010/2011 | 0.5-1 day | Mist chamber | Fibiger at al., 2016 |
| | $\delta^{15}$N/$\delta^{18}$O | March 2006 to Jul 2006 | >2 day | Mist chamber | Jarvis et al., 2009 |
| Surface snow | $\delta^{15}$N/$\delta^{18}$O | March 2006 to Jul 2006 | – | – | Jarvis et al., 2009 |
| | $\delta^{15}$N/$\Delta^{17}$O/$\delta^{18}$O | June to July 2010/2011 | 0.5-3 cm | 0.5-3 cm | Fibiger at al., 2016 |
| Snowpack | $\delta^{15}$N/$\Delta^{17}$O/$\delta^{18}$O | July 2004– July 2007 | 3-5 cm | 0-2.1 m | Geng et al., 2014 |
| | $\delta^{15}$N/$\delta^{18}$O | Spring 2000 to summer 2001 | 3 cm | 0-1 m | Hasting et al., 2004 |

| | | | | |
|---|---|---|---|---|
| $\delta^{15}N/\delta^{18}O$ | Summer 2005 to summer 2007 | 5 cm | 0-1 and 0-2 m | Jarvis et al., 2009 |
| $\Delta^{17}O$ | January 2004 to July 2007 | 5 cm | 0-2 m | Kunasek et al., 2008 |

**Table1**. Nitrate isotope data information and references. The atmospheric sampling technique in different studies is also listed in the table.

## 3. Results

### 3.1 Year-round atmospheric nitrate concentrations and isotopes at Summit, Greenland

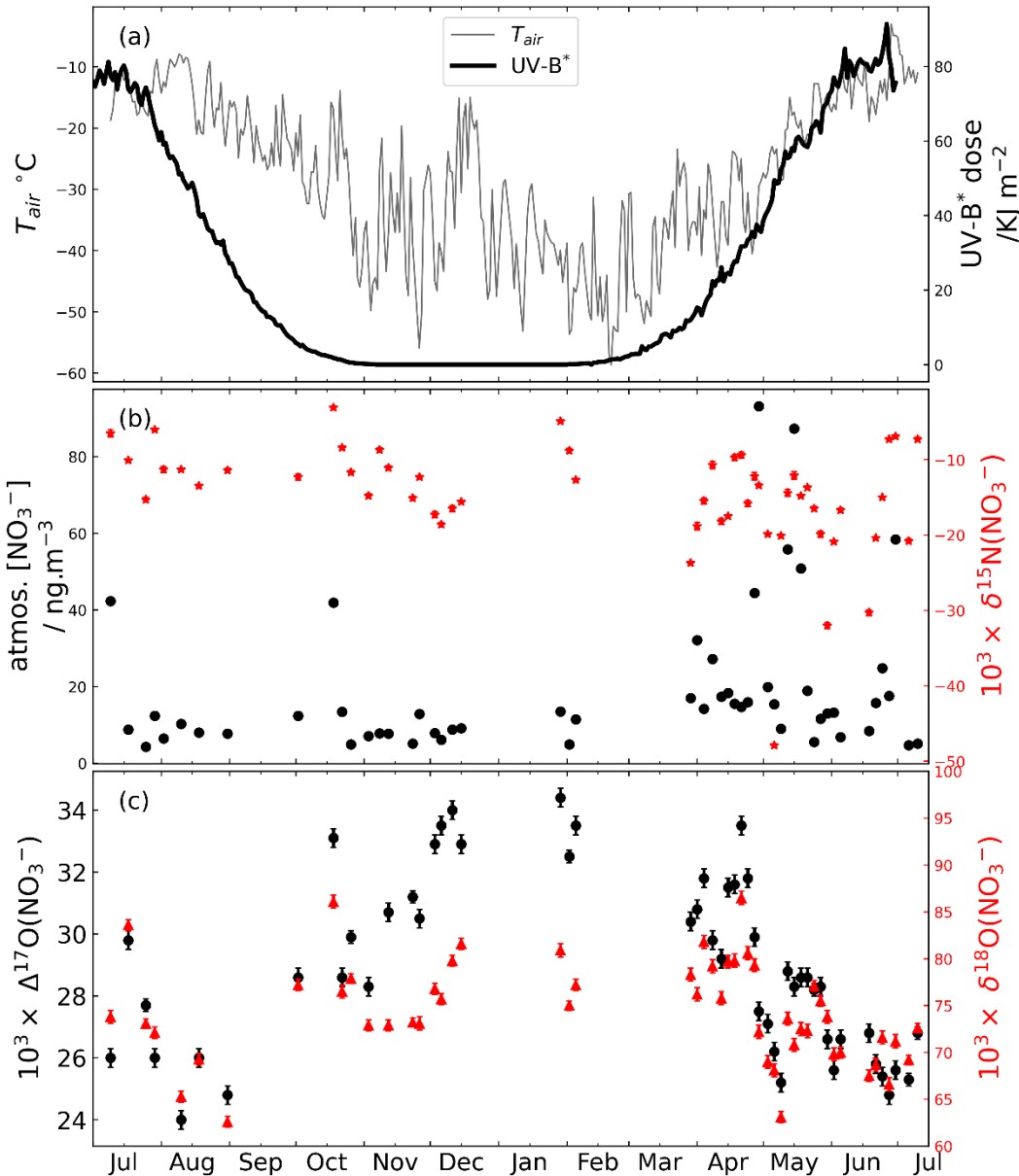

**Figure1.** Atmospheric nitrate concentrations and isotopes at Summit over the sampling period (July 2001 to July 2002). (**a**) Daily air temperature and observed UV-B* (280-320 nm) dose at Summit, Greenland from July 2001 to July 2002 (data source: NSF Arctic Data Center, link: https://arcticdata.io/catalog/data, last access: 13 June 2022), (**b**) $NO_3^-$ concentration (black circle) and $\delta^{15}N(NO_3^-)$ (red star), (**c**) $\Delta^{17}O(NO_3^-)$ (black circle) and $\delta^{18}O(NO_3^-)$ (red triangle).

The measured nitrate concentrations and its isotopic compositions ($\delta^{15}$N, $\Delta^{17}$O and $\delta^{18}$O) in the filter samples are shown in Figure 1, together with surface air temperature and UV-B* level (wavelength ranges from 280 to 320 nm) measured at Summit station. As shown in the figure, the annual mean atmospheric $NO_3^-$ concentration was (19.9 ± 19.1) ng m$^{-3}$ and most of them ranged from ~ 1 to 95 ng m$^{-3}$, consistent with the values reported by previous studies at Summit (SI). There was no distinct seasonal pattern in atmospheric nitrate concentrations, but some spikes (samples with much higher nitrate concentrations than average) in spring/summer months were observed, which are typical to intrusion of Arctic haze events at the altitude of the Ice Sheet (Quinn et al., 2007; Jaffrezo et al., 1997). Alternatively, these nitrate concentration spikes could reflect a more efficient scavenging of atmospheric nitrate by sea salt aerosol during transport, as indicated by the elevated $Na^+$ concentration in Summit aerosol during April and May (Rhodes et al., 2017).

The atmospheric $\delta^{15}$N($NO_3^-$) was negative throughout the year with an annual mean of (–14.8 ± 7.3) ‰. The springtime atmospheric $\delta^{15}$N($NO_3^-$) exhibited a significantly lower shift comparing to other seasons (two-sided t-test, p = 0.001), and the average for the winter half year was (–12.0 ± 4.2) ‰ slightly higher than that (–16.0 ± 3.9) ‰ in the summer half year. The mean atmospheric $\delta^{15}$N($NO_3^-$) from May to June was (–19.3 ± 9.6) ‰, close to the value of (–16.8 ± 8.7) ‰ reported by Fibiger et al. (2016) covering the same months. In addition, some extremely negative $\delta^{15}$N($NO_3^-$) values ($< -30$ ‰) were observed in spring/summer months. Such very low $\delta^{15}$N($NO_3^-$) values were also observed by Fibiger et al. (2016).

The atmospheric $\Delta^{17}$O ($NO_3^-$) values ranged from 24.0 ‰ to 34.4 ‰ with a seasonal minimum near mid-summer, concurrent with the maximum UV-B$^*$ radiation intensity (Figure 1a), and a peak in winter. The atmospheric $\delta^{18}$O($NO_3^-$) data ranged from 49.7 ‰ to 86.5 ‰ and displayed an almost identical seasonal pattern with $\Delta^{17}$O($NO_3^-$). The similar seasonality between $\delta^{18}$O($NO_3^-$) and $\Delta^{17}$O($NO_3^-$) is expected. At the seasonal scale, the primary controlling factor of atmospheric $\delta^{18}$O($NO_3^-$) and $\Delta^{17}$O($NO_3^-$) is the relative importance of $O_3$ versus $HO_x$ to nitrate formation in different seasons. In summer, HOx oxidation is more important and leads to nitrate with lower

$\delta^{18}O(NO_3^-)$ and $\Delta^{17}O(NO_3^-)$, while in winter $O_3$ oxidation is more important and leads to higher $\delta^{18}O(NO_3^-)$ and $\Delta^{17}O(NO_3^-)$ (Alexander et al., 2020; Michalski et al., 2012). The $\delta^{18}O(NO_3^-)$ values between March to June ranged from 63.1 to 86.5 ‰, much higher than the values (24 to 50‰) reported by Jarvis et al. (2009), and in the upper band of that (37.4 to 93.4 ‰) reported by Fibiger et al. (2016) over the same months (but in different years). Note that the Jarvis et al. (2009) and Fibiger et al. (2016) studies reported values for atmospheric gas-phase $HNO_3$ instead of bulk nitrate. Overall, the absolute values and the seasonal patterns of $\Delta^{17}O(NO_3^-)$ were similar to those observed in snowpack samples at Summit (Kunasek et al., 2008; Geng et al. 2014), while those of $\delta^{18}O(NO_3^-)$ were similar to that reported for snowpack samples by Hastings et al. (2004).

## 3.2 Compiled seasonal $\delta^{15}N$, $\delta^{18}O$ and $\Delta^{17}O$ in atmospheric, surface snow and snowpack nitrate

The compiled nitrate isotopes (i.e., $\delta^{15}N$/$\delta^{18}O$/$\Delta^{17}O$) with monthly or seasonal resolutions are plotted in Figure 2. These compiled data of atmospheric, surface snow and snowpack averages should represent the status of nitrate before deposition, after deposition, and archival, respectively. To validate our dating results on the snowpack data, we also plotted the resampled monthly snowpack $Na^+$ concentration and $Cl^-/Na^+$ ratio. As shown in Figure 2e, the $Na^+$ concentration and $Cl^-/Na^+$ ratio displayed clear winter and summer peak, respectively, indicating a general reliability of our dating method. We also calculated the accumulated UV-B[*] daily dose for nitrate deposited in different weeks of a year using Eq (2):

$$UVB^*_{toal} = \sum UVB^*(t) * \exp\left(-\sum \frac{A(t)}{z_e}\right) \qquad (2)$$

where $A$(t) and $z_e$ represent the weekly snow accumulation rate and e-folding depth (12.3 cm, Jiang et al. (2021)) at Summit, respectively. The daily UV-B[*](t) dose was shown in Figure 2a. The accumulated UV-B[*] dose computed here represents the integrated UV-B[*] radiation that snow nitrate received from being deposited to surface snow until being buried below the photic zone ($\approx 40$ cm according to Jiang et al. (2021)).

This gives a first order estimation of the total radiation (i.e., the degree of post-depositional processing) that the archived nitrate experienced at Summit.

The snowpack samples from Geng et al. (2014) cover ~ 3 years snow accumulation, and we averaged the monthly data of the three years for each month. As shown in Figure 2b, its seasonal $\delta^{15}N(NO_3^-)$ variation displays an overall good agreement with that

reported by Hastings et al. (2004) and Jarvis et al. (2009) with a spring peak. In general, the $\delta^{15}N(NO_3^-)$ data among different sample types indicated a systematic pattern for spring/summer samples, with the atmospheric samples the most depleted $(-16.0 \pm 7.9)$ ‰ and the snowpack samples the most enriched $(2.7 \pm 3.0)$ ‰, while the surface snow samples were in between $(-5.8 \pm 0.7)$ ‰. In addition, the snowpack $\delta^{15}N(NO_3^-)$ data

indicated a clear spring/summer maximum coincident with the maximum accumulated UV-B[*] dose (Figure 2a), while the surface snow $\delta^{15}N(NO_3^-)$ were only moderately enriched in spring/summer compared to other seasons. For atmospheric $\delta^{15}N(NO_3^-)$, although uncertainties of the monthly averages were large, they were moderately depleted in spring/summer compared to other seasons, opposite to the surface snow and

snowpack data. In addition, for fall and winter seasons, the $\delta^{15}N(NO_3^-)$ values of different sample types converged, opposite to their behaviors in spring/summer when they diverged. Note that the atmospheric samples from Jarvis et al (2009) collected in April and May were for gas-phase $HNO_3$, and their $\delta^{15}N(NO_3^-)$ values were higher than that in atmospheric nitrate measured by this study, but within the range of those in

surface snow, and lower than those in snowpack.

The compiled monthly $\Delta^{17}O(NO_3^-)$ values are shown in Figure 2c. Atmospheric $\Delta^{17}O(NO_3^-)$ values were consistent with the snowpack values throughout the year, and both atmospheric and snowpack $\Delta^{17}O(NO_3^-)$ reached a seasonal minimum in summer. Surface snow $\Delta^{17}O(NO_3^-)$ were only available in May and June as reported by Fibiger

et al. (2016), and although highly variable, their averages were consistent with the May and June $\Delta^{17}O(NO_3^-)$ averages in the atmosphere and snowpack.

The compiled $\delta^{18}O(NO_3^-)$ results are shown in Figure 2d. Although the summer minimum for the snowpack data from Geng et al. (2014) was not as obvious as those reported by Hastings et al. (2004) and the atmospheric data reported in this study, the

$\delta^{18}O(NO_3^-)$ data in general indicated a summer minimum. In comparison, the surface

snow and snowpack data from Jarvis et al. (2009) indicated a fall minimum, and the

original data of Jarvis et al. (2009) indicated a clear summer maximum in the year of

2005. These data are nevertheless difficult to interpret given the current understanding

of nitrate formation mechanisms which should lead to a summer low and winter high

for $\delta^{18}O(NO_3^-)$. But we note that caution should be taken when interpreting the Jarvis

et al. (2009) data, as there was a large difference in $\delta^{18}O(NO_3^-)$ data from one winter

(69.5 ± 5.0) ‰ (n = 7) to the next (101.1 ± 7.9) ‰ ($n$ = 4). In addition, the averaged

$\delta^{18}O(NO_3^-)$ of atmospheric nitrate in gas-phase samples collected by Jarvis et al. (2009)

in March and June is (34.1 ± 1.7) ‰, and by Fibiger et al. (2016) in May and June is

(54.2 ± 8.5) ‰ for the year of 2010 and (90.5 ± 12.5) ‰ for the year of 2011. These

values are out of range of the snow samples as well as our atmospheric samples, and in

order to better show the seasonality of $\delta^{18}O(NO_3^-)$ in snow and atmospheric samples as

indicated by other data, we didn't plot these data in Figure 2d. Overall, the $\delta^{18}O(NO_3^-)$

data were more variable than the $\Delta^{17}O(NO_3^-)$ data and there were inconsistences among

different observations.

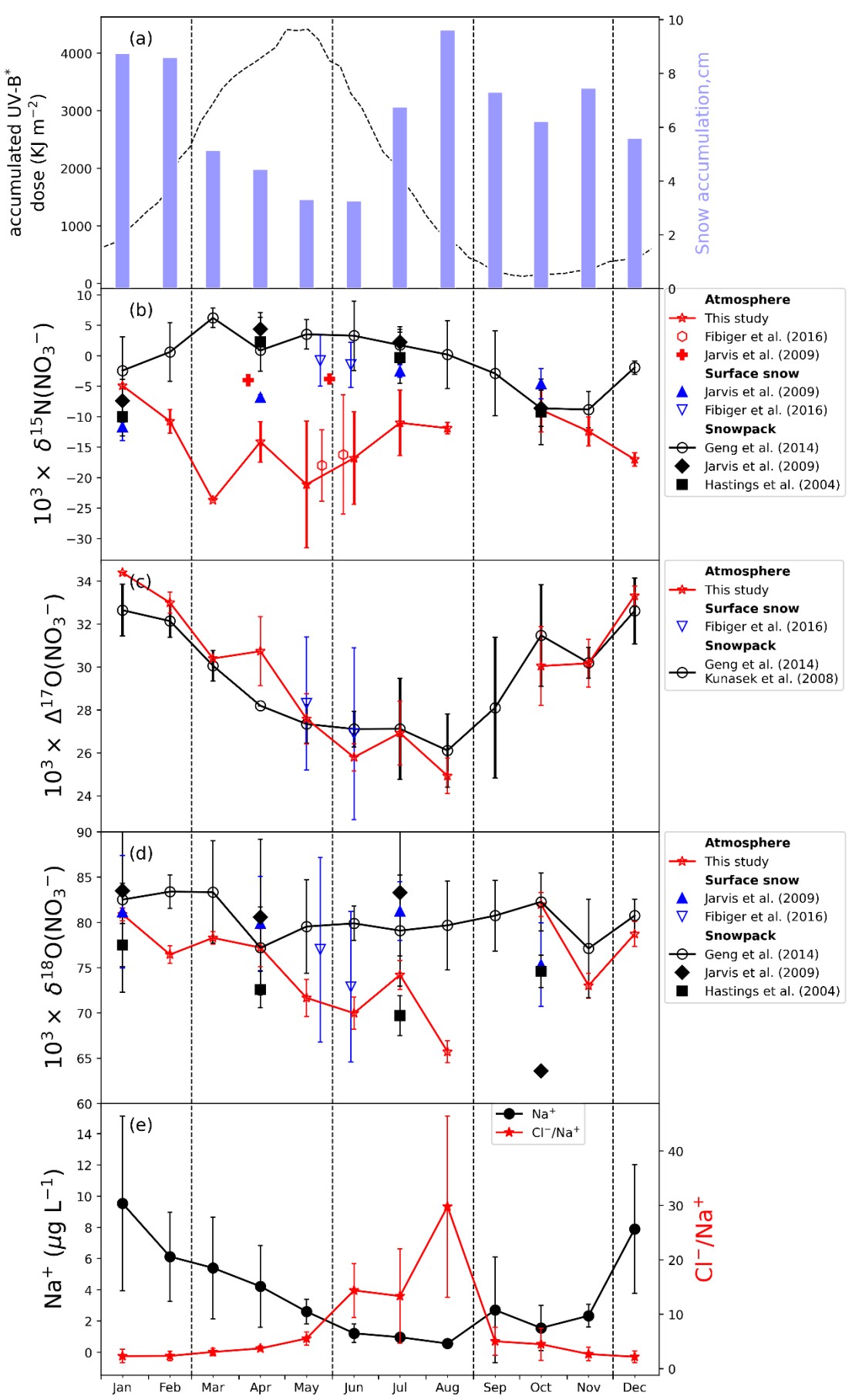

**Figure 2**. (**a**) Cumulative UV-B* dose as function of the deposited time (black dashed line) calculated according to Eq (2) and monthly snow accumulation in cm (bar plot). The cumulative UV-B* dose represents the actinic dose that would have been experienced by snowpack nitrate deposited in different times. (**b-d**) Compiled monthly and/or seasonal atmospheric, surface snow and snowpack $\delta^{15}N/\Delta^{17}O/\delta^{18}O(NO_3^-)$ data at Summit, Greenland. (**e**) Compiled monthly snowpack $Na^+$ concentration and $Cl^-/Na^+$ ratio. The seasonal data (represented by the filled markers) were plotted against the central month of each season. The monthly data were represented by the hollow markers. The vertical lines represent the interval of seasons. The error bar represents one standard error for the monthly or seasonal mean. Data sources were summarized in Table 1. The atmospheric $\delta^{18}O(HNO_3)$ data in Fibiger et al. (2016) and Jarvis et al. (2009) are both out of range of the snow samples as well as our atmospheric samples thus are not shown here.

## 4. Discussion

### 4.1 Seasonal $\delta^{15}N(NO_3^-)$ and its difference between atmospheric, surface snow and snowpack nitrate

The atmospheric, surface snow and snowpack samples represent different stages of nitrate in the deposition and preservation processes. The compiled results in Figure 2b indicated a systematic enrichment in $\delta^{15}N(NO_3^-)$ from deposition to preservation for spring/summer nitrate. This systematic enrichment refutes the previous hypothesis that seasonal variation in snowpack $\delta^{15}N(NO_3^-)$ at Summit was driven by shifts in the relative importance of $NO_x$ sources (Hastings et al., 2004). Instead, local processes leading to fractionations in $\delta^{15}N(NO_3^-)$ are needed to reconcile the observed differences between atmospheric and snowpack $\delta^{15}N(NO_3^-)$. Previous studies suggest there were several processes occurring at the air-snow interface related to nitrate deposition and preservation that could lead to nitrogen fractionation, including (i) fractionations during snow nitrate photolysis and physical release (Berhanu et al., 2014; Erbland et al., 2013; Frey et al., 2009; Jiang et al., 2021; Shi et al., 2019), and (ii) the proposed fractionation during nitrate deposition related to the different deposition mechanisms (Erbland et al.,

2013). Jiang et al. (2021) have discussed the effect of the physical release on nitrate isotopes and suggest that this effect is negligible at Summit. This is because that the physical release rate and the associated isotope effects are relatively small at cold temperatures. Shi et al. (2019) performed field $NO_3^-$ volatilization experiments and found no isotope fractionation occurring in $\delta^{15}N(NO_3^-)$ when the temperature was set to –24 °C. When the temperature increased to –4 °C, a small positive fractionation constant (4.9 ± 2.1‰) was observed, while at Summit the temperature is below –10 °C throughout the year as shown in Figure 1a. In the following sections, we discuss the other processes and compare with the modeling study results from Jiang et al. (2021), to discern the exact cause(s) of the observed systematic changes in $\delta^{15}N(NO_3^-)$ from the atmosphere to snowpack.

### 4.1.1 The effects of snow nitrate photolysis

The $\delta^{15}N(NO_3^-)$ pattern in the summer half year among different types of samples, i.e., atmospheric $\delta^{15}N(NO_3^-)$ < surface snow $\delta^{15}N(NO_3^-)$ < snowpack $\delta^{15}N(NO_3^-)$, is qualitatively consistent with the effects of snow nitrate photolysis which enriches snow $\delta^{15}N(NO_3^-)$ while providing a snow-source of depleted $\delta^{15}N(NO_3^-)$ to the atmosphere. In fact, the negative isotope fractionation factor associated with nitrate photolysis would favor the release of NOx with lighter $^{14}N$, which would rapidly reform nitrate in the overlying atmosphere given the short lifetime of NOx at Summit (typical several hours in summer). The snowpack $\delta^{15}N(NO_3^-)$ variations within a year showed a similar trend with the accumulated UV-B[*] dose (Figure 2a and 2b), i.e., the $\delta^{15}N(NO_3^-)$ peak and valley corresponded to the seasons with the highest (i.e., spring) and the lowest (i.e., fall) accumulated UV-B[*] dose, respectively. The accumulated UV-B[*] dose reflects the total amount of radiation leading to photolysis (wavelength of 280 to 320 nm) that snow nitrate received before archival for a given snow layer. In contrast, during the winter half year when there is an absence of sunlight, $\delta^{15}N(NO_3^-)$ among different types of samples are similar, suggesting that the physical transfer between atmosphere and snowpack (deposition, evaporation) leads to negligible $^{15}N$ isotopic fractionations.

The atmospheric $\delta^{15}N(NO_3^-)$ in the summer half year should represent the combined signal of primary nitrate from long-range transport and the snow-sourced nitrate from photolysis (Jiang et al., 2021), while in winter atmospheric $\delta^{15}N(NO_3^-)$ should be less influenced by snow-sourced nitrate and perhaps dominated by primary nitrate. Snow-sourced atmospheric nitrate is very depleted in $\delta^{15}N(NO_3^-)$ (< -70 ‰ at Summit, Jiang et al., 2021), and its flux to the overlying atmosphere should maximize in summer when surface UV radiation is the strongest. All else being equal, one should expect the summer atmospheric $\delta^{15}N(NO_3^-)$ to be the lowest throughout the year. This appears to be in conflict with the observations which indicated the spring atmospheric $\delta^{15}N(NO_3^-)$ was the lowest. Possible explanations for this could be related to spring-summer differences in the export fraction of the snow-sourced nitrate or the $\delta^{15}N(NO_3^-)$ of primary nitrate. Cohen et al. (2006) conducted studies on the boundary layer dynamics at Summit and found that sustained stable surface layer conditions were frequently observed during spring at Summit, while in summer the boundary layer became more convective. The more stable boundary layer conditions in spring may lower the export fraction of the snow-sourced nitrate compared to summer, which tends to lower the spring atmospheric $\delta^{15}N(NO_3^-)$ as more snow-sourced nitrate with extremely low $\delta^{15}N$ will accumulate in the local boundary layer. Honrath et al. (2002) found that at Summit, in summer the snow-sourced nitrate (their measured form was $NO_x$) was not balanced by downward $HNO_3$ flux and suggested that without wet deposition the emitted $NO_x$ and reformed $HNO_3$ should be largely exported from the local boundary layer. In addition, Jiang et al. (2021) suggested that the primary nitrate flux dominates the nitrate budget at Summit, and even in mid-summer the snow-sourced nitrate only accounts for about 25% of total atmospheric nitrate. If $\delta^{15}N$ of primary nitrate in summer was higher than that of primary nitrate in spring, the local atmospheric $\delta^{15}N(NO_3^-)$ at Summit could be still higher in summer even when the contribution of snow-sourced nitrate was larger. Other possible explanations could be (i) the area of snow cover in the Arctic basin is larger in spring than summer, which acts to increase the snow-sourced nitrate with depleted $\delta^{15}N(NO_3^-)$ and may offset the effects of higher summer actinic flux on snow nitrate photolysis; (ii) the planetary

boundary layer in summer is probably higher than that in spring at Summit, so the effects of snow-sourced nitrate on atmospheric nitrate budget is greater in spring than in summer.

To better understand the effects of the photo-driven post-depositional processing, we quantitatively compared and analyzed the $\delta^{15}N(NO_3^-)$ averages in spring when the isotopic differences between surface snow and snowpack are the most pronounced as indicated by the compiled data and the modeling results by Jiang et al. (2021). Since the surface snow $\delta^{15}N(NO_3^-)$ data in Fibiger et al. (2016) only covered two months, we mainly focus on the seasonal data covering two years from Jarvis et al. (2009). However, we note the surface snow $\delta^{15}N(NO_3^-)$ data in Fibiger et al. (2016) was remarkably higher than that in Jarvis et al. (2009) for the same months, which likely indicated the heterogeneity among data from different years. Compared to surface snow nitrate, snowpack nitrate was enriched by (12.8 ± 2.6) ‰ in spring in our compiled dataset, as seen in Fig 2b. This value should reflect the effect of post-depositional processing on snow nitrate throughout its preservation, i.e., from being deposited in the surface to being archived below the photic zone. In Jiang et al. (2021), this effect was defined as PIE, i.e., the photo-induced isotope effect, and calculated as the difference between surface snow $\delta^{15}N(NO_3^-)$ and archived snow $\delta^{15}N(NO_3^-)$. The PIE in spring calculated by the TRANSITS model is averaged at (14.3 ± 1.1) ‰, consistent with the compiled data. Calculating the PIE only requires one to compute the relative nitrate loss induced by nitrate photolysis, which makes the PIE independent of the initially deposited nitrate $\delta^{15}N$ and a good tracer of the isotopic effect of post-depositional processing. Here we propose a simplified formula of PIE for quick assessment of the photo-driven post-depositional processing effect on $\delta^{15}N(NO_3^-)$ at any sites of interest:

$$\text{PIE}(t_0) = \delta^{15}N(t_a) - \delta^{15}N(t_0) = -\int_{t_0}^{t_a} \varepsilon(t) J(t) \exp\left(-\frac{1}{z_e}\int_{t_0}^{t} A(t)dt\right) dt \quad (3)$$

Where $t_0$ represents the time of nitrate deposited on snow surface in a year (i.e., the starting time of photolysis), and $t_a$ is the time for snow nitrate to reach a depth below the snow photic zone (i.e., the archival layer) (3 times the e-folding depth). $t$ is the time variable between $t_0$ and $t_a$. $\varepsilon$ and $J$ represent the N isotope fractionation factor and nitrate

photolysis rate constant for snow nitrate at surface conditions, respectively. Both $\varepsilon$ and $J$ varies seasonally owing to the timely-varied actinic flux, while the decrease of nitrate photolysis rate constant with depth is constrained by the exponential term. $z_e$ and $A(t)$ represent the e-folding depth and snow accumulation rate, respectively. Here we don't consider the changes of $\varepsilon$ with depth as both the TRANSITS model calculation and

laboratory experimental results suggested $\varepsilon$ is not sensitive to the attenuation of radiation in snow (Berhanu et al., 2015). The diffusion smoothing in $\delta^{15}N(NO_3^-)$ is also not considered, as the observed multi-year snowpack $\delta^{15}N(NO_3^-)$ profiles don't show any distinct smoothing (Frey et al., 2009; Shi et al., 2015). The cage effect is also neglected in Eq(3) , which may not hold when the snow accumulation rate is relatively

low. Essentially Eq(3) is the same as Eq(2), because they both describe the total actinic flux received by a specific snow layer before archival, but Eq(3) provides a direct way to evaluate the induced isotope effects on $\delta^{15}N$. For illustrative purposes, we calculated PIE of snow nitrate deposited at different times of the year under typical Summit conditions and compare with the model output from Jiang et al. (2021). As shown in

Figure 3, the calculated PIE according to Equation (3) is consistent with the output from the TRANSITS model. The small departure is likely caused by the using of more simplified $J$ value with time in calculation, as the TRANSITS model also considers the changes in TCO. Using Equation (3), one should be able to quickly assess the effect of photolysis on the preserved snow $\delta^{15}N(NO_3^-)$ as long as the $J$ value and weekly or

seasonal accumulation rate are known.

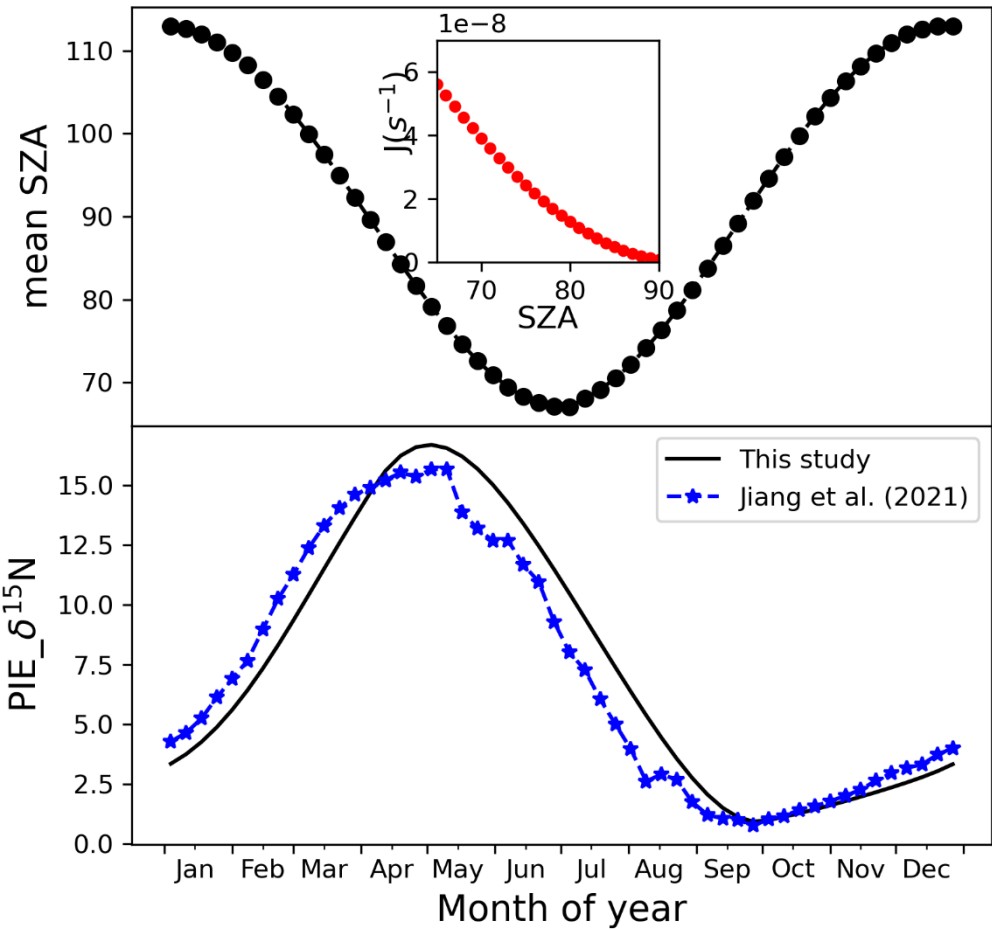

**Figure 3.** Upper panel: weekly average solar zenith angle (SZA) (shown in black circles) at Summit within a year and nitrate photolysis rate constant ($J$) as a function of SZA (shown in red circles in the inset). The summertime $J$ value is from Galbavy et al. (2007) and we scaled it with SZA to obtain $J$ values for other months for simplicity. Lower panel: calculated PIE using Equation (3) and that from the TRANSITs model in Jiang et al. (2021), weekly snow accumulation rates were from bamboo stake measurements by Dibb et al. (2004).

The $\delta^{15}N(NO_3^-)$ of atmospheric nitrate was depleted by $(9.8 \pm 5.1)$ ‰ relative to surface snow nitrate during spring (Fig 2b). In summer, the depletion was $(9.1 \pm 5.1)$ ‰, and decreased to $(4.0 \pm 4.3)$ ‰ in fall and became negligible in winter. Fibiger et al. (2015) and Jarvis et al. (2009) made short-term field observations at Summit in spring/early summer in different years by simultaneously collecting atmospheric gas-phase $HNO_3$ and surface snow for isotope analyses. The Fibiger et al. (2015) results

suggested that the surface snow nitrate was enriched in $\delta^{15}N(NO_3^-)$ by ~12-15 ‰ compared to atmospheric nitrate in May and June in average, close to result (~10 ‰) of the compiled data. However, despite using similar sampling techniques (mist chamber) for collection of gas-phase $HNO_3$ in March and May, Jarvis et al. (2009) found no significant $\delta^{15}N(NO_3^-)$ differences between gas-phase and surface snow nitrate. Note that the reported nitrate concentration from Jarvis et al. (2009) was high (>3 nmol m$^{-3}$ STP, which equals to 67.2 pptv) compared to other studies (ranges from 10 to 20 pptv) (SI). This may imply potential contamination during sampling of the gas-phase $HNO_3$ in Jarvis et al. (2009) collections.

Nevertheless, the enrichments of $\delta^{15}N(NO_3^-)$ in surface snow compared to atmospheric nitrate and its seasonal difference (larger in the summer half year) also imply the effect of the photo-driven post-depositional processing. Erbland et al. (2013) also observed enriched $\delta^{15}N(NO_3^-)$ in surface snow nitrate compared to atmospheric nitrate at Dome C, Antarctica. At Dome C, the seasonal pattern of the surface snow-atmosphere enrichments was similar to that at Summit, being the largest in Austral spring (~30 ‰) and the smallest in Austral winter (~10 ‰). In addition, the enrichment at Dome C was observed throughout the year, and even in winter there was still a ~10 ‰ enrichment. The elevated enrichment of $\delta^{15}N(NO_3^-)$ in surface snow nitrate compared to atmospheric nitrate in spring/summer observed both at Summit and Dome C suggest the role of photolysis as proposed by Erbland et al. (2013). Compared to surface snow, atmospheric nitrate is more influenced by snow-sourced nitrate which is severely depleted in $\delta^{15}N$ (–60 ‰ to –100 ‰, Jiang et al. (2021)). In addition, surface snow nitrate has experienced photolysis which tends to increase its $\delta^{15}N$ relative to the originally deposited nitrate. Winton et al. (2020) also suggested that at DML low snow accumulation rate and ample solar radiation tends to alter the original deposited nitrate signal through photolysis even for the skin layers (defined as the upper most 0.5 cm snow). At Summit, although the snow accumulation rate is high compared to the East Antarctic plateau, unless frequent snowfall occurs to wash out atmospheric nitrate to refresh the surface snow $\delta^{15}N(NO_3^-)$, dry deposition of atmospheric nitrate is unable to influence the budget of nitrate in surface snow (1-3 cm) and disturb its $\delta^{15}N(NO_3^-)$ even

in a period of a few weeks (Jiang et al. 2021). This is because (i) snow is a much larger reservoir of nitrate compared to the atmosphere and (ii) the nitrate dry deposition flux is very low (Bergin et al., 1995; Dibb et al., 1998).

### 4.1.2 The potential role of nitrogen isotope fractionation during deposition

Different from Summit, around +10 ‰ enrichment in surface snow $\delta^{15}N(NO_3^-)$ compared to atmospheric nitrate exists at Dome C during winter in the absence of sunlight. Erbland et al. (2013) attributed this winter enrichment to nitrogen isotope fractionation during nitrate deposition which increases $\delta^{15}N(NO_3^-)$ in the deposited nitrate compared to the atmospheric pool, and suggested this also contributes to the observed surface snow to atmospheric enrichment in spring/summer. However, the Summit data indicated no such enrichment in the winter, and this appears to be in conflict with the suggested deposition fractionation by Erbland et al. (2013). Although detailed physical mechanism leading to the deposition fractionation remains unknown, we speculated that the fractionation might be related to the form of deposition. Given the large difference in snow accumulation rate at Summit (250 kg m$^{-2}$ a$^{-1}$) and Dome C (25 kg m$^{-2}$ a$^{-1}$), their main nitrate deposition mechanism might be quite different. At Dome C, nitrate concentration in the skin layer is mainly controlled by adsorption and co-condensation of atmospheric nitrate (Bock et al., 2016; Frey et al., 2009; Chan et al., 2018). While at Summit, the dominant mechanism for nitrate incorporation into snow grain is the surface uptake during wet scavenging of atmospheric nitrate (Röthlisberger et al., 2002). Since wet deposition can efficiently scavenge atmospheric nitrate, a more complete removal of atmospheric nitrate at Summit compared to Dome C may occur, which would induce little to no isotope fractionation in $\delta^{15}N$ due to mass balance. However, for surface snow that continues to incorporate atmospheric nitrate via co–condensation or dry deposition (adsorption/desorption) after snowfall events, isotope fractionation could occur and leads to detectable enrichments in surface snow nitrate. The surface snow to atmospheric nitrate enrichments of $\delta^{15}N(NO_3^-)$ at Summit also appears to support the speculated role of fractionation during nitrate deposition. As shown in Figure 2b, the maximum enrichments occurred in spring/summer, which was

also the time with the lowest weekly average snow accumulation rate in a year (Burkhart et al., 2004) and dry deposition of atmospheric nitrate would account for a larger fraction of the total deposited nitrate, which leads to large isotope fractionation effect.

In summary, the systematic differences in $\delta^{15}N(NO_3^-)$ between atmospheric, surface snow and snowpack samples are consistent with the expected effects of the photo-driven post-depositional processing, while the occurrence and mechanism(s) of nitrogen isotope fractionation during deposition and its contribution to the surface snow-atmospheric $\delta^{15}N(NO_3^-)$ enrichment need to be further explored and confirmed.

## 4.2 The oxygen isotope systematics
### 4.2.1 The similarity of $\Delta^{17}O(NO_3^-)$ in atmospheric and snowpack nitrate

The atmospheric and snowpack $\Delta^{17}O(NO_3^-)$ display similar seasonality and their absolute values were similar (Figure 2c). The seasonal variations in $\Delta^{17}O(NO_3^-)$ is well understood as the seasonal shift of dominant $HNO_3$ formation pathways from summer ($NO_2 + OH \rightarrow HNO_3$ with low $\Delta^{17}O$) to winter ($N_2O_5$ hydrolysis with high $\Delta^{17}O$) (Alexander et al., 2020), so we don't discuss the cause of the seasonality in further detail. In the following discussion, we focus on the processes occurring at the air-snow interface and in snow and their effects on $\Delta^{17}O(NO_3^-)$.

Frey et al. (2009) proposed that nitrate in the uppermost layer of snow should reach equilibrium with atmospheric nitrate to maintain consistent isotope ratios. However, the large difference between atmospheric and surface snow $\delta^{15}N(NO_3^-)$ at Dome C Antarctica and Summit Greenland suggests no equilibrium. Conversely, an equilibrium in $\Delta^{17}O(NO_3^-)$ appears to exist. Erbland et al. (2013) made year-round observations of atmospheric nitrate and nitrate in the skin layer at Dome C, and found that $\Delta^{17}O(NO_3^-)$ in the skin layer was similar to atmospheric $\Delta^{17}O(NO_3^-)$ except in spring when $\Delta^{17}O(NO_3^-)$ was ~ 5 ‰ higher than the former. This was explained by a reservoir effect by Erbland et al. (2013), as the surface snow is always a much larger reservoir for nitrate relative to the atmosphere, and there might be a delay in skin layer nitrate variations compared to the changes in atmospheric nitrate.

Although annual surface snow $\Delta^{17}O(NO_3^-)$ data are not available at Summit, the two short-term observations by Fibiger et al. (2016) show that the atmospheric and surface snow $\Delta^{17}O(NO_3^-)$ are not significantly different. Note that the Fibiger et al (2016) surface snow data have much finer temporal resolution (4-12 hours) and show larger variability, but the averages fell well within the ranges of the atmospheric and snowpack data at longer time resolution. As proposed by Frey et al. (2009), one should expect a similar trend in atmospheric and surface snow $\Delta^{17}O(NO_3^-)$ (i.e., an equilibrium). This is because $\Delta^{17}O(NO_3^-)$ is a mass-independent fractionation signal and won't be affected by deposition process nor directly affected by snow nitrate photolysis, as these processes only induce mass-dependent fractionation. Once deposited, the only process that would influence snow $\Delta^{17}O(NO_3^-)$ is the cage effect (Frey et al., 2009; McCabe et al., 2005; Meusinger et al., 2014). The cage effect incorporates water with $\Delta^{17}O$ around 0 ‰ in the reformed nitrate and therefore lowers the overall $\Delta^{17}O$ of the nitrate compared to nitrate first deposited onto snow. But as observed and discussed by Erbland et al. (2013) and Jiang et al. (2021), the cage effect is likely an accumulated effect over long time periods and it won't significantly affect $\Delta^{17}O(NO_3^-)$ in the skin layer nor surface snow, which is also supported by the laboratory experimental results that decrease in the apparent quantum yield was observed with longer photolysis time (Meusinger et al., 2014). Therefore, the surface snow nitrate should possess similar or identical $\Delta^{17}O$ signal as atmospheric nitrate, as is observed.

From the surface snow to its final archival, $\Delta^{17}O(NO_3^-)$ would be further modified by the cage effect. The cage effect on snow $\Delta^{17}O(NO_3^-)$ is most evident at sites with low snow accumulation rate such as Dome C (Erbland et al., 2013; Frey et al., 2009), where nitrate stays in the photic zone for several years. In comparison, at Summit, the cage effect is negligible ($< 0.3$ ‰ upon archival, calculated by Jiang et al. (2021)) owing its fast archival (less than a half year) given the high snow accumulation rate, the archived snow nitrate should carry similar $\Delta^{17}O$ signal to its deposited value at the surface, which is in turn determined by atmospheric nitrate. Therefore, snowpack $\Delta^{17}O(NO_3^-)$ should be very similar to that of atmospheric nitrate, as is observed (Figure 2c). However, this doesn't mean that snow nitrate $\Delta^{17}O(NO_3^-)$ can be directly linked to

primary nitrate. Locally reformed nitrate under sunlight in the summer half year would possess low $\Delta^{17}O$ compared to primary nitrate deposited earlier in the season (Kunasek et al., 2009; Jiang et al., 2021) and contributes to the local atmospheric nitrate budget

(Jiang et al., 2021).

### 4.2.2 The atmospheric, surface snow and snowpack $\delta^{18}O(NO_3^-)$

Compared to $\delta^{15}N(NO_3^-)$ and $\Delta^{17}O(NO_3^-)$, $\delta^{18}O(NO_3^-)$ displays a much larger variability in terms of monthly averages as well as in the magnitude of the seasonal

variations, and are sometimes inconsistent even for the same type of samples (i.e., atmospheric vs. snow) measured by the same group. For example, Fibiger et al. (2016) reported average atmospheric $\delta^{18}O(NO_3^-)$ in May and June in one year of ~ 54 ‰ while in the other year it was ~ 91 ‰. The larger variability in $\delta^{18}O(NO_3^-)$ is somewhat expected, as it is influenced by $\delta^{18}O$ in precursor gases (NO, NO$_2$), radicals (O$_3$, OH,

BrO, HO$_2$, RO$_2$, etc), and atmospheric water, as well as fractionations during formation (Michalski et al., 2012). Additionally, snow nitrate photolysis also directly influences $\delta^{18}O$ with a fractionation factor calculated to be –34 ‰ by Frey et al. (2009), but does not affect $\Delta^{17}O$ owing to its mass-independent nature. Some of these processes act to enrich $\delta^{18}O$ (e.g., photolysis) while others act to deplete $\delta^{18}O$ (e.g., OH oxidation and/or

exchange with water).

Conventionally, variations in $\delta^{18}O(NO_3^-)$ are also used to track nitrate oxidation formation mechanisms, similar to $\Delta^{17}O$ (Michalski et al., 2012; and references therein). In general, under sunlight, nitrate formed from NO$_2$ + OH reaction possesses lower $\delta^{18}O$ than that formed from N$_2$O$_5$ hydrolysis under dark conditions. The latter involves

more oxygen atoms transferred from O$_3$ which possesses very high $\delta^{18}O$ (90-120 ‰, Johnston et al., 1997; Krankowsky et al., 1995). As a result, higher winter $\delta^{18}O(NO_3^-)$ and lower summer $\delta^{18}O(NO_3^-)$ should be expected, as observed for the atmospheric nitrate in this study and many others (Erbland et al., 2013; Savarino et al., 2007; Walters et al., 2019). This is also why we noted that the $\delta^{18}O(NO_3^-)$ data in Jarvis et al. (2009)

should be treated with caution as it indicated a summer maximum, which is difficult to understand given current knowledge. In the following discussion, we do not attempt to

describe this discrepancy in Jarvis et al. (2009) compared to other observations and our understanding of processes controlling nitrate $\delta^{18}$O.

Theoretically, after deposition of nitrate to the snow surface, both snow nitrate photolysis and the cage effect will all affect $\delta^{18}$O(NO$_3^-$) but in opposite directions. Similar to $\Delta^{17}$O(NO$_3^-$), there was also an (quasi) equilibrium in $\delta^{18}$O(NO$_3^-$) between atmospheric and skin layer snow nitrate observed at Dome C, Antarctica (Erbland et al., 2013). Atmospheric gas-phase and surface snow nitrate $\delta^{18}$O(NO$_3^-$) at Summit has been reported by Jarvis et al. (2009) and Fibiger et al. (2016) for spring and summer months. While the Jarvis et al. (2009) study suggested that the surface snow nitrate $\delta^{18}$O(NO$_3^-$) was on average 40 ‰ higher than atmospheric gas phase HNO$_3$, the Fibiger et al. (2016) study found that surface snow $\delta^{18}$O(NO$_3^-$) was lower than atmospheric nitrate in one year but higher in another. The atmospheric gas-phase nitrate $\delta^{18}$O(NO$_3^-$) reported by Jarvis et al. (2009) and Fibiger et al. (2016) were also lower than the atmospheric $\delta^{18}$O(NO$_3^-$) data reported by this study. The seasonal atmospheric and surface snow $\delta^{18}$O(NO$_3^-$) data at Summit also didn't indicate an equilibrium. Overall, the proposed equilibrium between atmospheric and surface snow nitrate $\delta^{18}$O(NO$_3^-$) is not supported by current observations.

Because of the lack of sufficient surface snow samples, and the relatively large variability among the limited observations by Jarvis et al. (2009) and Fibiger et al. (2015), we are unable to assess the potential oxygen isotope fractionation effects during nitrate deposition. But we note that this could also alter $\delta^{18}$O(NO$_3^-$) in analogy with $\delta^{15}$N(NO$_3^-$) and therefore this point needs to be further explored. After deposition, the post-depositional processing will impact the snow $\delta^{18}$O(NO$_3^-$) in a similar matter as it impacts $\delta^{15}$N. The typical photolysis isotope fractionation factor ($^{18}\varepsilon_p$) for $^{18}$O at Summit was calculated to be –32.8 ‰ using the ZPE shift method following Frey et al. (2009). Using the maximum loss fraction of 21% for spring snow from Jiang et al. (2021) and applying the Rayleigh equation, we calculated a maximum PIE of 7.7 ‰ for $\delta^{18}$O(NO$_3^-$). This means upon archival, snow $\delta^{18}$O(NO$_3^-$) would be enriched by up to 7.7 ‰ by considering only the photolysis fractionation. Conversely, the cage effect works to decrease snow $\delta^{18}$O(NO$_3^-$) by exchanging oxygen atoms with water. A

quantification of this effect (but an over-simplified one) is to consider the fraction of exchange of nitrate oxygen atom with water during the recombination chemistry, but one should keep in mind that the complex kinetic isotope fractionation during the recombination reactions could also affect $\delta^{18}O(NO_3^-)$ in snow. Here we used a simple mass balance method to assess the magnitude of changes in $\delta^{18}O(NO_3^-)$ through the apparent "exchange" caused by the cage effect:

$$\delta^{18}O(NO_3^-)_{finnal} = \frac{f_c * (1 - f_{rem}) * \delta^{18}O(H_2O) + f_{rem} * \delta^{18}O(NO_3^-)_{initial}}{f_c * (1 - f_{rem}) + f_{rem}} \quad (4)$$

where $f_{rem}$ and $f_c$ represent the remaining fraction of snow nitrate after photolysis and the fraction of exchange of nitrate oxygen atom with water via cage effect, respectively. Taking snow $\delta^{18}O(H_2O)$ to be –35 ‰ (Hastings et al., 2004) and snow nitrate $\delta^{18}O(NO_3^-)$ to be 80 ‰ (Geng et al., 2014) at Summit, and adapting the $f_{rem}$ and $f_c$ calculated by Jiang et al. (2021) to be 0.79 and 0.15, respectively, we calculated a maximum decrease in $\delta^{18}O(NO_3^-)$ of 4.4 ‰ upon archival caused by the cage effect. This is in contrast to $\Delta^{17}O$, as the very different $\delta^{18}O(H_2O)$ and $\delta^{18}O(NO_3^-)$ value makes the effect significant even for small amount of exchange. Note $f_c$ used here was a purely empirical parameter adapting from Erbland et al. (2015), by best fitting the decreasing trend in $\Delta^{17}O(NO_3^-)$ observed in Dome C snowpack. If we doubled $f_c$ (from 15 % to 30 %), an 8.4 ‰ decrease in $\delta^{18}O(NO_3^-)$ could be caused by the cage effect at Summit.

These simplified calculations suggest that there might be a difference in atmospheric $\delta^{18}O(NO_3^-)$ and snowpack $\delta^{18}O(NO_3^-)$ at Summit, but the magnitude and direction depend on the relative degrees of photolysis fractionation, the cage effect, and also other processes mentioned above (e.g., the kinetic isotope fractionation during secondary nitrate formation).

### 4.2.3 The relationship between $\Delta^{17}O(NO_3^-)$ and $\delta^{18}O(NO_3^-)$

Our atmospheric $\Delta^{17}O(NO_3^-)$ and $\delta^{18}O(NO_3^-)$ data exhibited some interesting features. As seen in Fig 1c, atmospheric $\Delta^{17}O(NO_3^-)$ and $\delta^{18}O(NO_3^-)$ appear to diverge during winter while in summer they were closely linked. The different $\Delta^{17}O/\delta^{18}O(NO_3^-)$ relationships in different seasons likely suggest different nitrate sources into the local

atmosphere, more specifically, the perturbation from snow-sourced nitrate in summer. In winter, owing to the low temperature and lack of sunlight, local nitrate production is suppressed and atmospheric nitrate is dominated by primary nitrate via long-range transport. In summer, the reformed atmospheric nitrate from NOx emitted by sunlit snow would possess oxygen isotope signals imprinted by local oxidation conditions that is different form primary nitrate. Although the $\Delta^{17}O/\delta^{18}O(NO_3^-)$ relationships for primary nitrate could also vary seasonally, the above explanation is further supported by the observed substantial NOx flux from snow in summer (Honrath et al., 2002) as well as the very negative atmospheric $\delta^{15}N(NO_3^-)$.

Fibiger et al. (2013) found a strong linear relationship between their measured $\Delta^{17}O(NO_3^-)/\delta^{18}O(NO_3^-)$ in surface snow samples at Summit. Based on this relationship they proposed a direct transfer of atmospheric oxygen isotope signals to surface snow at Summit. However, as discussed in Jiang et al. (2021), this relationship should not be viewed as an evidence of little to no post-depositional processing. Instead, examining the $\Delta^{17}O(NO_3^-)/\delta^{18}O(NO_3^-)$ relationships among atmospheric, surface snow and snowpack samples may provide some clues on whether or not the photo-driven post-depositional processing impacts the $\Delta^{17}O(NO_3^-)/\delta^{18}O(NO_3^-)$ ratio, since post-depositional processing influences $\Delta^{17}O(NO_3^-)$ and $\delta^{18}O(NO_3^-)$ differently. We note that different types of observations are different in their time resolutions. Our atmospheric measurement is typically 3 days per sample, while the surface snow samples (1-2 cm thickness) in Fibiger et al. (2013) represented weekly accumulation and snowpack sample resolution (5 cm per sample, Geng et al., 2014) is closer to monthly resolution. The linear regression relationship in surface snow shall not be changed by aggregation if post-depositional processing was negligible. Here we plotted our atmospheric and snowpack $\Delta^{17}O(NO_3^-)/\delta^{18}O(NO_3^-)$ data together with the four months (in year 2010 and 2011) of surface snow data from Fibiger et al (2013) in Figure 4.

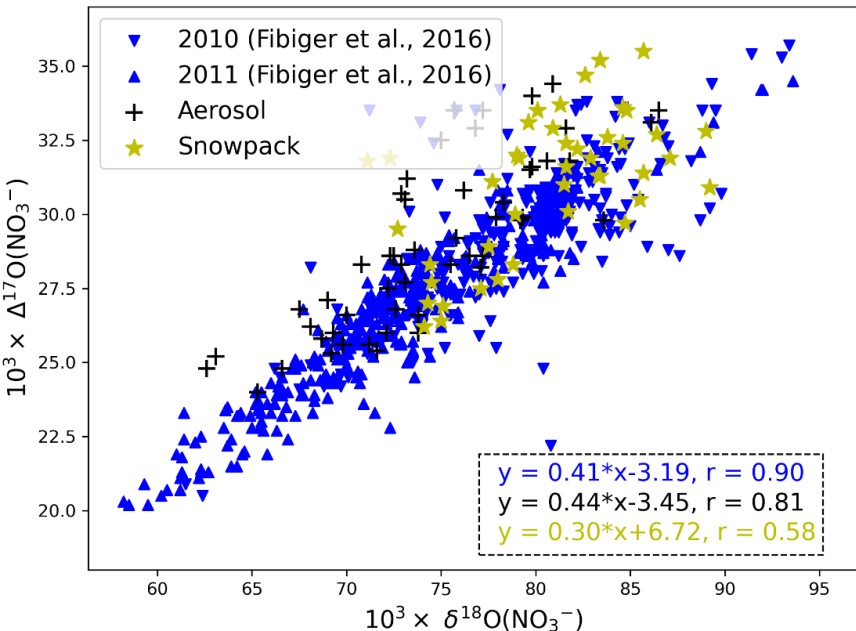

**Figure 4**. Relationship between $\Delta^{17}O(NO_3^-)/\delta^{18}O(NO_3^-)$ for all atmospheric (this study), surface snow (Fibiger et al., 2013) and snowpack data (Geng et al., 2014). Note the Fibiger et al. (2013) data was only for four months (May to June in 2010 and 2011), and the abnormal $\Delta^{17}O(NO_3^-)$ value less than 20 ‰ was abandoned.

As shown in Figure 4, the linear relationship between atmospheric $\Delta^{17}O/\delta^{18}O(NO_3^-)$ ($\Delta^{17}O(NO_3^-) = (0.44 \pm 0.04) \times \delta^{18}O(NO_3^-) - (3.45 \pm 3.28)$, r = 0.81) is very similar to the reported surface snow relationship ($\Delta^{17}O(NO_3^-) = (0.41 \pm 0.01) \times \delta^{18}O(NO_3^-) - (3.19 \pm 0.41)$, r = 0.90) despite their different time coverages. Such a relationship suggests that the linearity of $\Delta^{17}O/\delta^{18}O(NO_3^-)$ in surface snow may directly originate from atmospheric nitrate, consistent with the conclusion of Fibiger et al. (2013). The conservation of $\Delta^{17}O/\delta^{18}O(NO_3^-)$ relationship during deposition is somehow unexpected, as the current observed air-snow $\delta^{18}O(NO_3^-)$ difference is highly variable in both magnitude and sign (Jarvis et al., 2009; Fibiger et al., 2016). Further studies are required to understand why these observed atmospheric $\delta^{18}O(NO_3^-)$ are so different between different years. However, in the snowpack data, the linearity between $\Delta^{17}O$ and $\delta^{18}O(NO_3^-)$ ($\Delta^{17}O(NO_3^-) = (0.30 \pm 0.06) \times \delta^{18}O(NO_3^-) + (6.72 \pm 5.29)$, r = 0.58) was distinctly different from that of atmospheric or surface snow nitrate, suggesting that post-depositional processing likely has changed the originally deposited oxygen

isotope signals upon archival. We note that similar observations, i.e., better linearity of $\Delta^{17}O/\delta^{18}O(NO_3^-)$ in atmosphere and surface snow nitrate than that in the whole snowpack, were also observed at Dome C where the photolysis of snow nitrate has been unambiguously shown to be dominant (Erbland et al., 2013). This emphasizes again that, when evaluating the degree of post-depositional processing, one should consider samples covering the all depth of the photic zone, not only surface samples.

**4. Conclusion**

Nitrate isotopes in polar ice cores have been thought to reflect past changes in NOx emissions and atmospheric oxidation environments (Alexander et al., 2015; Geng et al., 2017; Hastings et al., 2009; Wolff, 1995). Although some important progress has been made (e.g., Geng et al., 2017), most interpretations of ice core nitrate records remain qualitative because the effects of post-depositional processing on nitrate and its isotopes have not been quantified. The latter requires a comprehensive understanding of the degree of post-depositional processing, as well as its influences on ice-core nitrate isotope preservation at different time scales. This is also true for ice-core drilling sites with high snow accumulation rates, where to what degree nitrate isotopes are changed upon archival is a subject of debate (Fibiger et al., 2013; Geng et al., 2015; Hastings et al., 2005; Jiang et al., 2021).

In this study, we reported the first year-round atmospheric nitrate isotopes measurements for Summit, Greenland. The atmospheric $\delta^{15}N(NO_3^-)$ displayed systematic differences from surface snow and snowpack $\delta^{15}N(NO_3^-)$ values at Summit compiled from the literature. In general, atmospheric, surface snow, and snowpack $\delta^{15}N(NO_3^-)$ diverged when there was sunlight but converged in the absence of sunlight. The gradual enrichments in $\delta^{15}N(NO_3^-)$ from atmospheric nitrate to surface snow nitrate, and finally to snowpack nitrate can only be explained by the effect of the photo-driven post-depositional processing, and the enrichment after deposition can also be quantitatively explained by the photo-induced effect (PIE). We proposed a simplified method for estimating PIE that can quickly assess the degree of $\delta^{15}N(NO_3^-)$ enrichment from the time of deposition to preservation in snow beneath the snow photic zone.

Unlike $\delta^{15}N(NO_3^-)$, snowpack and atmospheric $\Delta^{17}O(NO_3^-)$ displayed very similar seasonal patterns and absolute values, suggesting that it is well preserved, consistent with Jiang et al. (2021). We emphasize that atmospheric nitrate is not solely dependent on primary nitrate from long-range transport as it is also influenced by snow-sourced nitrate in the summer half year. The $\delta^{18}O(NO_3^-)$ data were more variable and showed some inconsistence among different observations. We analyzed the relationships between $\Delta^{17}O$ and $\delta^{18}O(NO_3^-)$ among different types of samples, and found that the slope and intercept of $\Delta^{17}O/\delta^{18}O(NO_3^-)$ correlations in snowpack is different from that of atmospheric and surface snow. This suggests that the degree of preservation for $\Delta^{17}O$ and $\delta^{18}O(NO_3^-)$ are likely different from each other at Summit, mainly due to the fact that photo-driven post-depositional processing causes mass-dependent fractionation of isotopes which directly affects $\delta^{18}O(NO_3^-)$ but not $\Delta^{17}O(NO_3^-)$. Overall, our analyses suggest that the photo-driven post-depositional processing impacts both $\delta^{15}N$ and $\delta^{18}O(NO_3^-)$ at Summit. As a result, the signals of primary nitrate $\delta^{15}N(NO_3^-)$ is unlikely preserved at this site, and $\Delta^{17}O$ and $\delta^{18}O(NO_3^-)$ of primary nitrate are also disturbed but to different degrees. These conclusions reinforce the importance of quantitative assessment of the post-depositional processing on snow nitrate isotopes even at sites with relative high snow accumulation rate (Jiang et al., 2021). Further numerical modeling is needed to correct the effects of post-depositional processing on $\delta^{15}N(NO_3^-)$, which is critical for the retrieval of information on past atmospheric NOx emissions using ice core $\delta^{15}N(NO_3^-)$ records (Hasting et al., 2009, 2015).

In the end, we note the limitations of the compiled data. These data were collected by different groups at different times, and with different sampling methods as well as different temporal resolutions. Although theoretically, the seasonality of the isotopes should be similar in different years or for samples collected and measured by different groups, and the heterogeneity of the samples was reduced by taking weighted average, there were some aspects and inconsistencies in the data that are difficult to interpret. Simultaneous collection of atmospheric, surface snow and snowpack samples with similar resolution for at least one complete year in the future should be conducted. This will provide a more consistent and solid dataset to improve or confirm the current

understanding of nitrate preservation and isotope variations at Summit, Greenland. This is not only important for nitrate isotope record interpretation at this site, but also for other sites with similar or higher snow accumulation rate such as the WAIS (West Antarctic Ice Sheet) Divide.

*Data availability.* The atmospheric nitrate isotope data and the compiled dataset will be provided upon direct request to the corresponding author.

*Author contributions.*

L.G conceived this study. JL.J. and J.E. collected and analyzed the atmospheric samples respectively. Z.J. compiled the dataset, analyzed the data, developed the formula used in calculation and wrote the manuscript with L.G. J. S. and B.A. provided suggestions for data interpretation. All authors gave feedback on the paper writing.

*Competing interests*. Some authors are members of the editorial board of *The Cryosphere*. The peer-review process was guided by an independent editor, and the authors have also no other competing interests to declare.

**Acknowledgements:** L.G. acknowledges financial support from the National Natural Science Foundation of China (Awards: 41822605, 41871051 and 41727901), the Fundamental Research Funds for Central Universities, the Strategic Priority Research Program of Chinese Academy of Sciences (XDB 41000000), and the National Key R&D Program of China (2019YFC1509100). This work was partially supported by the French national programme LEFE/INSU (IMAGO), the ANR grants ANR-15-IDEX-02 (project IDEX Université Grenoble Alpes) (J.S.). J.S. and JL.J. thank the French Polar institute (Institut Polaire Français IPEV, previously IFRTP) for field and funding support, and PNCA / CNRS-INSU for funding the SCIRA program (JL. J.). Gilles Aymoz is warmly acknowledged for setting up the experiments on site, together with winter over American people employed by PICO for sample maintenance during the sampling year. B. A. acknowledges support from NSF (award PLR 1542723).

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
