# Peer review of "Impacts of post-depositional processing on nitrate isotopes in the snow and the overlying atmosphere at Summit, Greenland"

_The Cryosphere, 2021_

## Referee Comment (RC2)

This paper presents some previously unpublished data on the isotopic values in nitrate in air samples from Summit, Greenland. It also compiles previous data on isotopes in nitrate in air, surface snow and "archived" snow at the same site. The paper then focusses on a discussion of the role of snow photolysis of nitrate in influencing the observed isotope values, their seasonality, and the differences between air and snow.

There are a few general comments to make about the paper. Firstly, the new data are potentially very useful and deserve to be published, even if I have a few questions about them. The authors have also done a nice job of compiling previous data, shown in Figure 2, which serves as an excellent starting point for a discussion.

The discussion is quite a tough read, even for someone who is quite well-versed in the issues but this probably reflects the difficulty of making clear statements in the light of sparse data, and a definite divergence in opinion between the major groups working on this topic. My main concern with the paper is that some statements are made, sounding quite definite, that are based on differences that are highly marginal. I realise it is disappointing when the conclusion of a study is "we're not sure if this is real", but in some cases this would have been a fairer conclusion. I think the overall structure of the paper and the way it tries to use the different datasets is good, so my comments mainly focus on particular statements that seem too definite or not to be well-supported by the data shown. For that reason, I just go through the paper in order, with both minor and major comments mixed in.

Title: the word "reality" seems a bit misplaced here. Of course one can read in the text an undercurrent that the message of the paper is directed at a competing group and that this paper is saying "there really is an effect". But I think for the neutral reader it would be less provocative and more accurate to write "evidence for the postdepositional effect".

Abstract: line 24: since you argue that you have collected aerosol and gas-phase nitrate, the word "aerosol" should be removed here.

Abstract: line 28: you should review the wording "no apparent seasonality". I will discuss this later, at line 248.

Abstract: please review what you have written in the light of edits elsewhere in the paper. I am particularly concerned that lines 37-43 are stronger than the data really allow (see later).

Line 51: Wolff 2008 is not in the reference list, whereas Wolff 1995 is, but is not cited in the paper.

Line 148, 156 and surrounds. Obviously a lot hangs on the quality of the atmospheric data. I have two issues I'd like clarified. The first concerns the use of GF filters. I agree they have often been assumed to collect gas phase nitrate as well as aerosol but the evidence is quite minimal for polar sites; the mechanism is assumed to be through attachment to sea salt loads on the filters (see eg Wagenbach et al, JGR, 103, 11007-11020, 1998 for a discussion of this, albeit related to cellulose filters). Given this I propose that Fig 2 (or a supplementary figure) should show a comparison of the concentrations of nitrate in this study compared to those found in previous studies (including Fibiger and Jarvis) that used mist chambers. This would allow a more informed discussion of whether this study is reporting a similar fraction of total nitrate to earlier studies.

In addition, I am a bit alarmed by the observation that nearly half the collected samples were discarded because they were too close to the blank (for nitrate concentration, I assume). This might imply that there remains a significant blank component in many of the filters that were not discarded and this could then affect the isotopic ratios measured if the blank is contributing

significantly. Please comment on this (I would assume you have some isotopic measurements on blank filters?).

Figure 1: I find it a bit strange that you choose not to plot the data chronologically but instead that you have Jan to Jul 2002 followed by Jul-Dec 2001. I would propose plotting the data chronologically (jul-Jul) in Fig 1, and from Jan-Jan in Fig 2.

Line 208-209. I don't request a change but I note that this is a bit circular. You use the similarity in seasonality to support your seasonal assignments in snowpack, and then later you use the same alignment as evidence that capdelta-17O in particular is unaffected by photochemistry.

Line 243. You attribute spikes to Arctic Haze events. Could it also be that it reflects more efficient scavenging during inputs of high sea salt (you would be able to support or deny this by looking at seasalt in the aerosol data)?

Line 248. You say there is no distinct seasonality in the atmospheric 15N. But I look at Fig 2b, where you also show the snowpack 15N from Geng, which you claim has a clear seasonality. While obviously the aerosol data have large variability within each month, I see just as strong a seasonal dip in the aerosol data as I see a seasonal peak in the snowpack data. In the end this isn't crucial because it's the differences between the air and the snowpack in different months (which is clear) that you focus on, but still please reword more cautiously.

Line 257: "(18O) displayed an almost identical seasonal pattern with Δ17O(NO3) as expected". I'm sure if I'd read your previous papers I would know why this was expected but it's not obvious, given that the former is a mass dependent fractionation and the latter is a mass independent one that could be quite separate. Please spell out why it's expected.

Figure 2. In part a, please clarify that the curve refers to the actinic dose (not "does") that would have been experienced by the snowpack samples.

Fig 2e: I assume these seasalt data refer to the snowpack data (but then which: Geng et al?). Please clarify this in the caption. Also, I'd be really surprised if the Na are in mg/L, surely they are ppb or ug/L?

Fig 2 caption. "The vertical lines represent the interval of seasons". I don't understand what this means. Are the error bars the differences between years for the same month/period, or are they the variability within a month or season. This is crucial to understanding what values are significantly different to others.

Line 290. This is the first case where I really feel you say things the data don't support. You refer to a progression of 15N from atmosphere to surface to snowpack. However when looking at Figure 2, it would be really stretching it to say that the surface snow data of Fibiger et al are significantly different from the snowpack data, taking into account the error bars shown. I agree there is a difference between the single Jarvis data point but as you later question this data I don't feel it's justified to make a wide-ranging and repeated statement about a progression on the basis of that. To me this is a place where you have to say that there is a clear difference between atmosphere and snow, but the data are insufficient to state with any certainty whether the surface snow and snowpack are different. The same issue is repeated in line 340.

(As an aside if the Jarvis atmospheric data in Fig 2b are right then the variability in the atmosphere between years is also too high to make a clear statement but I think it's Ok just to have noted the discrepancy).

Fig 3: Why do you only show J against SZAs (in the inset) almost entirely smaller than those experienced at Summit in the main figure?

Line 413, Fig 3 and surrounding discussion. I am really mystified by this lengthy section. Of course it's a nice advance that you can find a simple formula to represent the complex output of the model for PIE. However you don't then use it. Apparently PIE is the "difference between surface snow 15N(NO3–) and archived snow δ15N(NO3–)" . So why don't you plot the actual data in Fig 3 and see if it agrees with the model and its simplified (eq 3) representation. And of course the answer is that it doesn't. The observed PIE in May-July looks from Fig 2b to be about 3 permil, not the predicted values of about 10. (We can argue about the significance of the single value for spring, which you later suggest you don't believe). In any case my point is that there is no point having this section and figure unless you also show and discuss the data.

Line 423 – "J also varies with depth". This is wrong because J is the surface photolysis rate constant. The exponential term changes the actinic flux seen at depth.

Line 458-9. This is a strange statement in that it's not clear how a data point obtained by another group can be "explored and confirmed".

Line 680, 685 and what follows, plus 715 and following. A quite definite conclusion is based on the assertion that the slope of the snowpack data is significantly different from that of the atmosphere (not aerosol) and surface snow data. Your reported uncertainties on the slopes might indeed suggest that but sometimes it's better just to look at the data – would anyone really say that the yellow data are a significantly different population from the blue and black data?  In fact you suggest that what is happening is an enrichment of 18O in the snowpack samples. But again, just look at the data: it's quite obvious that if there is a difference it is that the snowpack data have a subpopulation that is enriched in 17O (I admit I am not sure how that leads to a lower slope and indeed by eye its very hard to see how the yellow points can have a slope of 0.3). Please reconsider this whole discussion; I think you are building a lot on very shaky differences.

Lines 724-736. Yes, I like this paragraph.

---

## Author Comment (AC1)

We appreciate the reviewers for their time and efforts to review this manuscript. Below we list detailed responses to their suggestions and comments. The suggestions and comments are in italics, followed by the response in normal font with changes highlighted in blue.

**Comments from Meredith Hastings**

*This work utilizes newly reported atmospheric nitrate measurements from Summit, Greenland to compare with others sets of data (from different years) from the atmosphere, surface snow and snowpack. The purpose of the study is to conclude that post-depositional processing can explain nitrate isotope systematics in the snow and air at Summit, Greenland. The overall subject matter is of relevance to Cryosphere and its readership. The conclusions drawn, and the title of the paper, do not fit with the evidence presented and the authors should consider the points below and better justify their conclusions.*

**Response:** First we would like to thank Dr. Hastings for her detailed comments on this manuscript. But as stated by Dr. Hastings in her comments and will be discussed here, clearly there is debate on the interpretation of the data and the embedded information in the data. Based on data of currently available and many of the reasons we have discussed in our previous papers (Jiang et al., 2021 TC), we can't agree with the interpretations provided by Dr. Hastings's group.

*While there are clearly differences of interpretation in the already published literature between my group and the authors here, new data definitely raises the potential for new interpretations. But it is important that the hypotheses, data, discussion and conclusions be consistent with what is "prove-able" within the constraints of the new data. **Using "reality" in the title is inappropriate** – at best it an overemphasis that this new work is somehow more important or more "realistic" than the large body of previously published work; at worst it is a direct insult to the work that has come before (including work by the authors of this manuscript). In fact, this quote from the conclusion negates the use of "reality "in the title: "In the end, we note the limitation of the compiled data. These data were collected by different groups at different time, and with different sampling methods as well as different temporal resolutions." (page 29, line 724)*

**Response:** As both reviewers pointed out, we notice that the usage of word 'reality' may have overstated the conclusion bases on current data of available. So in the revised manuscript, we have changed the title to "Impacts of post-depositional processing on nitrate isotopes in the snow and the overlying atmosphere at Summit, Greenland".

*A serious matter to consider first and foremost is what this new data represents in terms of the budget of nitrate in the air, snow and snowpack at Summit, Greenland. The new data is isotope measurements of aerosols collected using a high-volume air sampler with glass fiber filters. Quoting from the manuscript "Glass fiber filters have been shown to collect both aerosol and gas-phase nitrate with high efficiency (Erbland et al., 2013; Frey et al., 2009)." This has, in fact, not been shown directly.*

*And it is critical to consider since the data here is used as a comparison to other datasets and conclusions are drawn based upon the difference in isotope values amongst the different sample types, which were also collected with different methods. Therefore, the difference in the isotopes of different sample types has to be real and represent the environmental values and be sure not to represent any fractionation or changes associated with the collection technique. **Validation of the method for complete collection of HNO3(g) is critical to the interpretation here.** The manuscript refers to Erbland et al (2013) for evidence that complete collection of both nitrate phases takes place. Erbland et al. (2013) reported that concurrently collected samples of atmospheric HNO3 by denuder tubes coated in sodium bicarbonate matched well with the average values found by the hi-vol sampling and therefore were deemed representative of atmospheric HNO3 + aerosol nitrate at Dome C. The climate conditions, phase partitioning of atmospheric nitrate, accumulation rate, etc can all be very different at Summit than Dome C and therefore it is appropriate to review and consider what evidence there is that HNO3 and aerosol nitrate are quantitatively collected at Summit. Please note too that Frey et al. (2009) study at Dome C does not present any evidence for quantitative collection; Frey et al (2009) refers to Morin et al., 2007 which was an ocean-based cruise collection using the same methods but the conditions were such that the likelihood of complete HNO3 collection on filtered alkaline (sea salt) aerosols was high. Due to the difficulty of capturing both gas and particle phase nitrate under different conditions there is a body of literature that include methods discussions (see for instance Huang et al., Atm Research, 2004; Chiwa et al., Env Ass Monitoring, 2008; Ames and Malm, Atm Env, 2001; Lavery et al., Air & Waste Management Ass, 2009; and EPA CASTNET methods papers). The ideal method depends on the time of deployment, how polluted or pristine the expected air masses are, temperature, flow rate, and location (marine atmosphere, coastal, or inland). According to EPA recommendations, a filter pack with a particle filter (Teflon, quartz, GF/F, etc.) and a cellulose backup filter impregnated with NaCl or Na$_2$CO$_3$ will quantitatively collect particulate nitrate and nitric acid in a large variety of sampling conditions. As the addition of NaCl improves the efficiency of collecting nitric acid it is frequently assumed that both nitrate and nitric acid are collected with high efficiency in the marine boundary layer where there are high sea salt concentrations as Morin et al., 2009 points out (underlining added for emphasis): "The exact nature of the nitrate species trapped on the filters during sampling has been an issue for decades, owing to evaporative loss of ammonium nitrate species and sampling of nitric acid together with particulate nitrate [Schaap et al., 2004]. Prospero and Savoie [1989] have advocated that filters loaded with sea salt should quantitatively collect nitric acid together with particulate nitrate, which should then be the case for these samples collected in the MBL… Therefore, the analyzed nitrate is referred to as atmospheric nitrate, assumed to be the sum of gas phase nitric acid and particulate nitrate."*

**Response:** We thank the reviewer to bring this issue. Indeed, collection efficiency using filters is a long lasting issue. But as the reviewer pointed out, glass fiber filter

loaded with NaCl or used in MBL has been demonstrated to effectively collect $HNO_3$ and p-$NO_3^-$. In fact, the glass fiber filters used in this study were with high NaCl blank (>5 μmol on rinsed glass fiber filters), which should be able to quantitatively collect both $HNO_3$ and p-$NO_3^-$, as evidenced by the various studies quoted by the reviewer. In particular, Erbland et al. (2013) study found good agreement in their measured atmospheric nitrate concentration by using HVAS+glass fiber filter with the annular denuder method and suggested **that this was caused by the high NaCl blank in the glass filter.** At Dome C, gaseous nitric acid dominates total atmospheric nitrate (>90%), similar to the situation at Summit where gaseous nitric acid account for 94% of total atmospheric nitrate (Dibb et al., 1994). The high collection efficiency of glass fiber filter for gaseous $HNO_3$ as seen for Dome C comparison, should be applicable to Summit, unless there are other factors influencing the collection efficiency of $HNO_3$ and p- $NO_3^-$. But at least from the literature (already largely quoted by the reviewers), NaCl appears to be the most important factor.

In addition, we can compare our aerosol data with other studies to verify the collection efficiency. We have found another study at Summit that used the filter method to collect atmospheric nitrate and tested its collection efficiency (Silvente and Legrand, 1993). The Silvente and Legrand (1993) study simultaneously collected atmospheric nitrate with a denuder system and a Nylon filter system with flow rate of 30 L m$^{-3}$. Their results suggested that under conditions with atmosphere nitrate concentration less than 30 ng m$^{-3}$ the filter method produced similar results compared with the denuder method. Although we note the different filter type used, our measured average nitrate concentration (19.9 ± 19.1) ng m$^{-3}$ from the filter is very close to Silvente and Legrand (1993) measured nitrate concentration at Summit (20 ng m$^{-3}$).

Our measured atmospheric $\delta^{15}N(NO_3^-)$ (–19.1 ± 7 ‰, n=10) is also comparable to the results from Fibiger et al. (2016) (–16.0 ± 6 ‰ for 2010 and –17.7 ± 13 ‰ respectively) covering the same months but in different years (2001-2002 vs 2010/2011) with different methods (HVAS vs mist chamber). But we note, the Jarvis et al. (2009) study, despite using the same collection method with Fibiger et al. (2016) study (i.e., mist chamber), gave very different results in both $\delta^{15}N(NO_3^-)$ and concentrations from our study and the Fibiger et al. (2016) study. As discussed in the original manuscript, we don't know the reason since information on the sampling details were not available.

**We have also compared our measured atmospheric nitrate concentration with many other previous studies at Summit**. The results in different years are summarized below. As can be seen, the measurement results are generally in the same range except for Jarvis et al. (2009) which suggested a much higher concentration than other studies.

| Year | Month | type | Conc | Reference |
|------|-------|------|------|-----------|

|  |  |  | (ng m$^{-3}$) |  |
| --- | --- | --- | --- | --- |
| 1991 | 7-8 | denuder | 38 ± 53 | Silvente and Legrand, 1993 |
| 1993 | 6-7 | Mist chamber | 55 ± 37 | Dibb et al., 1994 |
| 1993 | 5-7 | Teflon Zefluor filter | 26 ± 2.9 | Bergin et al., 1995 |
| 1994 | 5-8 | Mist chamber | 32 ± 37 | Dibb et al., 1994 |
| 1995 | 4-7 | Mist chamber | 27 ± 32 | Dibb et al., 1998 |
| 2001-2002 | annual | glass fiber filters | 19.9 ± 19.1 | This work |
| 2006 | 5-7 | Mist chamber | **202** | Jarvis et al., 2009 |
| 2010 | 5-6 | Mist chamber | 32 ± 30 | Fibiger et al., 2016 |
| 2011 | 5-6 | Mist chamber | 42 ± 22 | Fibiger et al., 2016 |

In summary, we think the filter we used in this study have effectively collected $HNO_3$ and p-$NO_3^-$. In the revised manuscript, to validate the quantitatively collection of atmospheric nitrate, we have add the flowing statement in the main text:

"Glass fiber filters have been shown to be capable of collecting atmospheric nitrate with high efficiency even when gas phase nitrate dominates total atmospheric nitrate (Erbland et al., 2013). This is likely due to the high NaCl blank in the glass fiber filter, which is known to promote the collection efficiency of atmospheric nitrate (Morin et al., 2007; Erbland et al., 2013).".

*The vast majority of nitrate at Summit is wet-deposited via scavenging of HNO3(g). This is stated in the manuscript at the bottom of page 20 (though see specific comments below on references for this). This is important to validation of the method (above) since the air is filtered through a GF/F with no pre-treatment and must quantitatively collect all HNO3(g). Additionally, when it snows, both in cloud and below cloud processes (rainout + washout) will contribute to the nitrate that is deposited. However, the arguments and interpretation in this work appear to require that the atmospheric samples represent all (or most) of the nitrate that is deposited to the snow. For the interpretation here to stand, the input from the atmosphere must be constant in d15N and any differences in the snowpack from that in the air are only due to post-depositional processing. **Yet, the input of primary nitrate could change remarkably due to HNO3(g) scavenged from above the surface at Summit (i.e from cloud to ground) and the snow then would not represent only that which is near the surface (i.e. it could differ in d15N because it contains more than just what is at the surface).** This needs to be addressed in the context of the interpretation here.*
**Response:** The quantitative collection of nitrate was addressed in a previous response. Regarding the representativeness of the filter sampling at the surface, we think the fast mixing in the boundary layer will homogenize the boundary layer during one sampling period. The average eddy diffusion time for Summit summer conditions is 3.4 h for 25 m boundary layer height according to Cohen et al. (2006). Choosing a mean boundary layer height of 200 m (Cohen et al., 2006), this means the boundary layer would be mixed in 1.1 day, which is less than the time resolution of our atmosphere samples (3 to 4 days per sample). So the filter sampled nitrate is similar to snow which scavenges nitrate in air when it falls.

In addition, we should always expect that the local boundary layer reflects a combined effect from primary nitrate and locally reformed nitrate. This is especially evident for the measured $\Delta^{17}O(NO_3^-)$ value. If our collected atmospheric nitrate only reflect impact from snow source nitrate, then according to the Kunasek et al. (2009) and Jiang et al. (2021)'s calculation, the $\Delta^{17}O(NO_3^-)$ in middle summer should be close to 20 ‰ (19.7 ± 0.3 ‰ in Jiang et al., 2021 and 18.9 ‰ in Kunasek et al., 2009) considering the measured HOx and ozone levels at Summit. This is however much lower than the observed $\Delta^{17}O(NO_3^-)$ (>24 ‰).

*The atmospheric data (from a single year and almost half the samples are not used b/c of blank issues) suggests that within the surface atmosphere at Summit there is no seasonal variation in the d15N signal. It is entirely possible that this pool of atmospheric nitrate is controlled by local processes. **This does NOT negate that the snow can still represent a vast majority of primary nitrate deposition and that the***

*seasonal differences found in the snow represent much more than the local dynamics. The data we collected in Fibiger et al. 2013 and 2016 were concurrent collections of snow and atmospheric samples so that we could detail, at the same time, the dynamics in the snow and in the air. We went looking to detail that post-depositional release of NOx and reformation of nitrate locally COULD explain the isotopic composition of nitrate in the snow, especially D17O – but this hypothesis was not supported by what we found. Our results suggest that the nitrate in surface snow does not change isotopically in concert with what is happening in the surface air. It is hypothesized in Fibiger et al. that this can be explained by the fact that the surface air represents a small fraction of the nitrate that is deposited in the snow in real time. (Wet deposition at Summit is frequent, and in spring and summer, even when there is not fresh snow, fog deposition often occurs (see Bergin et al., 1993 already cited). This interpretation agreed with a snow-air model that was able to reproduce a suite of gas phase concentrations in the air at Summit, including NOx, HNO3(g) and halogens with as little as 6% of the nitrate being photolyzed (as discussed in Fibiger et al., 2016).* **It must be proven in this new work that the atmospheric data near the surface at Summit is what is most important in terms of the input of total nitrate to the snow such that the snowpack does NOT represent much more than what is happening in the air AT Summit.**

**Response:** It is too speculative to assume the atmospheric data is only controlled by local processes. Long-range transport nitrate depositing to snow has to first pass through the boundary layer, while as discussed and responded to earlier, the filter effectively collects $HNO_3$ and p-$NO_3^-$ in the air which is a combined signal of nitrate from long-range transport and local production. Again, the mixing of nitrate in the boundary layer is much faster than the sampling duration so what we sampled should represent atmospheric boundary layer nitrate instead of something only controlled by snow nitrate photo-recycling. This is especially clear given our atmospheric $\delta^{18}O/\Delta^{17}O(NO_3^-)$ relationships are almost identically to the surface snow relationships reported by Fibiger et al. (2013), despite that these samples were collected in different years. We also wanted to note that we don't attempt to refute that snowpack represents the majority of primary nitrate deposition, as our modeling work has already suggested that the reformed nitrate flux at most contributes 25% to local atmospheric nitrate burden at Summit.

Back to the Fibiger et al. (2016) study, they didn't observe an expected increase in surface snow $\Delta^{17}O(NO_3^-)$ when atmospheric BrO concentration was increased by a few ppt. This is perhaps their most important direct (from our reading) evidence to conclude or suggest that "*the nitrate in surface snow does not change isotopically in concert with what is happening in the surface air*". What happens in the air related to nitrate can only be reflected by collecting nitrate, but not by something deduced from BrO observations. We have discussed the reasons in detail in our previous paper (Jiang et al., 2021 TC) as well as in our responses to Dr. Hastings (who posted general comments during the open discussion). Here we just repeat our main points: First of all, increased BrO concentration (by a few ppt) in the air won't necessarily result in higher atmospheric $\Delta^{17}O(NO_3^-)$, nor snow $\Delta^{17}O(NO_3^-)$. This is because the production of BrO

will consume $O_3$. This is a tradeoff regarding their effects on $\Delta^{17}O$ of $NO_2$ which determines $\Delta^{17}O(NO_3^-)$. In addition, observations at Summit suggested BrO concentration always co-varied with OH/ $HO_2$/$RO_2$ (Liao et al., 2011) because they are both controlled by local photochemistry. If OH and $HO_2$/$RO_2$ concentration also increased (the authors didn't assess these radicals) at the same time, they would decrease $\Delta^{17}O(NO_3^-)$, offsetting the effects of increased BrO on atmospheric $\Delta^{17}O(NO_3^-)$. Second, whether or not the reformed nitrate in the air during the short duration of increased BrO (only a few hours) was able to influence local nitrate budget is questionable. The Jiang et al. (2021) model results suggested that locally formed nitrate can account for at most 25% of the deposited nitrate in summer and the rest is from transport. Additional nitrate due to BrO oxidation is only a small part of this locally formed nitrate, and whether its effect on atmospheric $\Delta^{17}O(NO_3^-)$ is detectable or not is also questionable. Third, the $\Delta^{17}O(NO_3^-)$ of surface snow (1-3 cm) was used by Fibiger et al. (2016) to compare with the effect of atmospheric BrO concentration increase. But the dry deposition flux of atmospheric nitrate is too low to significantly impact this "surface" snow nitrate concentration in short time scales (e.g., the duration of observed BrO). In summary, the Fibiger et al. (2016) cannot provide any evidence that nitrate in the air (or even the surface air) is disconnected with surface snow nitrate. In fact, the Fibiger et al. (2016) stated in their paper "***BrO chemistry does not have a significant influence on the formation of local $HNO_3$ at Summit***" (because they cannot explain the observed lower **atmospheric** $\delta^{18}O(HNO_3)$ with higher BrO concentration). This is exactly what we think their observations can demonstrate. If BrO chemistry is not important for atmospheric nitrate at Summit, it's not surprise to expect no correlation between BrO concentration and **atmospheric** $\Delta^{17}O(NO_3^-)$, let along in **surface snow**.

In addition, we also would like to repeat that '*6% of the nitrate being photolyzed*' is wrong. According to Fibiger et al. (2013) this value should be 2%. Fibiger et al. (2013) estimated the loss fraction by multiplying 0.1 % (loss fraction in 3 days in the upper 10 cm snow, they cited from Thomas et al. (2011)) with a factor of 21 (resident time in photic zone at Summit). The '0.1 %' they used in this calculation is an underestimate. From the supplemental file of Thomas et al. (2011) (Figure 9), the '0.1 %' value actually should be around 1% to 2%.

*I commend the authors on the collection and measurement of this important dataset and compiling many other datasets for comparison. The authors do a very good job of compiling the data comparison in the fairest way possible (e.g. mass-weighting sub-monthly data into monthly averages to compare with other data that is only reported as monthly averages), but we need to acknowledge that there is still a fair amount of comparing apples to oranges here. **There are no statistical comparisons presented in the manuscript and this should be addressed** (see specific comments below). While I appreciate the purpose in reducing "the spatial and temporal heterogeneities" for the comparison here, this heterogeneity is real and the standard deviations/variability should be included in any statistical comparisons. The manuscript argues that there are "systematic changes" in d15N between the air, surface snow and snowpack (all*

*from different years and different sample collection methods) that can be explained by post-depositional processing. But* **the differences shown between 2 out of 3 months in the wintertime are as large as observed in the spring and summer.** *Yes, the spring and summer months compiled data are consistently different, but the difference in December appears to about 16 per mil and the difference in February is 9 per mil (Figure 2b)! This needs to be explained within the framework presented here. This difference does not fit with the conclusion that post-depositional photolysis of nitrate can explain these types of differences nor does it fit with the idea that deposition of nitrate imparts a fractionation since this large difference does not exist in either October or January.* **Also left unexplained is the divergence of d18O and D17O behavior in the winter months atmospheric data** *(Figure 1c).*

**Response:** First, we would appreciate the reviewer's point on the data collection and reduction. It is difficult to compare data covering different years, however if one looks at the seasonality of isotopes in snow, these studies in general show very similar patterns, especially for $\delta^{15}N(NO_3^-)$ and $\Delta^{17}O(NO_3^-)$. As the seasonal patterns should be the same under a same background climate, we would expect that compiled seasonal patterns in snowpack should also represent that in the year of the aerosol sample collected. Regarding statistical assessments, we could have conducted student-T test. However, most of the data we found in the literature is already averaged, and some of them are monthly averages while others are seasonal averages (but only sampled in one or two months of a particular season). This means the sample size is difficult to determine and without this (freedom) t-test can't be conducted. Nevertheless, as seen in Figure 2b, in the seasons with sunlight, the atmospheric and snowpack difference is real, as averages of the atmospheric data plus one standard deviation is still lower than the snowpack averages plus one standard deviation.

Regarding the atmospheric and snowpack $\delta^{15}N(NO_3^-)$ difference in December, we realized it is somewhat out of range. Typically, the winter $\delta^{15}N(NO_3^-)$ valley shall be around -10 ‰ in Summit snowpack (Hastings et al., 2004; Jarvis et al., 2009). But our compiled data indicated the winter mean is (–1.2 ± 1.6) ‰. This is likely due to the dating uncertainties of the Geng et al. (2014) snowpack: the assigned Dec. snow in **2005** possesses $\delta^{15}N(NO_3^-)$ of ~ 2-3 ‰ which is abnormally high. It is possible that in dating the 2006 spring snow were treated as 2005 winter snow, or even in the year winter snow was mixed with spring snow by wind. In any case, we noted there is an anomaly in Geng et al. (2014) snowpit that the $\delta^{15}N(NO_3^-)$ value in the winter valley of 2005 is higher than other two valleys (~ 0 vs. -10‰). The February $\delta^{15}N(NO_3^-)$ difference in atmosphere and surface snow is not unexpected as the average PIE in February is 8.4 ‰ that could fully account for observed enrichment in snowpack $\delta^{15}N(NO_3^-)$.

Regarding oxygen isotopic data, it appears in Figure 1c that the relationship between $\delta^{18}O$ and $\Delta^{17}O$ in winter and summer is different. We thank the reviewer for this point, and think this could be an evidence that summer half-year atmospheric nitrate is influenced by local processes (local recycle) while the winter nitrate is not. We have included this evidence in the revised manuscript. We have added the following discussion about the $\delta^{18}O$ and $\Delta^{17}O$ relationships in Sec 4.2.3:

"Our atmospheric $\Delta^{17}O(NO_3^-)$ and $\delta^{18}O(NO_3^-)$ data exhibited some interesting features. As seen in Fig 1c, atmospheric $\Delta^{17}O(NO_3^-)$ and $\delta^{18}O(NO_3^-)$ appears to diverge during winter while in summer they were closely linked. The different $\Delta^{17}O/\delta^{18}O(NO_3^-)$ relationships in different seasons likely suggest different nitrate sources into local atmosphere, more specifically, the perturbation from snow-sourced nitrate in summer. In winter, owing to the low temperature and lack of sunlight, local nitrate production is suppressed and atmospheric nitrate is dominated by primary nitrate via long-range transport. In summer, the reformed atmospheric nitrate from NOx emitted by sunlit snow would possess oxygen isotope signals imprinted by local oxidation conditions that is different form primary nitrate. Although the $\Delta^{17}O/\delta^{18}O(NO_3^-)$ relationships for primary nitrate could also vary seasonally, the above explanation is further supported by the observed substantial NOx flux from snow in summer (Honrath et al., 2002) as well as the very negative atmospheric $\delta^{15}N(NO_3^-)$."

*Finally, it is a stretch to use the word "systematic" when several months (out of 12) do not follow this systematic response and the aerosol data represent 1 year of data collection (with only 55% of the data included in this study) while much of the snow and snowpack representing repeated sampling of the the snow/multiple snowpits/multiple studies.*

**Response:** Here we used the word "systematic" to represent the overall patterns of the seasonal atmospheric and snow $\delta^{15}N(NO_3^-)$, as they appear to follow the amounts of accumulated UV-B dose (but oppositely). And as shown in Figure 2b, only one month (i.e., the Dec. $\delta^{15}N(NO_3^-)$ values) but not several months appears to not follow the accumulated UV-B dose pattern. In particular, Feb. snow is expected to be influenced by sunlight because polar sun rises in March when nitrate deposited to snow in Feb. is still in the photic zone. Especially, if looking at the patterns of the snowpack $\delta^{15}N(NO_3^-)$ and the accumulated UV dose (Figure 2a and 2b), in general the higher the accumulated UV dose, the larger the snowpack $\delta^{15}N(NO_3^-)$. Note the highest $\delta^{15}N(NO_3^-)$ appears in spring when the accumulated UV dose is also the highest in a year.

*The assumption that the isotopic composition in the air must stay constant underlies discussion of the d18O data from the air and snow as well and does not agree with modeling or observational studies. Kunasek et al cannot explain higher than expected D17O values in snowpack summertime snow based on local photochemistry. Global modeling studies by Alexander et al do an excellent job of predicting the seasonal cycle in D17O throughout the year (based on long-range transport of nitrate and no post-depositional processing!) but shows a mismatch in spring (model overestimates) and summer (model underestimates). Fibiger et al. (2013, 2016) show interannual isotopic variability in the observations of surface snow and atmospheric samples (HNO3(g) only). This manuscript reports interannual differences in d18O of nitrate as "inconsistent." Fibiger et al. (2016) show and explain interannual differences based on differences in long-range transport,*

*changing the source regions from which primary nitrate is transported to Summit (which also impacts chemistry).*

**Response:** First we are confused by "*The assumption that the isotopic composition in the air must stay constant*". We didn't make this assumption in this manuscript so we didn't completely catch the question.

Regarding the modeling work, in Alexander et al. (2020), their modeled atmospheric $\Delta^{17}O(NO_3^-)$ didn't involve transport of nitrate among different atmosphere grids. Instead Alexander et al. (2020) modelled local HOx and ozone radical concentrations and calculated $\Delta^{17}O$ of nitrate produced **in situ**. The Alexander et al. (2020) study basically is not different from the method used in Jiang et al. (2021) and Kunasek et al. (2009) except for involving more elaborate chemistry schemes. All these three studies suggested local chemistry can not fully account for the observed atmospheric $\Delta^{17}O(NO_3^-)$ and this is why both Jiang et al. (2021) and Kunasek et al. (2009) invoked seasonal changes in $\Delta^{17}O$ of primary nitrate to explain the summer mismatch. Please also note, in TRANSITs model the wintertime atmospheric nitrate is completely controlled by primary nitrate as there is no locally reformed nitrate owing to the lack of sunlight, which is different with the other two models.

Regarding the $\delta^{18}O(NO_3^-)$ data inconsistences, there are way more questions to be answered. Fibiger et al. (2016) attributed the different **surface snow** $\delta^{18}O(NO_3^-)$ in different years to changes in nitrate source region but did not explain why the air-snow $\delta^{18}O(NO_3^-)$ relationship as well as the observed **atmospheric $\delta^{18}O(NO_3^-)$** were so different in two years. Also remains unexplained in Fibiger et al. (2016) was why the BrO concentration in 2010 was much higher than 2011 while the atmospheric $\delta^{18}O(NO_3^-)$ was much lower in 2011.

*Specific comments on manuscript:*

*It would be helpful if the abstract and introduction better reviewed prior work and results in Greenland. Much of what we are able to quantify about the impacts of post-depositional loss and recycling come from the body of work by Savarino in colleagues at Dome C. It's important to contextualize this and also contextualize the differences between the records in Greenland versus Antarctica. Currently in the introduction this all presented as "this is what happens to nitrate in snow period". (In my mind, the question is why don't we see more loss of nitrate in Greenland than we do?!? The exposure of the snow to sunlight, despite the accumulation rate, should still lead to more loss than is actually observed).*

**Response:** We think we have made a thoughtful introduction with relevant studies regarding the post-depositional processing loss of snow nitrate at Greenland as much as possible. **In addition, we don't agree that the observed snow nitrate loss is less than we expect**. Using the observed snow-flux of $NO_x$ at Summit, we have estimated the rate of snow nitrate photolysis (i.e., quantum yield of snow nitrate photolysis in Jiang et al. 2021), and then according to the actinic flux at Summit, the snow photochemical model calculated a maximum loss of 21% which is within the range (<7 % to 25 %) estimated by comparing surface snow nitrate concentration and

snowpack nitrate concentration in two observational studies (Burkhart et al., 2004; Dibb et al., 2007). What is more, under this level of estimated snow nitrate loss, the caused $\delta^{15}N(NO_3^-)$ change is also consistent with the observations, i.e., spring summer snow has more nitrate loss (due to the more accumulated UV dose received upon archival), and with higher $\delta^{15}N(NO_3^-)$ than fall and winter snow.

If we use the approach in Fibiger et al. (2013), but correct the value used to estimate the loss fraction from 0.1 % to 1% (which is 1 % nitrate loss in three days in 10 cm depth, details in response above), this will produce a loss fraction of ~21 % that is consistent with our model estimation (Jiang et al., 2021). We have discussed this in our previous response to the general comments by Dr. Hastings in Jiang et al. (2021).

In summary, we believe that based on current observations and modelling results, the observed snow nitrate loss upon archival at Summit can be well explained by the effects of snow nitrate photolysis.

*It would be useful in the introduction to clearly explain the differences between post-depositional loss versus recycling/processing of nitrate versus nitrate-snow sourced NOx-back to nitrate that is now different than originally deposited.*
**Response:** Thanks for this suggestion. We have added the following statement in the revised manuscript:
"…Photolysis of snow nitrate would emit NOx to the overlying atmosphere, which would subsequently reform nitrate under local oxidation conditions and redeposit to the surface. This recycling of snow nitrate not only changes the initially deposited nitrate (isotope) signal, but also leads to a redistribution of snowpack nitrate…".

*It's also important to be abundantly clear about how the words archived versus preserved are used in the text and it is likely worthwhile to define thsee in the introduction.*
**Response:** Thanks for this suggestion. We have added the following statement in our main text:
"…Thus, the final archived snow nitrate, defined as nitrate buried below the photic zone, would be largely impacted by post-depositional processing and this need to be fully understanding when interpreting ice core nitrate records …".

*Line 21: "...hinders interpretation of ice-core nitrate concentrations and isotope records." Given that Geng et al. alone have at least 3 different published papers where they interpret ice core records (let alone the many other papers that could also be named here), this sentence is not useful nor descriptive of the literature. This debate also hinders current understanding of atmospheric chemistry and deposition processes.*
**Response:** We meant "quantitative interpretation", so in the revised manuscript, we have added "quantitative" before "interpretation of ice-core".

*Line 24: this line says "atmospheric aerosol nitrate" which does not reflect that the atmospheric measurements are used as aerosol + gas phase nitrate*

**Response:** Thanks for this suggestion. We have deleted the word "aerosol" and added definition of atmospheric nitrate in the main text.

*Line 27: suggest rephrasing this line as with several negatives it currently reads as if the seasonality is the same between the snow and atmosphere; perhaps the following "…displayed no apparent seasonality, which is distinct from seasonal d15N-NO3- variations observed in snowpack."*

**Response:** Thanks for this suggestion. We have changed this sentence as follows: "…displayed minima in spring which is distinct from the observed spring $\delta^{15}N(NO_3^-)$ maxima in snowpack…".

*Please indicate what the standard deviations represent and how many samples (n) are included.*

**Response:** We have added in the revised version accordingly:

"**…**The atmospheric $\delta^{15}N(NO_3^-)$ remained negative throughout the year, ranging from –3.1 ‰ to –47.9 ‰ with a mean of (–14.8 ± 7.3) ‰ (n = 54), and displayed minima in spring which is distinct from the observed spring $\delta^{15}N(NO_3^-)$ maxima in snowpack. The spring average atmosphere $\delta^{15}N(NO_3^-)$ was (–17.9 ± 8.3) ‰ (n = 21), significantly depleted compared to snowpack spring average of (4.6 ± 2.1) ‰, with surface snow $\delta^{15}N(NO_3^-)$ of (–6.8 ± 0.5) ‰ that is in between…**"**

*Line 43: The degree of change in d18O being larger than that in D17O is a weird comparison to make. Since D17O represents the difference between d17O and d18O and both of those isotopes change with mass-dependent processes the D17O remains the same. Since the manuscript is to be read by an audience that includes non-isotope specialists, it would be useful to be clear about this.*

**Response:** We have changed the statements regarding the $\delta^{18}O/\Delta^{17}O(NO_3^-)$ relationship as following: "…This likely suggests the oxygen isotopes are also affected before preservation in the snow at Summit, but the degree of change for $\delta^{18}O(NO_3^-)$ should be larger than that of $\Delta^{17}O(NO_3^-)$ given that photolysis is a mass-dependent process that directly affects $\delta^{18}O(NO_3^-)$ in snow but not $\Delta^{17}O(NO_3^-)$.".

*Line 51: There is no citation for Alexander et al., 2019 in the manuscript. (And as an aside, the modeling work in Alexander 2009 and 2020 does not deal with the impacts of post-depositional processing).*

**Response:** Thanks for this comment. The right citation is Alexander et al., 2020 and has been corrected in the revised manuscript.

*Line 59: "…increases in d15N and decreases in d18O/D17O…" is only consistently true in Antarctica. See general comment above on the need to better discuss results and interpretation from Antarctica versus Greenland. This difference is compelling and would set the paper up better for how and why it's really important to try to resolve our understanding of post-depositional processing of nitrate and the interpretation of isotopes of nitrate.*

**Response:** This sentence describes the general patterns of the impacts of post-depositional processing on observed in Antarctica. But it doesn't mean it can be detected anywhere regardless of the degree of post-depositional processing. In this case, the patterns can't be detected in Summit, Greenland. But in Western Greenland, Curtis et al. (2018) study provides an example on how post-depositional processing changes snow nitrate $\delta^{15}N(NO_3^-)$. Curtis et al. (2018) found a significant coastal to inland gradient in snow $\delta^{15}N(NO_3^-)$ which they attributed to different degree of post-depositional processing, similar to the coastal to inland gradient observed in Antarctica.

*Line 69: "...has not been directly observed/evidenced in the field." Please see Shi et al., Isotope Fractionation of Nitrate During Volatilization in Snow: A Field Investigation in Antarctica, Geophysical Research Letters, 2019.*
**Response:** We are well aware of this paper, but it was a laboratory experiment conducted by collecting Dome A snow and measuring its changes in a room at Zhongshan station (a coastal site). It is not a direct observation nor can be viewed as direct evidence, and in any case they concluded "the evaporation of nitrate is minimal under typical polar area temperature range".

*Line 73: Geng et al. 2015 is not an appropriate reference here as it deals with deep ice core samples (the sentence refers to snowpack). It's important to include more context here – none of the other references are work done in Greenland.*
**Response:** We have deleted this citation and added the Curtis et al. (2018) study.

Line 84: *impurities in the snow also affect the chemistry and the form of nitrate in the snow – e.g. NaNO3 or CaNO3 and that can also impact post-depositional processing.*
**Response:** Thanks for this suggestion. We have added the following statement:
"…The higher dust concentration during glacial periods could also reduce the volatilization of snow nitrate (Röthlisberger et al., 2000). …".

*Lines 125-130: This is only true if the majority of the nitrate in snow comes from the surface atmosphere at Summit. See general comments above.*
*I think it could be better explained here that what is being referred to is the loss of nitrate from depth changes the isotopic composition in that snow layer. The snow sourced NOx from at depth, IF it re-forms nitrate and is re-deposited would change the surface snow values. It's also important to explain that the at depth layer should then reflect loss only based on the fractionation values presented in this manuscript (ie increase in d15N, increase in d18O). The surface value would be a mix of reformed nitrate and the original nitrate deposited.*
*The last sentence in this paragraph is really important so I think it is worth re-visiting the explanation here.*
**Response:** Thanks for the suggestion. Regarding the representativeness of atmospheric or surface atmosphere, again as we have discussed earlier, within the time scale of boundary layer mixing and sampling duration, the air sample collected at the surface

should well represent the boundary layer. When snow fall occurs it effectively scavenges atmospheric nitrate and brings it to the ground. To elucidate more on this point, we note that most of the snow precipitation at Summit is formed via low-level mixed-phase cloud, the height of which is typically several hundred meters above the surface (Guy et al., 2021; Pettersen et al., 2018). The precipitation as well as the low-level mixed-phase cloud could both increase turbulent mixing down into the surface mixed layer (Pettersen et al., 2018). Thus at Summit, a more realistic picture regarding the mixing state of the local atmospheric boundary layer should be that the local boundary layer nitrate represents a mixture of upper tropospheric nitrate as well as snow sourced nitrate in summer. And we agree that in summer, the summer snow nitrate is a mix of the re-formed nitrate and primary nitrate.

*Lines 145-150: Were the GFFs pre-cleaned or pre-combusted before use in the field? If so, how? If not, why not?*
**Response:** All glass fiber filters were pre-cleaned by an overnight soak and several rinses with ultra-pure water, then dried in a clean room and stored in clean plastic food storage bags till used. We have added the following statement:
"…All glass fiber filters were pre-cleaned by an overnight soak and several rinses with ultra-pure water, then dried in a clean room and stored in clean plastic food storage bags till used…".

*Line 156-167: Only 54 samples "out of 97 were determined to be valid". Please report the concentrations of the blank and how they were determined. Do they represent lab blanks or field blanks? What concentration is deemed not valid? What times of year are the dropped sample from? Does this skew the data in favor of particular months? For instance Figure 2 has no atmospheric data at all in September reported? And why is there no surface snow for Aug, Sep, Nov or Dec?*
**Response:** To make the description of the experiment procedure clearer, we have added the following statement:

"…Each sample covering 3-4 days were routinely collected over the year, with a total of 97 samples. We have also collected 9 blanks during the sampling period in different months, with the same sampling procedure but limited the sampling time to 1 minute. These samples were stored frozen until analysis.

Measurements of nitrate concentrations and isotopes were conducted in the laboratory at the Institute des Géosciences de l'Environnement, Grenoble, France in 2013. Nitrate collected on the glass fiber filters was first extracted by about 40 ml of 18 MΩ water via centrifugation using Millipore Centricon™ filter units. The samples were then measured for nitrate concentrations by colorimetry using the Saltzman method (Vicar et al., 2012). The average nitrate concentration in the filtrate for all atmospheric samples were (1363 ± 1603) ng g$^{-1}$, while that of the nine blank samples were (183 ± 44) ng g$^{-1}$. Among these samples, 54 out of 97 were determined to be valid by comparing the extracted nitrate concentration with blank, i.e., only samples with concentration exceeding 3 times the blank samples were judged as valid for further analyses. These samples were then individually concentrated on a 0.3 mL resin

bed with anionic exchange resin (Bio– Rad™ AG 1–X8, chloride form) and eluted with $5 \times 2$ mL of NaCl solution (1M). The isotopic compositions of each sample were determined by using the bacterial denitrifier method. Briefly, $NO_3^-$ in each sample was converted to $N_2O$ by denitrifying bacteria under anaerobic conditions. $N_2O$ was then thermally decomposed into $N_2$ and $O_2$ on a gold tube heated at 800 °C. The $N_2$ and $O_2$ were then separated by a gas chromatography column and injected into an isotope ratio mass spectrometer (Thermo Finnigan™ MAT 253) for isotope analyses of $^{15}N/^{14}N$, $^{17}O/^{16}O$ and $^{18}O/^{16}O$. To correct for the potential isotope fractionation during laboratory isotope analysis, international reference materials (IAEA-NO3, USGS-32, 34 and 35) were used for data calibration. We treated the reference materials the same as the filtrations from filter samples, e.g., making the reference material solution using 1M NaCl solution. The measured nitrate isotope ratio of each atmosphere sample was further corrected by deducting the contribution of the filter blanks. The overall measurement uncertainties were estimated to be 0.6 ‰ for $\delta^{18}O$, and 0.3 ‰ for both $\Delta^{17}O$ and $\delta^{15}N(NO_3^-)$. …", line 185-210

The lack of atmospheric isotope data mainly centered in September and October owing to the low nitrate concentrations in filters in these months (not exceed 3 times of filed blank). We don't find monthly surface snow data in these months reported in the literature.

*Were reference materials treated to the same procedures as the samples? i.e. were the reference materials put through the concentrating method as the samples were. Was a nitrate blank measured on the NaCl? or on the concentrating process on the whole? This is important to ensure no artificial isotopic change to the environmental sample. The denitrifier method induces fractionation of the d18O during the conversion to N2O and is corrected for by the samples being compared to reference materials treated in the same way. Any other pre-treatment of samples should also apply to reference materials to be sure it can be corrected for unless it is made clear that the pre-treatment causes no isotopic effects.*
**Response:** We have made careful calibration for the NaCl solution blank. Please see our response above.

*What are the measurement uncertainties reported here based upon?*
**Response:** The measurement uncertainty was based on the reduced standard deviations of the residuals from the linear regression between the measured reference materials and their expected values, as described in Erbland et al. (2013).

*Line 207: please be careful to distinguish between aerosol (only) nitrate, aerosol + gas phase nitrate, and gas phase only.*
**Response:** Thanks for this suggestion. We have changed the usage of "aerosol nitrate" into "atmospheric nitrate" throughout the text.

*Table 1: It would be useful to include the type of method used (i.e. filter type) and whether the collections represent aerosol, aerosol + gas, or gas only to help in*

*summarizing the different types of collections. In general when making comparisons of different datasets in the text, the difference in the isotope values is presented and THEN it is discussed that the samples actually represent different things (ie aerosol + gas versus gas only).*

**Response:** Thanks for this suggestion. We have added the corresponding items in the table 1.

*Line 217-220: II agree that it is odd to have a higher summertime d18O. But the validity of the data should be questioned based on evidence that somehow the higher values are biased in some way. For instance, an outlier test could be used. Is it not at all possible that a higher d18O could represent something anomalous that summer at Summit? could there have not been any different chemistry that could contribute to higher values (for instance, stratospheric intrusions)?*

**Response:** We agree that statistical methods would provide a more firm comparison here. Unfortunately, we didn't have the original surface snow data reported by Jarvis et al. (2009). Stratospheric intrusions are unlikely to be reasonable here as it's known that stratospheric transport is rather weak in Arctic region, such as suggested by Stohl. (2006).

*The line that "different groups" data were not averaged is also a bit odd. The Jarvis, Kunasek, and several Geng studies all measured samples at the University of Washington IsoLab. I suggest providing a different wording of justification here.*

**Response:** We deleted the usage of "different groups" in the sentence.

*Figure 1: It would be useful to also plot the calculated accumulation rate over the year since this is used in several calculations within the paper.*

**Response:** Thanks for this suggestion. We have added the monthly accumulation rate in Figure 2.

*Line 323: This is another place where a statistical test should be used to justify "out of range" and therefore why the data is not included.*

**Response:** We think that when the data is overlapped within each other's uncertainty range, using statistical method such as a T-test could help to discern whether the difference between two groups of data are statistically different. But when the range of data is not overlapped, it's unnecessary to rely on statistical method to judge whether it is out of range, as in the scenario described in the text.

*Line 342: "This systematic enrichment refutes the previous hypothesis that seasonal variation in snowpack....was driven by shift in the relative importance of NOx source…". See my argument against "systematic enrichment" in the general comments above. This alone does not negate the interpretation of nitrate in snow majorly representing long-range transported primary nitrate rather than local only. Additionally, the Jiang et al (2021) model could just as well explain the seasonality in the d15N snowpit profile when varying the d15N of primary nitrate (ie the "source"*

*value). In fact, this is noted on line 393-395 that the Jiang et al (2021) model shows "that the primary nitrate flux dominates the nitrate budget at Summit, even in mid-summer". This seems to directly contrast with statements in the paper where primary nitrate does not play an important role at Summit.*

**Response:** We have responded to the "systematic" question earlier, and repeated here: Here we used the word "systematic" to represent the overall patterns of the seasonal atmospheric and snow $\delta^{15}N(NO_3^-)$, as they appear to follow the amounts of accumulated UV-B dose (but oppositely). As in fact only the Dec. value in snowpack appeared to be out of the system. Feb. snow is also expected to be influenced by sunlight as polar sun rises in March when Feb. snow is still in the photic zone. Especially, if looking at the patterns of the snowpack $\delta^{15}N(NO_3^-)$ and the accumulated UV dose (Figure 2a and 2b), in general the higher the accumulated UV dose, the larger the snowpack $\delta^{15}N(NO_3^-)$. Note the highest $\delta^{15}N(NO_3^-)$ appears in spring when the accumulated UV dose is also the highest in a year.

And again, throughout this manuscript, as well as the Jiang et al. 2021 study, we never attempt to refute that the majority of snow nitrate deposited at Summit originates from primary nitrate deposition, as our modeling work suggested that the reformed nitrate flux at most contributes 25% to local atmospheric nitrate burden even in summer. It is the $\delta^{15}N(NO_3^-)$ that we focused on and its seasonality is determined or at least largely influenced by post-depositional processing and thus cannot be ignored in its interpretation, especially for ice core where the relative weight of post-depositional effect can vary a lot. In the Jiang et al. (2021) model, if we varied the $\delta^{15}N(NO_3^-)$ signature of primary nitrate as much as observed in snowpack, we have to turn off snow photochemistry, otherwise the predicted magnitude of seasonal $\delta^{15}N(NO_3^-)$ difference would be almost 2 times what is observed. Primary nitrate dominates the Summit nitrate budget, but the seasonal loss of nitrate is driven by the seasonal shift of actinic flux which drives snow nitrate photochemistry. As a result, even though the annual net loss of nitrate is small, the seasonal difference, especially the $\delta^{15}N(NO_3^-)$ could be large. The large N-isotope fractionation associated with snow nitrate photolysis can result in large $\delta^{15}N(NO_3^-)$ changes despite small amount of nitrate loss.

*Line 360: this is a bit confusing and connects to my comment about clarifying the different processes in the snow in the introduction. I think readers might be confused here that a depleted d15N source to the atmosphere requires that that NO3- be lost from the snow and the isotopic composition in the snow where the NO3- was lost will be changed (which has a different isotopic impact if that snow-sourced NOx is re-deposited locally).*

**Response:** Thank, to be clearer, we have added the following statement in the revised manuscript:

"…. is qualitatively consistent with the effects of snow nitrate photolysis which enriches snow $\delta^{15}N(NO_3^-)$ while providing a snow-source of depleted $\delta^{15}N(NO_3^-)$ to the atmosphere. In fact, the negative isotope fractionation factor associated with nitrate photolysis would favor the release of NOx with lighter [14]N, which would

rapidly reform nitrate in the overlying atmosphere with depleted $\delta^{15}N(NO_3^-)$, given the short lifetime of NOx at Summit (typical several hours in summer)"

*Line 367-370: this does not explain nor include mention of the fact that Dec and Feb look very different in the atmosphere versus the snow (this is related to my general comment above).*
**Response:** We have explained this earlier.

*Line 395-400: This does not address that differences in transport were connected to different isotopic compositions of nitrate at Summit in Fibiger et al (2016).*
**Response:** We admit that they are many issues related the atmospheric and snow nitrate chemistry as well as the isotopes, and that this data set cannot address them all. For example, the differences in transport were invoked to account for the **surface snow** $\delta^{18}O(NO_3^-)$ differences in two years. But at the time we noted that in Fibiger et al (2016) the $\delta^{15}N(NO_3^-)$ in atmosphere and surface snow is not different for two different years, in contrast to $\delta^{18}O(NO_3^-)$ which is very different. This is difficult to explain as to why difference in transport only induce changes in $\delta^{18}O$ but not impact $\delta^{15}N$, as it's $\delta^{15}N(NO_3^-)$ typically used to track the source region instead of $\delta^{18}O$. This is out of the scope of this manuscript, but can be explored once more data are available.

*Line 407-415: there needs to be clarifying language in here to distinguish between observations being shown here, prior model calculations, prior observations and calculation being done here.*
**Response:** Thanks for this suggestion. We have made the statements more specific when talking about different types of data (observation vs model):
"Compared to surface snow nitrate, snowpack nitrate was enriched by (12.8 ± 2.6) ‰ in spring in our compiled dataset, as seen in Fig 2b. This value should reflect the effect of post-depositional processing on snow nitrate throughout its preservation, i.e., time from being deposited at the surface to being archived below the photic zone. In Jiang et al. (2021), this effect was defined as PIE, i.e., the photo-induced isotope effect, and calculated as the difference between surface snow $\delta^{15}N(NO_3^-)$ and archived snow $\delta^{15}N(NO_3^-)$. The averaged PIE in spring calculated by the TRANSITS model is (14.3 ± 1.1) ‰, consistent with the observations."

*Line 408 says the difference between snowpack and the snow is 12.8 per mil in spring. Is this model or data calculated? the TRANSITS model predicts a difference value of 14.3 per mil, which is "consistent with the compiled data". In Figure 2b, the surface snow (blue symbols) appears to be ~ -8 per mil and the snowpack is +3 per mil in April (so the difference is ~11 per mil); there is no surface snow data shown for March; May (or between May and June?) the difference is only ~3 per mil. So how and when is the Jiang et al calculation and TRANSITS model calculation consistent with the compiled data?*

**Response:** The (12.8 ± 2.6) ‰ **seasonal difference** is based on the observed average surface snow $\delta^{15}N(NO_3^-)$ from Jarvis et al. (2009) (-6.8 ± 0.5) ‰, n=84) and snowpack seasonal average $\delta^{15}N(NO_3^-)$ from Geng et al. (2014) (6 ± 2.6 ‰). Note there are two years of data in Jarvis et al. (2009) but only the seasonal average value and standard error is reported.

*Then on line 444 snow and atmosphere are compared and it is reported that the difference in spring should be 9.8 per mil – this is true for Apr, but in Mar and May-June the difference is larger and again (close to 20 per mil in Mar!). If I am mis-reading this then the text needs to be clarified and it would be helpful to refer directly to figures when speaking of the data here.*
**Response:** Note here we compare atmospheric nitrate with surface snow nitrate. Owing to the seasonal resolution of the Jarvis et al. (2009) dataset, **we only compared the seasonal mean here instead of any specified month**. In addition, the surface snow $\delta^{15}N(NO_3^-)$ data is significantly higher in Fibiger et al. (2016) than in Jarvis et al. (2009) which call for more data to resolve this issue.

*Line 446 says the difference between atmosphere and snow is negligible in winter, but this is only true of 1 of the 3 months in winter in Figure 2b.*
**Response:** Here we compare atmospheric nitrate with surface snow nitrate. Similar to the response above, only **seasonal** average values were compared here. The winter average atmospheric $\delta^{15}N(NO_3^-)$ is (-12.9 ± 4.6) ‰, while that of surface snow is (-11.7 ± 2.3) ‰.

*It would be useful to include the Jiang et al (2021) model output as part of the figures since values produced in that work are referred to at least 4 times in the manuscript.*
**Response:** The modeled $\delta^{15}N(NO_3^-)$ change (PIE) from deposition to archival have been compared in Jiang et al. (2021) and we do not repeat here. Again, the strength of model is to predict changes in isotopes caused by post-depositional processing related to a starting point, and that was what we have been focused on when using the model.

*Line 408-415: I think preservation of the value below the photic zone is important and should be clarified in the introduction as to this expectation. However, I'll also note that Erbland's study with the TRANSITS model predicted values below the photic zone, which agreed with a few individual samples from depths below the photic zone compared to results from within the photic zone in several areas of the East Antarctic Ice Sheet. In a follow up study, Shi et al. (ACP, 2015) presented complete profiles along the EAIS between the surface and below the photic zone and found that the case for an exponential increase in the isotopes was highly sensitive to the depth over which it is assumed the photic zone is relevant. This does need a response but it would be good for the authors to re-review that work to be sure to be consistent with the peer-reviewed literature.*
**Response:** Thanks for this suggestion, we have already added the definition of archival nitrate in the introduction as mentioned above. However, as pointed out by

reviewer 1, Shi et al. (2015) may have misunderstood the extrapolation method in Erbland et al. (2013). That Erbland et al. (2013) assumed an exponential increase in $\delta^{15}N(NO_3^-)$ in the photic zone is only true when assuming local meteorological and chemical **conditions remain unchanged**. Below the photic zone, the archived nitrate $\delta^{15}N$ is a combination of photolysis-induced fractionation plus surface condition changes, so it's natural to see the asymptotic values vary with depth, **and this is not because the assumption of the photic zone depth**, but instead due to changes in other factors such as snow accumulation rate/impurities concentrations, TCO, and even primary nitrate inputs.

*Line 430: Equation 3 – a simplified form of the PIE is really valuable. It's agreement with TRANSITS seems a bit circular since the equation is based upon the TRANSITS model. It should be better explained here what this simplified equation does NOT include relative to TRANSITS so that the simplified version is applied by other groups in the future under situations that are appropriate. For instance, does the A(t) in Eq 3 considers an e-folding depth that is impacted by impurities in the snow?*
**Response:** Eq(3) is the mathematical representation of the integral effect of photolysis on $\delta^{15}N(NO_3^-)$ in TRANSIST model. The difference is that the *J* value used here is in a simplified form, i.e., *J* decreases exponentially with depth. The other difference includes: 1) the changes in nitrogen isotope fractionation factor with depth is also not considered as both calculation and experiment suggested it's insensitive to the attenuation with depth (Berhanu et al., 2015);2) the diffusion smoothing effect is not considered here either (it should not be important as the observed $\delta^{15}N(NO_3^-)$ profile in Dome A and Dome C doesn't show distinct smoothing in $\delta^{15}N(NO_3^-)$);3) the cage effect is not considered. We have changed the word of "agree with" into "is consistent with". A(t) in Eq 3 represents the accumulation rate at a given time t, only $z_e$ depends on the impurities in snow. We have added the following statement as follow:

"…Here we don't consider the changes of $\varepsilon$ with depth as both the TRANSITS model calculation and laboratory experimental results suggested $\varepsilon$ is not sensitive to the attenuation of radiation in snow (Berhanu et al., 2015). The diffusion smoothing in $\delta^{15}N(NO_3^-)$ is also not considered, as the observed multi-year snowpack $\delta^{15}N(NO_3^-)$ profiles don't show any distinct smoothing (Frey et al., 2009; Shi et al., 2015). the cage effect is also neglected in Eq(3) , which may not hold when the snow accumulation is relatively low. Essentially Eq(3) is the same as Eq(2), because they both describe the total actinic flux received by a specific snow layer before archival, but Eq(3) provides a direct way to evaluate the induced isotope effects on $\delta^{15}N$…", line 489-495

*Line 450: Is this based on comparing means? Medians? The medians in Fig 3 of Fibiger et al show differences of only 9-12 per mil not 12-15 per mil. Please clarify.*
**Response:** Here we used the average value (thanks for the origin data we downloaded it from the Arctic Data Center). We have added the phrase "on average" when describing it.

*Also, why is the atmospheric nitrate oxygen isotopic data from Fibiger et al. NOT included in Fig 2?*

**Response:** The Fibiger et al. (2016) $\delta^{15}$N data has been plotted in Fig2b and $\delta^{18}$O data was not included because of the rather large discrepancy between the two years and they are out of range of other studies including the values for snow. In the revised manuscript, we have added a statement in the Figure caption, stating the range of the Fibiger et al. (2016) $\delta^{18}$O data and explaining it is out of the range of other data in the same figure as follows:

"The atmospheric $\delta^{18}$O(HNO$_3$) data in Fibiger et al. (2016) is out of range ((54.2 ± 8.5) ‰ in 2010, (90.5 ± 12.5) ‰ in 2010) and thus is not shown here."

*Line 471-483: "Compared to surface snow, atmospheric nitrate is more influenced by snow-sourced nitrate…" Yes. This is because the snow represent more than the surface atmosphere at Summit. And while this is stated as "snow is a much larger reservoir of nitrate compared to the atmosphere" on Line 481, the context here is not clear (see general comments above on this).*

**Response:** We think we have different meaning regarding this sentence. We meant to explain that owing to fact that atmospheric nitrate is such a small nitrate reservoir that it could be easily impacted by the snow sourced nitrate, while for the surface snow, once deposited, it would not be rapidly altered as the dry deposition of snow sourced nitrate was too low to significantly impact it.

*It's not clear why the Erbland reference is relevant here since it does not apply to Greenland (and for instance in the discussion above while dry deposition at Summit is infrequent fresh snow and fog deposition are frequent). You should also include these references in discussing wet versus dry deposition and the budget of nitrate in the atmosphere at Summit:*
*Dibb, J. E., R.W. Talbot, and M. H. Bergin (1994) Soluble acidic species in air and snow at Summit, Greenland; GRL; 1627-1630.*
*Dibb, J. E., R. W. Talbot, J. W. Munger, D. J. Jacob, and S.-M. Fan (1998) Air-snow exchange of HNO3 and NOy at Summit, Greenland; JGR; 3475-3486.*

**Response:** Thanks for this suggestion. We have deleted Erbland et al. (2013) and cited the more related references.

*Lines 494-510: this discussion would be useful earlier in the manuscript to layout an expectation for how the results will be interpreted.*

**Response:** Thanks for this suggestion. We add a brief description in the opening paragraph in this section:

"…Previous studies suggest there were several processes occurring at the air–snow interface related to nitrate deposition and preservation that could lead to nitrogen fractionation, including (i) fractionations during snow nitrate photolysis and physical release (Berhanu et al., 2014; Erbland et al., 2013; Frey et al., 2009; Jiang et al., 2021;

Shi et al., 2019), and (ii) the proposed fractionation during nitrate deposition related to the different deposition mechanisms (Erbland et al., 2013)…", line 404-405

*Line 512: Are there updated stake measurements at Summit that could be checked for this? The Burkhart et al. study is a bit dated, and accumulation rate at Summit could. be with climate change in the last decade? Also, how many years is the lowest weekly average accumulation rate observed since the Burkhart et al. study is not at all representative of accumulation measured during the time periods of the compiled data or the atmospheric data presented here.*

**Response:** The Burkhart et al. (2004) stake-measured accumulation rate data covered from 1997 to 2002. This study used data from 2004 to 2010 to calculate the average weekly accumulation rate. We have compared our calculated monthly accumulation rate with Burkhart et al. (2004) and Castellani et al. (2015) and found good agreement during three periods (1997 to 2002, 2004 to 2010, 2010 to 2013). The common feature is with a minimum accumulation rate in May-June and a maximum in August-September.

*Line 513: "and presumably more nitrate dry deposition occurred…" Why is this presumed? And is this consistent with one page ago where the dry deposition flux was considered to always be "very low" ?*

**Response:** Here we meant when wet precipitation is low, dry nitrate deposition would be relatively more important. It is a relative term, and the overall dry flux could be still low.

*Line 553: This should read as "The cage effect incorporates water D17O (~0 per mil) in the formation of nitrate and therefore lowers the overall D17O of the nitrate compared to nitrate formed in the atmosphere." (or something like that). The way this is written now is not actually correct and will definitely confuse readers.*

**Response:** Thanks for this suggestion. We have changed the statement as follows: "…The cage effect incorporates water with $\Delta^{17}O$ around 0 ‰ in the reformed nitrate and therefore lowers the overall $\Delta^{17}O$ of the nitrate compared to nitrate first deposited onto snow…".

*Line 563: "the cage effect is negligible". It needs to be said here that the Jiang et al (2021) model output does not match well AT ALL with the D17O snow profile. So I think it is unfair to bring conclusions from that study regarding D17O into here as if they are "proven" by a model that does not actually explain/match the observations.*

**Response:** We just wanted to explain a bit more that the Jiang et al. (2021) model study is explicitly focused on to what degree the magnitude of the observed seasonality can be explained by post-depositional processing, and the TRANSITS model we used here was not constructed to predict the isotopes of snow nitrate ($\delta^{15}N$ and $\Delta^{17}O$), but to assess their **changes** due to post-depositional processing. Using the model, the cage effects on $\Delta^{17}O$ is negligible, and the overall effects from post-depositional processing on snowpack $\Delta^{17}O$ is also small. We don't see why the model

has to reproduce the observed $\Delta^{17}O$ to conclude anything on the effects of "cage effect" and post-depositional processing. That the model only considering post-depositional processing can't explain the observed $\Delta^{17}O$ is exactly demonstrating that the effects of post-depositional processing (including the cage effect which is a part of it) is small.

*Line 569: "Locally reformed nitrate under sunlight in the summer half year would possess low D17O compared to primary nitrate deposited earlier in the season…" And yet, the Kunasek et al study based on local photochemistry at Summit cannot explain the HIGHER THAN EXPECTED summertime D17O values. And the global GEOS-Chem modeling studies (Alexander et al, 2009, 2020) fit the seasonality of D17O at Summit very well, but UNDERESTIMATE summertime values and OVERESTIMATE spring values.*

**Response:** We don't see any issue here. "*the Kunasek et al study based on local photochemistry at Summit cannot explain the HIGHER THAN EXPECTED summertime D17O values.*" - This is actually the same we are saying here: the reformed nitrate is with relatively low $\Delta^{17}O(NO_3^-)$ and thus primary nitrate with relatively high $\Delta^{17}O(NO_3^-)$ is necessary to be involved to explain the observations.

*Line 575-583: Data from different years should not necessarily be expected to be consistent. The wording should be changed here or the authors should justify why this should be expected. If photolysis alone can always explain the seasonality of nitrate isotopes then it SHOULD be the same every year. The observed (real) differences in the Fibiger et al dataset are explained in that work in terms of variability in transport and chemistry (ie source regions). There is no reason to dismiss some of the d18O data in Fibiger et al. I'll also point out (for my own gratification) that our lab is the only lab that independently reports d18O data from the N2O method and D17O from the N2/O2 method b/c the N2/O2 method is known to cause mass dependent inconsistencies in d18O and d17O (which do not affect the D17O result).*

**Response:** We don't fully understand the question. Here we are just stating that the $\delta^{18}O(NO_3^-)$ data are different, or more variable compared to $\Delta^{17}O(NO_3^-)$ and/or $\delta^{15}N(NO_3^-)$. This is from observations, and in line 580 and 581 of the original manuscript, we clearly stated that "The larger variability in $\delta^{18}O(NO_3^-)$ is somewhat expected,…." We are not saying they should be consistent. Regarding the Fibiger et al. (2016) study, again if transport could explain the large difference in $\delta^{18}O(NO_3^-)$ in the two field seasons, but then why is $\delta^{15}N(NO_3^-)$ the same for the two seasons? These are questions need to be explored but not in this manuscript.

*Line 616: again, why should the data from different years be consistent if transport, accumulation, deposition all change between seasons and interannually?*
**Response:** Here we wanted to describe that due to the lack of sufficient surface snow data, and the large difference of the $\delta^{18}O (NO_3^-)$ data reported by the two studies, we cannot deduce or explore whether or not oxygen isotope fractionation occurs during nitrate deposition.

*Line 633: What happens to the d15N during the cage effect reformation of nitrate? It is only discussed for d18O (and D17O).*

**Response:** The detailed secondary chemistry of nitrate photolysis is very complex and its induced isotope fractionation effect for $\delta^{15}N$, to best of our knowledge, still remains unexplored. However, current filed and laboratory photolysis experiments results indicate that photolytic process should dominate the total fractionation effect (Berhanu et al., 2014; Meusinger et al., 2014). The calculated fractionation factor using the absorption cross section for $^{14}NO_3^-$ and $^{15}NO_3^-$ provided theoretical results under different spectrum of indecent light consistent with the experiment results , suggesting that photolytic process shall be the primary factor inducing the $\delta^{15}N$ fractionation during nitrate photolysis.

*Line 653: I would argue that the framing in Fibiger et al (2013) is that the relationship between d18O and D17O is not evidence alone that post-depositional processing does not occur – we attempt to explain how that relationship should change is an important amount of processing were to occur and that does not fit with the observations, therefore we conclude that it is not very important. This was then followed up with the concurrent snow-air sampling in Fibiger et al (2016).*

**Response**: We agree that Fibiger et al (2013) indeed considered multiple factors impacting snow nitrate $\Delta^{17}O$ and $\delta^{18}O(NO_3^-)$ and they concluded that the found relationship between $\Delta^{17}O/\delta^{18}O(NO_3^-)$ can not explained by post-depositional processing. But this can only lead to the conclusion that the **surface snow** nitrate didn't undergo sufficient amount of photolysis, as only surface snow samples were involved. The same for the Fibiger et al (2016) study which was also only based on atmospheric and surface snow samples. Consider the high snow accumulation rate (~65 cm per year) as well as the depth of the photic zone (30-40 cm), the surface snow samples apparently cannot represent the whole snowpack. What is more, none of these two studies cover a complete year, and thus the effect of the seasonally different actinic flux cannot be assessed.

*Figure 4 discussion: The fact that relationship between d18O and D17O is so similar in the atmosphere and snow and the snowpack is different could just as easily be explained by deposition of more nitrate than what is apparent at the surface at Summit.   And if it cannot be explained by this, this hypothesis needs to be tested and dismissed.*

**Response:** We don't agree with this hypothesis since it sounds like the snowpack nitrate was directly buried without first deposited to the surface. We note that the correlation between surface snow $\Delta^{17}O/\delta^{18}O(NO_3^-)$ was similar in two different years, even if they were derived from different source region and different chemistry as suggested by Fibiger et al. (2016).

*Line 682: Earlier in the manuscript it states that the isotopes are preserved upon archival. Again, please take care to qualify the language in the introduction and then use it consistently in the manuscript.*

**Response:** Thanks for this suggestion. We have added the definition of archived nitrate in the introduction section:

"…This recycle of snow nitrate not only changes the initially deposited nitrate (isotope) signal, but also leads to a redistribution of snowpack nitrate. Thus, the final archival snow nitrate, defined as the nitrate buried below the photic zone, would largely be impacted by post-depositional processing and disturb the interpretation of ice core nitrate records…"

*Line 694: This also discussed in Shi et al, ACP, 2015.*

**Response:** Shi et al. (2015) study concluded that "*Predicting the impact of post-depositional loss, and therefore changes in the isotopes with depth, is highly sensitive to the depth interval over which an exponential change is assumed.*", which is distinctly different with the conclusion drawn here. As responded earlier, "*the depth interval over which an exponential change is assumed*" is a misunderstanding of the Erbland et al. 2013 study. One has to first determine the depth of the photic zone, but not to assume it.

*Line 703: "...snowpack nitrate can only be explained by the effect of the photo-driven post-depositional processing..." This is the only hypothesis tested here and there are a number of flaws that need to be visited before this conclusion can be drawn.*

**Response:** The detailed response to these "flaws" have been responded earlier and we don't see any issue with this conclusion.

*Line 710: Most of the previous work done does NOT say unequivocally that atmospheric nitrate is solely dependent on primary nitrate.*

**Response:** We agree, and we just wanted to emphasize it.

*In fact, Fibiger et al's work shows that indeed there is an impact of snow sourced nitrate on the atmosphere at Summit! It's just a small part of the pool contained in the snow such that what is ultimately preserved DOES reflect primary nitrate. This is not in fact disproved in this study.*

**Response:** We don't know how "*ultimately preserved DOES reflect primary nitrate*" can be concluded by the Fibiger et al 2013 and 2016 studies. **Their work only relied on atmospheric and surface snow samples in a few months (most in May and June) of a year, and these samples are far from "ultimately preserved".** For the archive what is important is the continuity between atmosphere, surface snow and buried snow up to the photic zone. All reservoirs have to be considered.

*Typographic errors:*
*Line 217: even inconsistence does not make sense grammatically*
*Line 253: was should be were*

*Line 313: Hasting should be Hastings*
*Figure 2 caption: does should be dose?*
*Line 376: remove be at the end of the line*
*Line 406: remove snow before snowpack*
*Line 451: closed should be close*
*Line 713: inconsistence is not a word*
*Line 720: the photo is a typo*
**Response:** Thanks for these suggestions. We have corrected these typos accordingly.

**Reference:**

Berhanu, T. A., Savarino, J., Erbland, J., Vicars, W. C., Preunkert, S., Martins, J. F., & Johnson, M. S. (2015). Isotopic effects of nitrate photochemistry in snow: a field study at Dome C, Antarctica. Atmospheric chemistry and physics, 15(19), 11243-11256.

Castellani, B. B., Shupe, M. D., Hudak, D. R., and Sheppard, B. E.: The annual cycle of snowfall at Summit, Greenland, 120, 6654-6668, https://doi.org/10.1002/2015JD023072, 2015.

Curtis, C. J., Kaiser, J., Marca, A., Anderson, N. J., and Whiteford, E. J. B. D.: Spatial variations in snowpack chemistry, isotopic composition of NO3− and nitrogen deposition from the ice sheet margin to the coast of western Greenland, Biogeosciences., 15, 1-32, 2018, https://doi.org/10.5194/bg-15-529-2018.

Cohen, L., Helmig, D., Neff, W. D., Grachev, A. A., & Fairall, C. W.: Boundary-layer dynamics and its influence on atmospheric chemistry at Summit, Greenland, Atmos. Environ., 41, 5044-5060, https://doi.org/10.1016/j.atmosenv.2006.06.068, 2007.

Dibb, J. E., Talbot, R. W., and Bergin, M. J. G. R. L.: Soluble acidic species in air and snow at Summit, Greenland, 21, 1627-1630, 1994.

Dibb, J. E., Talbot, R. W., Munger, J. W., Jacob, D. J., and Fan, S. M. J. J. o. G. R. A.: Air-snow exchange of HNO3 and NO y at Summit, Greenland, 103, 3475-3486, 1998.

Guy, H., Brooks, I. M., Carslaw, K. S., Murray, B. J., Walden, V. P., Shupe, M. D., ... & Neely III, R. R. (2021). Controls on surface aerosol particle number concentrations and aerosol-limited cloud regimes over the central Greenland Ice Sheet. Atmospheric Chemistry and Physics, 21(19), 15351-15374.

Meusinger, C., Berhanu, T. A., Erbland, J., Savarino, J., & Johnson, M. S. (2014). Laboratory study of nitrate photolysis in Antarctic snow. I. Observed quantum yield, domain of photolysis, and secondary chemistry. The Journal of chemical physics, 140(24), 244305.

Pettersen, C., Bennartz, R., Merrelli, A. J., Shupe, M. D., Turner, D. D., & Walden, V. P. (2018). Precipitation regimes over central Greenland inferred from 5 years of ICECAPS observations. Atmospheric Chemistry and Physics, 18(7), 4715-4735.

Stohl, A. (2006). Characteristics of atmospheric transport into the Arctic troposphere. Journal of Geophysical Research: Atmospheres, 111(D11).

Silvente, E.: Contribution à l'étude de la fonction de transfert air neige en régions polaires, Université Joseph-Fourier - Grenoble I, 1993.

---

## Author Comment (AC2)

We appreciate the reviewers for their time and efforts to review this manuscript. Below we list detailed responses to their suggestions and comments. The suggestions and comments are in italics, followed by the response in normal font with changes highlighted in blue.

**Comments from Reviewer2**

*This paper presents some previously unpublished data on the isotopic values in nitrate in air samples from Summit, Greenland. It also compiles previous data on isotopes in nitrate in air, surface snow and "archived" snow at the same site. The paper then focusses on a discussion of the role of snow photolysis of nitrate in influencing the observed isotope values, their seasonality, and the differences between air and snow.*

*There are a few general comments to make about the paper. Firstly, the new data are potentially very useful and deserve to be published, even if I have a few questions about them. The authors have also done a nice job of compiling previous data, shown in Figure 2, which serves as an excellent starting point for a discussion.*

*The discussion is quite a tough read, even for someone who is quite well-versed in the issues but this probably reflects the difficulty of making clear statements in the light of sparse data, and a definite divergence in opinion between the major groups working on this topic. My main concern with the paper is that some statements are made, sounding quite definite, that are based on differences that are highly marginal. I realise it is disappointing when the conclusion of a study is "we're not sure if this is real", but in some cases this would have been a fairer conclusion. I think the overall structure of the paper and the way it tries to use the different datasets is good, so my comments mainly focus on particular statements that seem too definite or not to be well-supported by the data shown. For that reason, I just go through the paper in order, with both minor and major comments mixed in.*

*Title: the word "reality" seems a bit misplaced here. Of course one can read in the text an undercurrent that the message of the paper is directed at a competing group and that this paper is saying "there really is an effect". But I think for the neutral reader it would be less provocative and more accurate to write "evidence for the postdepositional effect".*

**Response:** We're grateful to the reviewer's comments which make us more cautious about our statements in the paper. We agree that although the complied dataset reveals some general features that are more or less consist with our current understanding, the limited data resolution as well as the large uncertainties somewhat don't allow definite conclusions. We have weakened our statements in many places to better focus on whether the observed systemic trends can be explained by post-depositional processing, instead of directly concluded that these are the "reality". We also change the new title in to "Impacts of post-depositional processing on nitrate isotopes in the snow and the overlying atmosphere at Summit, Greenland".

*Abstract: line 24: since you argue that you have collected aerosol and gas-phase nitrate, the word "aerosol" should be removed here.*

**Response:** Thanks for this suggestion. We have substituted the phrase "aerosol nitrate" into "atmospheric nitrate" throughout the text.

*Abstract: line 28: you should review the wording "no apparent seasonality". I will discuss this later, at line 248.*
**Response:** We have rechecked the atmospheric $\delta^{15}N(NO_3^-)$ data in spring (March to May) and indeed found a significant negative shift compared to other seasons (two side T-test, p = 0.01). We have rewritten the sentence as follows:
"…The atmospheric $\delta^{15}N(NO_3^-)$ remained negative throughout the year, ranging from –3.1 ‰ to –47.9 ‰ with a mean of (–14.8 ± 7.3) ‰, and displayed a minimum in spring which is distinct from the observed spring $\delta^{15}N(NO_3^-)$ maxima in snowpack…".

*Abstract: please review what you have written in the light of edits elsewhere in the paper. I am particularly concerned that lines 37-43 are stronger than the data really allow (see later).*
**Response:** Thanks for this comment. In the revised manuscript, we have reduced the discussions on the relative importance of photolysis-induced fractionation and cage effect on snowpack $\delta^{18}O(NO_3^-)$, and reworded the abstract as follows:
 "…The atmospheric $\delta^{18}O(NO_3^-)$ varied similarly as atmospheric $\Delta^{17}O(NO_3^-)$, with summer low and winter high values. However, the difference between atmospheric and snow $\delta^{18}O(NO_3^-)$ was larger than that of $\Delta^{17}O(NO_3^-)$. We found a strong correlation between atmospheric $\delta^{18}O(NO_3^-)$ and $\Delta^{17}O(NO_3^-)$ that is very similar to previous measurements for surface snow at Summit, suggesting that that atmospheric $\delta^{18}O/\Delta^{17}O(NO_3^-)$ relationships were conserved during deposition. However, we found linear relationships between $\delta^{18}O/\Delta^{17}O(NO_3^-)$ that were significantly different for snowpack compared to atmospheric samples. This likely suggests the oxygen isotopes are also affected before preservation in the snow at Summit, but the degree of change for $\delta^{18}O(NO_3^-)$ should be larger than that of $\Delta^{17}O(NO_3^-)$ given that photolysis is a mass-dependent process.".

*Line 51: Wolff 2008 is not in the reference list, whereas Wolff 1995 is, but is not cited in the paper.*
**Response:** Sorry for this omission. The right reference shall be Wolff et al. (2008) ACP paper. We have added it in the revised manuscript.

*Line 148, 156 and surrounds. Obviously a lot hangs on the quality of the atmospheric data. I have two issues I'd like clarified. The first concerns the use of GF filters. I agree they have often been assumed to collect gas phase nitrate as well as aerosol but the evidence is quite minimal for polar sites; the mechanism is assumed to be through attachment to sea salt loads on the filters (see eg Wagenbach et al, JGR, 103, 11007-11020, 1998 for a discussion of this, albeit related to cellulose filters). Given this I propose that Fig 2 (or a supplementary figure) should show a comparison of the concentrations of nitrate in this study compared to those found in previous studies*

*(including Fibiger and Jarvis) that used mist chambers. This would allow a more informed discussion of whether this study is reporting a similar fraction of total nitrate to earlier studies.*

**Response:** We agree that quantitative collection efficiency is necessary to ensure no artifact in the measured isotope data. The HVAS+GF filter has been shown to be capable to quantitatively collect atmospheric nitrate at Dome C (Erbland et al., 2013) by comparing with the annular denuder method. Erbland et al. (2013) suggested that this is because the high NaCl blank in the GF filter can improve the collection efficiency, as it's known that sea salt aerosol could trap the gaseous nitric acid via chloride-substitute reaction. The similarity of Dome C and Summit is that at both sites gaseous nitric acid dominates total atmospheric nitrate (>90% at Dome C, 94% at Summit, Dibb et al., 1994).

We have compiled the measured atmospheric nitrate concentration in different years at Summit and summarized in the table below. As shown in the table, our results are in general consistent with others, except for Jarvis et al. (2009) who reported a much higher value than all other studies by using mist chamber. This should reflect the collection efficiency and in the revised manuscript, we added this table as SI.

| Year | Month | type | Conc (ng m$^{-3}$) | Reference |
|---|---|---|---|---|
| 1991 | 7-8 | denuder | $38 \pm 53$ | Silvente and Legrand, 1993 |
| 1993 | 6-7 | Mist chamber | $55 \pm 37$ | Dibb et al., 1994 |
| 1993 | 5-7 | Teflon Zefluor filter | $26 \pm 2.9$ | Bergin et al., 1995 |
| 1994 | 5-8 | Mist chamber | $32 \pm 37$ | Dibb et al., 1994 |
| 1995 | 4-7 | Mist chamber | $27 \pm 32$ | Dibb et al., 1998 |
| 2001-2002 | annual | glass fiber filters | $19.9 \pm 19.1$ | This work |
| 2006 | 5-7 | Mist chamber | **202** | Jarvis et al., 2009 |
| 2010 | 5-6 | Mist chamber | $32 \pm 30$ | Fibiger et al., 2016 |
| 2011 | 5-6 | Mist chamber | $42 \pm 22$ | Fibiger et al., 2016 |

*In addition, I am a bit alarmed by the observation that nearly half the collected samples were discarded because they were too close to the blank (for nitrate concentration, I assume). This might imply that there remains a significant blank component in many of the filters that were not discarded and this could then affect the isotopic ratios measured if the blank is contributing significantly. Please comment on this (I would assume you have some isotopic measurements on blank filters?).*

**Response:** Yes, we have collected a total of 9 blank filters during the sampling period and found a significant amount of nitrate in these blank samples. The average nitrate concentration in the extracted filtrate for all atmospheric samples were ($1363 \pm 1603$) ng g$^{-1}$, while that of the nine field blank samples were ($183 \pm 44$) ng g$^{-1}$. For the sake of validity, we excluded the samples with concentration less than three times of the average concentration of these blank samples in further analyses. This procedure excluded a fair amount of total samples especially in September. The blank samples were also collected together to measure its isotope ratio the same as measuring atmospheric samples. And we deducted the contribution from the blank to obtain the real isotope ratio for atmospheric samples. We have added these details about sample handling in the revised manuscript.

*Figure 1: I find it a bit strange that you choose not to plot the data chronologically but instead that you have Jan to Jul 2002 followed by Jul-Dec 2001. I would propose plotting the data chronologically (jul-Jul) in Fig 1, and from Jan-Jan in Fig 2.*

**Response:** Thanks for this suggestion. We now plot the data chronologically in the revised manuscript.

[Figure]

*Line 208-209. I don't request a change but I note that this is a bit circular. You use the similarity in seasonality to support your seasonal assignments in snowpack, and then later you use the same alignment as evidence that capdelta-17O in particular is unaffected by photochemistry.*

**Response:** We agree. But $\Delta^{17}O$ is not the sole indicator to support our snowpack dating, we also used $Na^+$ (peaked in winter) and $Cl^-/Na^+$ ratio (peaks in summer) to distinguish the seasons.

*Line 243. You attribute spikes to Arctic Haze events. Could it also be that it reflects more efficient scavenging during inputs of high sea salt (you would be able to support or deny this by looking at seasalt in the aerosol data)?*

**Response:** This is a good suggestion. Unfortunately, the high NaCl blank of the filters prevent this. We have checked the seasonality of aerosol $Na^+$ concentration at Summit

form another study (Rhodes et al., 2017) and found the $Na^+$ concentration is generally higher from January to May than June to September. This seems to coincide with our observation that nitrate concentration mostly peaks in April and May. Thus we cannot rule out this possibility. We add the following statement in our revised text:

"…There was no distinct seasonal pattern in atmospheric nitrate concentrations, but some spikes (samples with much higher nitrate concentrations than average) in spring/summer months were observed, typical to intrusion of Arctic haze events at the altitude of the Ice Sheet (Quinn et al., 2007; Jaffrezo et al., 1997). Alternatively, these nitrate concentration spikes could reflect more efficient scavenge of atmospheric nitrate by sea salt aerosol during transport, as indicated by the elevated $Na^+$ concentration in Summit aerosol during April and May (Rhodes et al., 2017).".

*Line 248. You say there is no distinct seasonality in the atmospheric 15N. But I look at Fig 2b, where you also show the snowpack 15N from Geng, which you claim has a clear seasonality. While obviously the aerosol data have large variability within each month, I see just as strong a seasonal dip in the aerosol data as I see a seasonal peak in the snowpack data. In the end this isn't crucial because it's the differences between the air and the snowpack in different months (which is clear) that you focus on, but still please reword more cautiously.*

**Response:** Thanks for pointing this out. We have rechecked our dataset and found that the spring $\delta^{15}N(NO_3^-)$ was in fact on average lower than other seasons (two-side t-test, p=0.001). In the revised manuscript, we have rewritten the relative statement as following:

"…The atmospheric $\delta^{15}N(NO_3^-)$ was negative throughout the year with an annual mean of (–14.8 ± 7.3) ‰. The springtime atmospheric $\delta^{15}N(NO_3^-)$ exhibited a significantly lower shift compared to other seasons (two-side t-test, p = 0.001), and the average for the winter half year (–12.0 ± 4.2) ‰ was slightly higher than that in the summer half year (–16.0 ± 3.9‰) …".

*Line 257: "($^{18}O$) displayed an almost identical seasonal pattern with $\Delta^{17}O(NO_3)$ as expected". I'm sure if I'd read your previous papers I would know why this was expected but it's not obvious, given that the former is a mass dependent fractionation and the latter is a mass independent one that could be quite separate. Please spell out why it's expected.*

**Response:** Thanks for this comment. We have added the following statement in the revised version:

"…The atmospheric $\delta^{18}O(NO_3^-)$ data ranged from 49.7 to 86.5 ‰ and displayed an almost identical seasonal pattern with $\Delta^{17}O(NO_3^-)$. The similar seasonality between $\delta^{18}O(NO_3^-)$ and $\Delta^{17}O(NO_3^-)$ is expected. At the seasonal scale, the primary controlling factor of atmospheric $\delta^{18}O(NO_3^-)$ and $\Delta^{17}O(NO_3^-)$ is the relative importance of $O_3$ versus $HO_x$ to nitrate formation in different seasons. In summer, HOx oxidation is more important and leads to nitrate with lower $\delta^{18}O(NO_3^-)$ and $\Delta^{17}O(NO_3^-)$, while in winter $O_3$ oxidation is more important and leads to higher $\delta^{18}O(NO_3^-)$ and $\Delta^{17}O(NO_3^-)$ (Alexander et al., 2020; Michalski et al., 2012)".

*Figure 2. In part a, please clarify that the curve refers to the actinic dose (not "does") that would have been experienced by the snowpack samples.*
**Response:** Thanks for this suggestion. We have made more detailed illustrations in the caption as follows:
"The cumulative UV-B* dose represents the actinic dose that would have been experienced by snow nitrate deposited at different times of a year."

*Fig 2e: I assume these seasalt data refer to the snowpack data (but then which: Geng et al?). Please clarify this in the caption. Also, I'd be really surprised if the Na are in mg/L, surely they are ppb or ug/L?*
**Response:** Thank you so much for this, the data is from Geng et al. (2014), and should be at $\mu g \ L^{-1}$. We have made the relevant corrections in the revised manuscript.

*Fig 2 caption. "The vertical lines represent the interval of seasons". I don't understand what this means. Are the error bars the differences between years for the same month/period, or are they the variability within a month or season. This is crucial to understanding what values are significantly different to others.*
**Response:** The four dashed lines in Fig 2 represent the beginning and ending of each season (simply defined as March $1^{st}$, June $1^{st}$, September $1^{st}$ and December $1^{st}$, respectively). The error bars represent one stander error calculated from all data available from a certain month or season from all years (some of them have data point from one year, but others have data from several years). We have made this clearer in the revised manuscript.

*Line 290. This is the first case where I really feel you say things the data don't support. You refer to a progression of 15N from atmosphere to surface to snowpack. However when looking at Figure 2, it would be really stretching it to say that the surface snow data of Fibiger et al are significantly different from the snowpack data, taking into account the error bars shown. I agree there is a difference between the single Jarvis data point but as you later question this data I don't feel it's justified to make a wide-ranging and repeated statement about a progression on the basis of that. To me this is a place where you have to say that there is a clear difference between atmosphere and snow, but the data are insufficient to state with any certainty whether the surface snow and snowpack are different. The same issue is repeated in line 340. (As an aside if the Jarvis atmospheric data in Fig 2b are right then the variability in the atmosphere between years is also too high to make a clear statement but I think it's Ok just to have noted the discrepancy).*
**Response:** We agree that the Fibiger's surface snow $\delta^{15}N(NO_3^-)$ data are not very different with the snowpack $\delta^{15}N(NO_3^-)$ data compiled from Geng et al. (2014). However, **when looking at the Jarvis et al. (2009) original surface snow and snowpack data,** these differences are quite clear, i.e., surface snow $\delta^{15}N(NO_3^-)$ in spring and summer were (-6.8 ± 0.5) ‰ and (-2.5 ± 1) ‰ respectively, while snowpack $\delta^{15}N(NO_3^-)$ in spring and summer were (4.4 ± 1.9) ‰ and (2.4 ± 2.1) ‰

respectively. These are also seen in Figure 2b but their snowpack data points were somewhat hidden behind the data from Geng et al (2014). In the revised manuscript, we have changed the color of the symbols to emphases the Jarvis et al 2009 data. Note the Jarvis 2009 atmospheric data is not justified (high concentration compared to all other studies), but **their surface snow and snowpack data should be valid**. What is more, their surface snow and snowpack samples are corresponding to each other for the same season, see the figure below, and a difference is clearly seen for spring and summer. So we think the difference between surface snow and snowpack is real, but Jarvis et al. (2009) only reported seasonal averages (the original data in the below figure was not available) so we only plotted the seasonal averages in Figure 2b of our manuscript.

[Figure]

**Figure 1.** The comparison between surface snow (solid circles) and snowpack $\delta^{15}N(NO_3^-)$ (open square and triangles for two different snowpits) at Summit (Jarvis et al., 2009).

*Fig 3: Why do you only show J against SZAs (in the inset) almost entirely smaller than those experienced at Summit in the main figure?*
**Response:** Thanks for this suggestion. We have narrowed the range of SZA in the inset.

*Line 413, Fig 3 and surrounding discussion. I am really mystified by this lengthy section. Of course it's a nice advance that you can find a simple formula to represent the complex output of the model for PIE. However you don't then use it. Apparently PIE is the "difference between surface snow 15N(NO3–) and archived snow $\delta^{15}N(NO_3^-)$". So why don't you plot the actual data in Fig 3 and see if it agrees with the model and its simplified (eq 3) representation. And of course the answer is that it doesn't. The observed PIE in May-July looks from Fig 2b to be about 3 permil, not the predicted values of about 10. (We can argue about the significance of the single value for spring, which you later suggest you don't believe). In any case my point is that there is no point having this section and figure unless you also show and discuss*

*the data.*

**Response:** The comparison between the modeled PIE (using TRANSITS model) and observed PIE has been presented in our previous paper in a more direct manner (please see the Fig2 below; Jiang et al., 2021). The observed PIE was calculated based on the observations in Jarvis et al. (2009). Again, the Jarvis et al. (2009) atmospheric data is not justified because of the high concentration, but we didn't find reasons why their snow data are also not justified. From the Figure 2 below it can be seen that the modelled PIE is generally consistent with the observations (adapted from Jiang et al., 2021). We think that the observed PIE is strong evidence of post-depositional processing occurring at Summit, though it was not the major conclusions in this study (detailed in Jiang et al., 2021 and we tried to avoid repeating the same discussion or comparison).

In addition, in this study we wanted to provide a simplified (but accurate) equation that we could be easily used by others to evaluate the impact of photolysis on snowpack $\delta^{15}N(NO_3^-)$ without a complex modelling exercise. We have stated it more clearly in our revised manuscript as follows:

"To better understand the effects of the photo-driven post-depositional processing, we quantitatively compared and analyzed the $\delta^{15}N(NO_3^-)$ averages in spring when the isotopic differences between surface snow and snowpack are the most pronounced as indicated by the compiled data and the modeling results by Jiang et al. (2021). Since the surface snow $\delta^{15}N(NO_3^-)$ data in Fibiger et al. (2016) only covered two months, we mainly focus on the seasonal data covering two years from Jarvis et al. (2009). However, we note the Fibiger et al. (2016)'s surface snow $\delta^{15}N(NO_3^-)$ data was remarkably higher than Jarvis et al. (2009) for the same months, which likely indicated the heterogeneity among data from different years. Compared to surface snow nitrate, snowpack nitrate was enriched by $(12.8 \pm 2.6)$ ‰ in spring, as seen in Fig 2b. This value should reflect the effect of post-depositional processing on snow nitrate throughout its preservation, i.e., time from being deposited to the surface to being archived below the photic zone. In Jiang et al. (2021), this effect was defined as PIE, i.e., the photo-induced isotope effect, and calculated as the difference between surface snow $\delta^{15}N(NO_3^-)$ and archived snow $\delta^{15}N(NO_3^-)$. The PIE in spring calculated by the TRANSITS model is averaged at $(14.3 \pm 1.1)$ ‰, consistent with the observations. Calculating the PIE only requires one to compute the relative nitrate loss induced by nitrate photolysis, which makes the PIE independent of the initially deposited nitrate $\delta^{15}N$ and a good tracer of the isotopic effect of post-depositional processing. Here we propose a simplified formula of PIE for quick assessment of the photo-driven post-depositional processing effect on $\delta^{15}N(NO_3^-)$ at any sites of interest:".

[Figure]

**Figure 2**. Modelled PIE (Jiang et al., 2021) compared with the observed PIE from Jarvis et al. (2009).

*Line 423 – "J also varies with depth". This is wrong because J is the surface photolysis rate constant. The exponential term changes the actinic flux seen at depth.*
**Response:** Thanks for pointing out this. We have rephrased as "Both $\varepsilon$ and $J$ varies seasonally owing to the time-varied actinic flux, while the decrease of nitrate photolysis rate constant with depth is constrained by the exponential term".

*Line 458-9. This is a strange statement in that it's not clear how a data point obtained by another group can be "explored and confirmed".*
**Response:** We agree and have deleted this sentence in the revised version.

*Line 680, 685 and what follows, plus 715 and following. A quite definite conclusion is based on the assertion that the slope of the snowpack data is significantly different from that of the atmosphere (not aerosol) and surface snow data. Your reported uncertainties on the slopes might indeed suggest that but sometimes it's better just to look at the data – would anyone really say that the yellow data are a significantly different population from the blue and black data? In fact you suggest that what is happening is an enrichment of 18O in the snowpack samples. But again, just look at the data: it's quite obvious that if there is a difference it is that the snowpack data have a subpopulation that is enriched in 17O (I admit I am not sure how that leads to a lower slope and indeed by eye its very hard to see how the yellow points can have a slope of 0.3). Please reconsider this whole discussion; I think you are building a lot on very shaky differences.*
**Response:** We double checked the regression line calculation, and found the same slopes. Although the snowpack data look like a subgroup of atmosphere and surface snow data, the time resolution for these data are quite different. For example, the thickness of surface snow is ~1-2 cm per sample while the snowpack is about 5 cm per sample (Geng et al., 2014). Given the annual accumulated snow depth at Summit is ~65 cm, the time resolution of surface snow is likely to be weekly, while for snowpack samples it shall be monthly. We note that the atmospheric samples

resolution is 3 days, so we feel it's unfair to directly compare how snowpack $\Delta^{17}O(NO_3^-)/\delta^{18}O(NO_3^-)$ was shifted as they had been averaged in time owing to the limited resolution.

We agree that using the regression relationship to infer how the $\Delta^{17}O(NO_3^-)/\delta^{18}O(NO_3^-)$ relationship changed after deposition is suspicious as it highly depends on the assumption of little changes in snowpack $\Delta^{17}O(NO_3^-)$. In the revised manuscript, we treat the different linearity as evidence of post-depositional processing altering snowpack oxygen isotope but avoid further discussion about the relative importance of photolysis fractionation and cage effect. The revised text is shown as follows below:

"Fibiger et al. (2013) found a strong linear relationship between their measured $\Delta^{17}O(NO_3^-)/\delta^{18}O(NO_3^-)$ in surface snow samples at Summit. Based on this relationship they proposed a direct transfer of atmospheric oxygen isotope signals to surface snow at Summit. However, as discussed in Jiang et al. (2021), this relationship should not be viewed as evidence of little to no post-depositional processing. Instead, examining the $\Delta^{17}O(NO_3^-)/\delta^{18}O(NO_3^-)$ relationships among atmospheric, surface snow and snowpack samples may provide some clues on whether or not the photo-driven post-depositional processing impacts the $\Delta^{17}O(NO_3^-)/\delta^{18}O(NO_3^-)$ ratio, since post-depositional processing influences $\Delta^{17}O(NO_3^-)$ and $\delta^{18}O(NO_3^-)$ differently. We note that different types of observations are different in their time resolutions. Our atmospheric measurement is typically 3 days per sample, while the surface snow samples (1-2 cm thickness) in Fibiger et al. (2013) represented weekly accumulation and snowpack sample resolution (5 cm per sample, Geng et al., 2014) is closer to monthly resolution. The linearity in surface snow shall not be changed by aggregation if post-depositional processing was negligible. Here we plotted our atmospheric and snowpack $\Delta^{17}O(NO_3^-)/\delta^{18}O(NO_3^-)$ data together with the four months (in year 2010 and 2011) of surface snow data from Fibiger et al (2013) in Figure 4.

As shown in Figure 4, the linear relationship between atmospheric $\Delta^{17}O/\delta^{18}O(NO_3^-)$ ($\Delta^{17}O(NO_3^-) = (0.44 \pm 0.04) \times \delta^{18}O(NO_3^-) - (3.45 \pm 3.28)$, r = 0.81) is very similar to the reported surface snow relationship ($\Delta^{17}O(NO_3^-) = (0.41 \pm 0.01) \times \delta^{18}O(NO_3^-) - (3.19 \pm 0.41)$, r = 0.90) despite their different time coverages. Such a relationship suggests that the linearity of $\Delta^{17}O/\delta^{18}O(NO_3^-)$ in surface snow may directly originate from atmospheric nitrate, consistent with the conclusion of Fibiger et al. (2013). The conservation of $\Delta^{17}O/\delta^{18}O(NO_3^-)$ relationship during deposition is somehow unexpected, as the current observed air-snow $\delta^{18}O(NO_3^-)$ difference is highly variable in both magnitude and sign (Jarvis et al., 2009; Fibiger et al., 2016). Further studies are required to understand why these observed atmospheric $\delta^{18}O(NO_3^-)$ are so different between different years. However, in the snowpack data, the linearity between $\Delta^{17}O$ and $\delta^{18}O(NO_3^-)$ ($\Delta^{17}O(NO_3^-) = (0.30 \pm 0.06) \times \delta^{18}O(NO_3^-) + (6.72 \pm 5.29)$, r = 0.58) was significantly different from that of atmosphere or surface snow nitrate, suggesting that post–depositional processing likely has changed the originally deposited oxygen isotope signals up on archival. We note that similar observations, i.e., better linearity of $\Delta^{17}O/\delta^{18}O(NO_3^-)$ in atmosphere and surface snow nitrate than that in the whole snowpack, were observed at Dome C where the photolysis of snow nitrate has been

unambiguously shown to be dominant (Erbland et al., 2013). This emphasizes again that, when evaluating the degree of post–depositional processing, one should consider samples covering all depths of the photic zone, not only surface samples.".

*Lines 724-736. Yes, I like this paragraph.*
**Response:** Thanks for your comments. We are aware of the limitations of the current dataset and looking forward to more systematic studies at Summit in the future.

**Reference:**
Dibb, J. E., Talbot, R. W., and Bergin, M. J. G. R. L.: Soluble acidic species in air and snow at Summit, Greenland, 21, 1627-1630, 1994.

Dibb, J. E., Talbot, R. W., Munger, J. W., Jacob, D. J., and Fan, S. M. J. J. o. G. R. A.: Air-snow exchange of HNO3 and NO y at Summit, Greenland, 103, 3475-3486, 1998.

Silvente, E.: Contribution à l'étude de la fonction de transfert air neige en régions polaires, Université Joseph-Fourier - Grenoble I, 1993.

Jiang, Z., Alexander, B., Savarino, J., Erbland, J., and Geng, L.: Impacts of the photo-driven post-depositional processing on snow nitrate and its isotopes at Summit, Greenland: a model-based study, The Cryosphere, 15, 4207-4220, 10.5194/tc-15-4207-2021, 2021.

---

## Author Comment (AC3)

We appreciate the reviewer for the time and efforts to review this manuscript. Below we list detailed responses to the suggestions and comments. The suggestions and comments are in italics, followed by the response in normal font with changes highlighted in blue.

**Comments from Reviewer3**

*Review of Atmospheric and snow nitrate isotope systematics at Summit, Greenland: the reality of the post-depositional effect by Jiang et al.*

*As said in the title this paper sets out to document the reality and state of knowledge of the post-depositional effect which changes the isotope distributions in nitrate after deposition and before archiving. The paper presents some nice data, but unfortunately it is not always strong enough to make definite conclusions. The discussion is sometimes too speculative as detailed below.*

*Please define exactly what is meant by 'post-depositional'. I presume it means 'after deposition to the surface and before becoming part of the permanent archive'. How long is this period? Why does post-depositional processing end? What evidence is there that there are not also long-term changes in deep ice? What is the physical mechanism ending post depositional processing?*

**Response:** Thanks for these comments. As all three reviewers pointed out, we realized that some of our conclusions are too strong based on the compiled data presented here. We have weakened some of our original discussions especially those regarding the relationship between $\Delta^{17}O(NO_3^-)/\delta^{18}O(NO_3^-)$, as well as the title. Please see our detailed responses below.

Post-depositional processing involving snow nitrate includes evaporation of gaseous $HNO_3$ and photo-decomposition at UV wavelengths (mainly 290-350 nm). Current studies have suggested that photolysis dominates snow nitrate loss (Erbland et al., 2013; Frey et al., 2009; Shi et al., 2019), so in this study "post-depositional" only considers the photolysis of snow nitrate. Since the actinic flux rapidly attenuates in the upper 30 to 60 cm snow layer (i.e., the photic zone), snow buried below the photic zone is considered to be archived. While the burial speed is determined by snow accumulation rate and the depth of the photic zone, which could be as short as about half a year (e.g., at Summit) or as long as up to ten years (Erbland et al., 2013; Shi et al., 2015) at Dome A/Dome C in East Antarctica. We have refined these relative explanations in the revised manuscript to make it clearer for readers who are not already familiar with these terms. We have added the following statement in our revised manuscript:

"…Thus, the final archival snow nitrate, defined as the nitrate buried below the photic zone, would largely be impacted by post-depositional processing and be important to consider in the interpretation of ice core nitrate records. The degree of the photo–driven post–depositional processing is influenced by three main factors including snow accumulation rate, surface actinic flux and light penetration depth in snow…"

*The abstract should do a better job of communicating the impact and implications of the study. What is known now that was unknown or uncertain before?*

**Response:** We thank the reviewer for this comment. In the revised manuscript, we have added one more sentence to state the implications of this study:

"Although with uncertainties, the data compiled in this study suggested post-depositional processing at Summit can results in changes in nitrate isotopes, especially $\delta^{15}N(NO_3^-)$, consistent with a previous modeling study. This reinforces the importance understanding the effects of post-depositional processing before ice-core nitrate isotope interpretation, even for sites with relatively high snow accumulation rate".

*It is clear that this research group is quite accomplished at the methods used. The main issue is in the value and implications of the results that are obtained. After decades of research on isotopic abundances in snowpack nitrate, I would ask the authors to make a clear statement in the discussion or conclusion about the state of the field, both what has been learned, and what the information could be used for if only post depositional processing could be understood in detail. I get the impression that there will always be some uncertainty. For example the uncertainties in delta values in the abstract are around 50%, and similar large uncertainties are shown in Figure 2. How much would these uncertainties have to be reduced in order to be able to derive useful numbers from the nitrate record, and is it reasonable to believe that this can be achieved?*

**Response:** Thanks for the suggestions. In the conclusion, we have added a paragraph as asked by the reviewer:

"Nitrate isotopes in polar ice cores have been sought to reflect past changes in NOx emissions and atmospheric oxidation environments (Alexander et al., et al., 2015; Hastings et al., 2005, 2009; Geng et al., 2014, 2017, Wolff, 1995). Although some important progress has been made (e.g., Geng et al., 2017), most interpretations of ice core nitrate records remain qualitatively because the effects of post-depositional processing on nitrate and its isotopes have not been quantified. The latter requires a comprehensive understanding of the degree of post-depositional processing, as well as its influences on ice-core nitrate isotope preservation at different time scales. This is also true for ice-core drilling sites with high snow accumulation rates, where to what degree nitrate isotopes are changed upon archival is a subject of debate (Fibiger et al., 2013; Geng et al., 2015; Hastings et al., 2005; Jiang et al., 2021).

To address this debate, in this study, we reported ..."

What is more, in the end of the second paragraph of the conclusion, we have added the following statements in the revised manuscript:

"…These conclusions reinforce the importance of quantitative assessment of the post-depositional processing on snow nitrate isotopes even at sites with relative high snow accumulation rate (Jiang et al., 2021). Further numerical modeling is needed to correct the post-depositional processing effects on $\delta^{15}N(NO_3^-)$, which is essential to the better use of snowpack/ice core $\delta^{15}N(NO_3^-)$ to retrieve information regarding the historical variability in NOx sources (Hasting et al., 2004, 2009).".

*Line 109, add a reference for the cage effect mechanism.*

**Response:** We have added McCabe et al. (2005) and Meusinger et al. (2014) as references therein.

*Line 380, are there physical mechanisms that could explain the spring-summer differences such as recrystallization?*

**Response:** We think recrystallization is irrelevant to the discussions here as we are seeking the explanation for why the atmospheric $\delta^{15}N(NO_3^-)$ is most depleted in spring instead of in summer, as the photolytic NOx flux from snow maximizes in summer with very depleted $\delta^{15}N$ values. Our hypothesis is that owing to the more unstable boundary layer in summer that favors the export of NOx instead of locally reforming nitrate, or an increase in $\delta^{15}N$ of primary nitrate which also contributes to local atmospheric nitrate. So far no known physical mechanisms can explain this detail of the observed seasonality.

*Please discuss the origin of the time lag between the mean SZA and the PIE plot shown in Figure 3. Very nice data here, thank you.*

**Response:** Thanks for the comment. SZA is the smallest in summer when actinic flux is the maximum, but PIE is determined by the total amount of actinic flux received by nitrate in snow between deposition to the surface and burial below the photic ozone. Owing to polar winter when there is no sunlight, over a year nitrate deposited in spring received the most actinic flux (accumulated UV-B dose in Figure 2). As a result, PIE is the largest in spring instead of in summer when actinic flux is the strongest. We have explained this in the manuscript.

*At line 498, it is not clear what 'kinetic adsorption' is and how this is different from 'adsorption'. Do you mean to say that at Summit, given higher snowfall, scavenging of nitrate is complete, while it is incomplete at Dome-C? Please rewrite and clarify, to benefit those outside your immediate research field.*
*At times the discussion is speculative and I would encourage the authors to keep it tight and focused - give numbers and reasons and try to conserve ink.*

**Response:** The "kinetic adsorption" is not different from "adsorption". To avoid confusions, we have deleted "kinetic" in the revised manuscript.

*Line 714, 'We analysed the relationships and found that the linearity of ..in snowpack is different from that of atmospheric and surface snow.' I am confused because isn't linearity always linearity? Maybe there is another word such as slope or curvature, that would be more appropriate.*

**Response:** Here we meant the slope of the regression between snowpack $\Delta^{17}O/^{18}O(NO_3^-)$ are different with that of atmospheric and surface snow $\Delta^{17}O/^{18}O(NO_3^-)$. We have avoided the use of the word "linearity" in the revised manuscript. Thanks.

*Technical*
*The writing is generally fine with only some minor issues that are easily addressed with*

*a good proofreading.*

*Line 130, I suggest change to read '...post-depositional processing, snow samples covering the entire photic zone must be considered.'*

**Response:** Thanks for this suggestion. We have changed the sentence accordingly.

*Line 132, check the sentence 'To thoughtfully evaluate', in simplified form it seems to say, 'Nitrate isotopes are necessary.' Please rewrite, just simply and clearly.*

**Response:** We have shortened this sentence as follows:

*"To thoughtfully evaluate the effects of post-depositional processing at Summit, nitrate isotopes in the atmosphere and in snow covering a full cycle of polar seasons with distinct actinic flux variations are necessary…".*

**Reference:**

Erbland, J., Vicars, W., Savarino, J., Morin, S., Frey, M., et al.: Air–snow transfer of nitrate on the East Antarctic Plateau-Part 1: Isotopic evidence for a photolytically driven dynamic equilibrium in summer, Atmos. Chem. Phys., 13, 6403-6419, https://doi.org/10.5194/acp-13-6403-2013, 2013.

Frey, M. M., Savarino, J., Morin, S., Erbland, J., & Martins, J.: Photolysis imprint in the nitrate stable isotope signal in snow and atmosphere of East Antarctica and implications for reactive nitrogen cycling, Atmos. Chem. Phys., 9, 8681-8696, https://doi.org/10.5194/acp-9-8681-2009, 2009.

Geng, L., Alexander, B., Cole-Dai, J., Steig, E. J., Savarino, J., Sofen, E. D., & Schauer, A. J.: Nitrogen isotopes in ice core nitrate linked to anthropogenic atmospheric acidity change, Proc. Natl. Acad. Sci., 111, 5808-5812. https://doi.org/10.1073/pnas.1319441111, 2014.

McCabe, J., Boxe, C., Colussi, A., Hoffmann, M., & Thiemens, M.: Oxygen isotopic fractionation in the photochemistry of nitrate in water and ice, J. Geophys. Res. Atmos., 110, D15310, https://doi.org/10.1029/2004JD005484, 2005.

Meusinger, C., Berhanu, T. A., Erbland, J., Savarino, J., & Johnson, M. S.: Laboratory study of nitrate photolysis in Antarctic snow. I. Observed quantum yield, domain of photolysis, and secondary chemistry, J. Chem. Phys., 140, 244305, https://doi.org/10.1063/1.4882898, 2014.

Shi, G., Buffen, A. M., Hastings, M. G., Li, C., Ma, H., et al.: Investigation of post-depositional processing of nitrate in East Antarctic snow: isotopic constraints on photolytic loss, re-oxidation, and source inputs, Atmos. Chem. Phys., 15, 9435-9453, https://doi.org/10.5194/acp-15-9435-2015, 2015.

Shi, G., Chai, J., Zhu, Z., Hu, Z., Chen, Z., et al.: Isotope fractionation of nitrate during volatilization in snow: a field investigation in Antarctica, Geophys. Res. Lett., 46, 3287-3297, https://doi.org/10.1029/2019GL081968, 2019.

---

## Author Response (AR2)

We are grateful to the reviewers and the editors for their time and efforts to review this manuscript. We have taken efforts to improve the grammar and structure of the sentences to make the manuscript more readable. Below we list detailed responses to the editor's comments and suggestions. The comments and suggestions are in italics, followed by the response in normal font with changes highlighted in blue. The line number for each modification in the change tracked version is also listed.

**Comments from the Editor**
*Minor Edits:*
1) *In the Methods section, please include your method for correcting data with the nitrate blank concentration and isotopic composition, as you describe in your response to the referees.*
**Response:** We have added the following text in the revised manuscript:
"…reference material solution using 1M NaCl solution. The blank filter samples were processed following the same procedure as atmospheric samples and measured for their isotope ratios. The measured nitrate isotope ratio of each atmospheric sample was further corrected by deducting the contribution from filter blanks.", line 211-213.

2) *In the Methods section, please include your explanation of how the uncertainty associated with the isotopic measurements was calculated, as you describe in your response to the referees.*
**Response:**
"…deducting the contribution of the filter blanks. The measurement uncertainty was assessed based on the reduced standard deviations of the residuals from the linear regression between the measured reference materials and their expected values as detailed in Erbland et al. (2013). The overall measurement uncertainties…", line 215-217.

3) *In Line 801, the authors state the snowpack data was "significantly different" from that of the atmospheric and surface-snow nitrate. Please provide statistics to back of this statement of significance or change the wording here.*
**Response:** We have changed the word "significantly" to "distinctly".

4) *Please double-check your reported d15N values from Jarvis et al (2009), and one referee notes that the values reported in your manuscript are different than the 2006 and 2007 data reported in Jarvis et al (2009). If your reported values are an average of the two years, please include that information in the manuscript.*
**Response:** We have checked the $\delta^{15}N(NO_3^-)$ reported in Jarvis et al. (2009) and are sure that the used value is consistent. Jarvis et al. (2009) reported monthly means of atmospheric $\delta^{15}N(NO_3^-)$ and seasonal means of surface snow and snowpack over **two years** (i.e., 2006 and 2007), and in this manuscript we used the averages of the two years. We have added more explanations in the main text:
"…for samples with coarser than monthly resolution, seasonal averages were used, and

we here reported seasonal averages of multiple years if more than one year's data are available in the literature.", line 234-235.

5) *In Lines 401-403, please add justification for the reader as to why the Fibiger et al. (2016) data is out of range and not shown.*
**Response:** We have added the following justification in the main text in the revised manuscript:
"…In addition, the averaged $\delta^{18}O(NO_3^-)$ of atmospheric nitrate in gas-phase samples collected by Jarvis et al. (2009) in March and June is (34.1 ± 1.7) ‰, and by Fibiger et al. (2016) in May and June is (54.2 ± 8.5) ‰ for the year of 2010 and (90.5 ± 12.5) ‰ for the year of 2011. These values are out of range of the snow samples as well as our atmospheric samples, and in order to better show the seasonality of $\delta^{18}O(NO_3^-)$ in snow and atmospheric samples as indicated by other data, we didn't plot these data in Figure 2d…", line 385-391.
      The caption of Figure 2 has also been changed accordingly:
"…The atmospheric $\delta^{18}O(HNO_3)$ data in Fibiger et al. (2016) and Jarvis et al. (2009) are both out of range of the snow samples as well as our atmospheric samples thus are not shown here.", line 408-410.

6) *The referee's suggestion of citing the Shi et al. study should be included as its evidence for a small to minimal change in the isotopes can be used to back up this study's claim that this process is not important under the conditions at Summit.*
**Response:** We have added the following text in the revised manuscript:
"…Jiang et al. (2021) have discussed the effect of the physical release on nitrate isotopes and suggest that this effect is negligible at Summit. This is because that the physical release rate and the associated isotope effects are relatively small at cold temperatures. Shi et al. (2019) performed field $NO_3^-$ volatilization experiments and found no isotope fractionation occurring in $\delta^{15}N(NO_3^-)$ when the temperature was set to –24 °C. When the temperature increased to –4 °C, a small positive fractionation constant (4.9 ± 2.1‰) was observed, while at Summit the temperature is below –10 °C throughout the year as shown in Figure 1a.", line 429-435.

7) *Supplementary Table 1: either explain why the Jarvis et al. (2009) data is bolded, or remove the bold font here*
**Response:** We remove the bold font in the table.

*Technical Corrections:*
*Line 33: should be "atmospheric"; remove the comma*
*Line 34: add "the" before "snowpack"; change to "…, while the surface snow… was in between the atmosphere and the surface snow."*
*Line 35: change to "atmospheric"*
*Line 39: add "the" before "snowpack" here*
*Line 75: add "an" before "increasing"; add "sites" after "inland"*
*Line 84: add comma after "contribute"*

*Line 89: change "in general" to "generally"*

*Line 93: change "recycle" to "recycling"*

*Line 109: change to "periods"*

*Line 127: add "a" before "seasonal"*

*Line 129: add "an" before "annual"*

*Line 133: changes to "scales"*

*Line 139: add "the" before "snowpack"*

*Line 153: add comma after "depth"*

*Line 154: add comma after "zone"*

*Line 169: change "critical to assess" to "critical for assessing"*

*Line 172: capitalize "Atmospheric" in the subsection title here*

*Line 174: remove "by"*

*Line 177: change "till" to "until"*

*Line 185: add "that" after "assumed"*

*Line 188: should this be "field" instead of "filed"?*

*Line 197: change "were" to "was"*

*Line 198: change "were" to "was"*

*Line 199: add "the" before "blank"*

*Line 200: change to "…concentrations exceeding 3 times that of the blank…"*

*Line 214: change to "atmospheric"*

*Line 232: change to "atmospheric"*

*Line 251: add "to" after "matched"*

*Line 257: change to "atmospheric"*

*Line 267: "inconsistence" is not a word, perhaps "being inconsistent" instead*

*Line 269: add "the" before "current"*

*Line 297: change to "… a more efficient scavenging…"*

*Line 343: missing a ")" somewhere in here*

*Line 380: add "the" before "current"*

*Line 400: change to "standard"*

*Line 402: add "and" before "thus"*

*Line 421: change to "have discussed"*

*Line 422: change to "suggest"*

*Line 423: change to "… discuss the other processes and compare…"*

*Line 428: change to "types"*

*Line 453: change "to" to "for"*

*Line 606: add "the" before "total"*

*Line 752: change to "appear"*

*Line 754: add "the" before "local"*

*Line 801: change to "atmospheric"*

*Line 817: should this be "thought" instead of "sought"?*

*Line 821: change to "qualitative"*

*Line 857: change to "modeling"*

*Line 862: change to "times"*

**Response:** We appreciate a lot for the editor's careful checking on these typos and we

have revised them accordingly.

*Lines 49-52: multiple referees are confused by this sentence because the isotopes are by photolysis. Please clarify or reword here.*
**Response:** The original sentence has been revised as follows:
"…This likely suggests the oxygen isotopes are also affected before preservation in snow at Summit, but the degree of change for $\delta^{18}O(NO_3^-)$ should be larger than that of $\Delta^{17}O(NO_3^-)$. This is because photolysis is a mass-dependent process that would directly affect $\delta^{18}O(NO_3^-)$ in snow but not $\Delta^{17}O(NO_3^-)$ as the latter is a mass-independent signal…", line 49-51.

*Line 52: the phrase "Although with uncertainties,…" doesn't make sense. Please revise.*
**Response:** The original sentence has been revised as follows:
"…Although there were uncertainties associated with the complied dataset, the results suggested that post-depositional processing at Summit can induce changes in nitrate isotopes especially $\delta^{15}N(NO_3^-)$, consistent with a previous modeling study…", line 51-53.

*Line 85: perhaps "suggested to have a minimal effect under…" instead; perhaps "…under typical ranges of temperatures in polar regions" instead*
**Response:** The original sentence has been revised as follows:
"The evaporation of nitrate from snow grains may also contribute, but this process has been suggested to only have a minimal effect under typical ranges of temperatures in the polar regions (Shi et al., 2019).", line 83-84.

*Lines 96-97: reword the phrase "…and this need to be fully understanding when interpreting ice core nitrate records." Perhaps ", which needs to be fully understood to interpret ice-core nitrate records."*
**Response:** The original sentence has been revised as follows:
"Thus, the final archived snow nitrate, defined as nitrate buried below the photic zone, would be largely impacted by post-depositional processing, which needs to be fully understood to interpret ice-core nitrate records.", line 94-95.

*Lines 103-107: this is a run-on sentence, consider breaking into two sentences*
**Response:** The original sentence has been revised as follows:
"…For example, Geng et al. (2014) found correlations between $\delta^{15}N(NO_3^-)$ and snow accumulation rate across the GISP2 ice core record except in periods with very low snow accumulation rate (<0.08 m ice a$^{-1}$) and high dust concentrations. In the latter situation, $\delta^{15}N(NO_3^-)$ became negatively correlated with dust concentration. These correlations reflect the effect of snow accumulation rate and snow light absorbing impurities on the degree of post-depositional processing, respectively…", line 104-105.

*Line 120: the phrase "…, but less to TCO" doesn't make sense. Please reword.*
**Response:** We have rephrased this sentence as follows:

"…while a recent study suggested the preserved $\delta^{15}N(NO_3^-)$ is more sensitive to snow accumulation rate and light penetration depth than to changes in TCO (Winton et al., 2020). Nevertheless, in periods…", line 119.

*Lines 123-125: the phrase "…, changes in the degree of post-depositional processing and thus the associated isotope effects are expected." doesn't make sense after the first half of the sentence, please reword.*
**Response:** We have rephrased this sentence as follows:
"…Nevertheless, in periods with relatively constant snow accumulation rate but distinct surface actinic flux (e.g., the switch of the polar night and polar day over a year, and the Antarctic ozone hole period), changes in the degree of post-depositional processing and the associated isotope effects should be expected…" line 123.

*Lines 137-138: the phrase "…which is however minimum at Summit given the high snow accumulation" doesn't make sense here, please reword.*
**Response:** We have changed this sentence as follows:
"…In contrast, the model predicted minimum changes in $\Delta^{17}O$ of snow nitrate on both seasonal and annual scales because the photo-driven post-depositional processing affects $\Delta^{17}O$ mainly from the cage effect (i.e., the intermediate photo-products ($NO_2^-$ and $NO_2$) exchange with water oxygen or react with radicals such as OH in snow grains to regenerate nitrate before being emitted to the atmosphere) (McCabe et al., 2005; Meusinger et al., 2014), and the cage effect is minimum at Summit given the high snow accumulation rate", line 137.

*Lines 310-314: you suddenly represent the isotopic values differently here (without parentheses), be consistent throughout the manuscript*
**Response:** We only added parentheses for the values with uncertainty before the unit. For the values here representing the upper and lower range, we don't feel it's necessary to add parentheses to be consistent.

*Lines 858-860: this phrase needs to be reworded*
**Response:** We have changed this sentence as follows:
"…Further numerical modeling is needed to correct the effects of post-depositional processing on $\delta^{15}N(NO_3^-)$, which is critical for the retrieval of information on past atmospheric $NO_x$ emissions using ice core $\delta^{15}N(NO_3^-)$ records (Hasting et al., 2009, 2015).", line 858-861.